# Activation mechanism of PINK1

Zhong Yan Gan[1,2], Sylvie Callegari[1,2], Simon A. Cobbold[1,2], Thomas R. Cotton[1,2], Michael J. Mlodzianoski[1,2], Alexander F. Schubert[3], Niall D. Geoghegan[1,2], Kelly L. Rogers[1,2], Andrew Leis[1,2], Grant Dewson[1,2], Alisa Glukhova[1,2,4,5] & David Komander[1,2]✉

Mutations in the protein kinase PINK1 lead to defects in mitophagy and cause autosomal recessive early onset Parkinson's disease[1,2]. PINK1 has many unique features that enable it to phosphorylate ubiquitin and the ubiquitin-like domain of Parkin[3–9]. Structural analysis of PINK1 from diverse insect species[10–12] with and without ubiquitin provided snapshots of distinct structural states yet did not explain how PINK1 is activated. Here we elucidate the activation mechanism of PINK1 using crystallography and cryo-electron microscopy (cryo-EM). A crystal structure of unphosphorylated *Pediculus humanus corporis* (*Ph*; human body louse) PINK1 resolves an N-terminal helix, revealing the orientation of unphosphorylated yet active PINK1 on the mitochondria. We further provide a cryo-EM structure of a symmetric *Ph*PINK1 dimer trapped during the process of *trans*-autophosphorylation, as well as a cryo-EM structure of phosphorylated *Ph*PINK1 undergoing a conformational change to an active ubiquitin kinase state. Structures and phosphorylation studies further identify a role for regulatory PINK1 oxidation. Together, our research delineates the complete activation mechanism of PINK1, illuminates how PINK1 interacts with the mitochondrial outer membrane and reveals how PINK1 activity may be modulated by mitochondrial reactive oxygen species.

Parkinson's disease (PD) is a neurodegenerative disorder in which progressive motor symptoms are caused by the loss of dopaminergic neurons in the substantia nigra pars compacta[13]. There is presently no treatment that slows or stops PD progression. Although PD is typically a disease of people aged 60 and above, one in ten cases of PD occurs early (<50 years) and can commonly be traced to a mutation in one of around 15 PARK genes[14].

Cell biological insights in the past decade have linked many PARK genes to mitochondrial health. *PARK6* (also known as *PINK1*), which encodes the ubiquitin kinase PINK1, and *PARK2* (also known as *PRKN*), which encodes the E3 ubiquitin ligase Parkin, are frequently mutated in autosomal recessive early onset PD (EOPD) and both function in mitophagy—a cell's disposal mechanism for damaged mitochondria[2,15,16]. The activation of PINK1 is one of the most upstream events in mitophagy. Usually, PINK1 is imported into the mitochondria, cleaved by proteases such as PARL, extracted and degraded by the proteasome[2,16]. Mitochondrial damage stops PARL cleavage and PINK1 is stabilized on the cytosolic face of the mitochondrial outer membrane (MOM), where it is associated with the translocase of the outer membrane (TOM)[17,18]. Stabilized PINK1 is activated, leading to phosphorylation of ubiquitin[4–8] and recruitment and activation of Parkin[19–22]. Active Parkin coats damaged mitochondria with ubiquitin, triggering mitophagy[15,16].

Parkin activation has been structurally resolved[16,21,22], but it has remained unclear how PINK1 is activated. Human PINK1 (*Hs*PINK1) is a divergent Ser/Thr protein kinase that comprises three insertions in the N-lobe as well as a C-terminal extension. The first crystal structures were generated using insect variants that contain only two insertions[23]

(Fig. 1a). *Tribolium castaneum* (*Tc*; flour beetle) PINK1 kinase domain structures after extensive engineering[11,12] adopt a typical bilobal kinase fold with an extended αC helix, but insertions were disordered or had been removed (Extended Data Fig. 1). Our structure of *Ph*PINK1 bound to a ubiquitin mutant (Ub TVLN) and a nanobody[10] revealed how an autophosphorylation event organizes the N-lobe through an unusual 'kinked' αC helix, and orders insertion-3 to form a ubiquitin-binding site[10]. All *Tc*PINK1 and *Ph*PINK1 structures are in active conformations seemingly poised for catalysis[24], yet the differences in conformations and species raised questions of how either related to human PINK1.

Using crystallography and cryo-EM, we resolved the activation mechanism of *Ph*PINK1 showing (1) the unphosphorylated state; (2) a dimerized state capturing *Ph*PINK1 just before *trans*-autophosphorylation; (3) a phosphorylated state undergoing a conformational change in the N-lobe; to arrive at (4) the phosphorylated, ubiquitin-binding state of *Ph*PINK1 (ref. [10]). We show that similar structural transitions apply to *Hs*PINK1 and reveal how PINK1 is not only regulated by phosphorylation, but also by oxidation.

## The structure of unphosphorylated *Ph*PINK1

To obtain a structure of unphosphorylated *Ph*PINK1, we expressed, purified and crystallized the entire cytosolic portion of *Ph*PINK1 with an inactivating D334A mutation of the catalytic base residue located in the HRD motif (Fig. 1a; kinase nomenclature is shown in Extended Data Fig. 1e). The final 3.53 Å crystal structure of unphosphorylated *Ph*PINK1 (Fig. 1b, Extended Data Figs. 1 and 2 and Supplementary Table 1)

[1]Walter and Eliza Hall Institute of Medical Research, Parkville, Victoria, Australia. [2]Department of Medical Biology, University of Melbourne, Melbourne, Victoria, Australia. [3]Medical Research Council Laboratory of Molecular Biology, Cambridge, UK. [4]Drug Discovery Biology, Monash Faculty of Pharmacy and Pharmaceutical Sciences, Monash University, Parkville, Victoria, Australia. [5]Department of Biochemistry and Pharmacology, University of Melbourne, Melbourne, Victoria, Australia. ✉e-mail: dk@wehi.edu.au

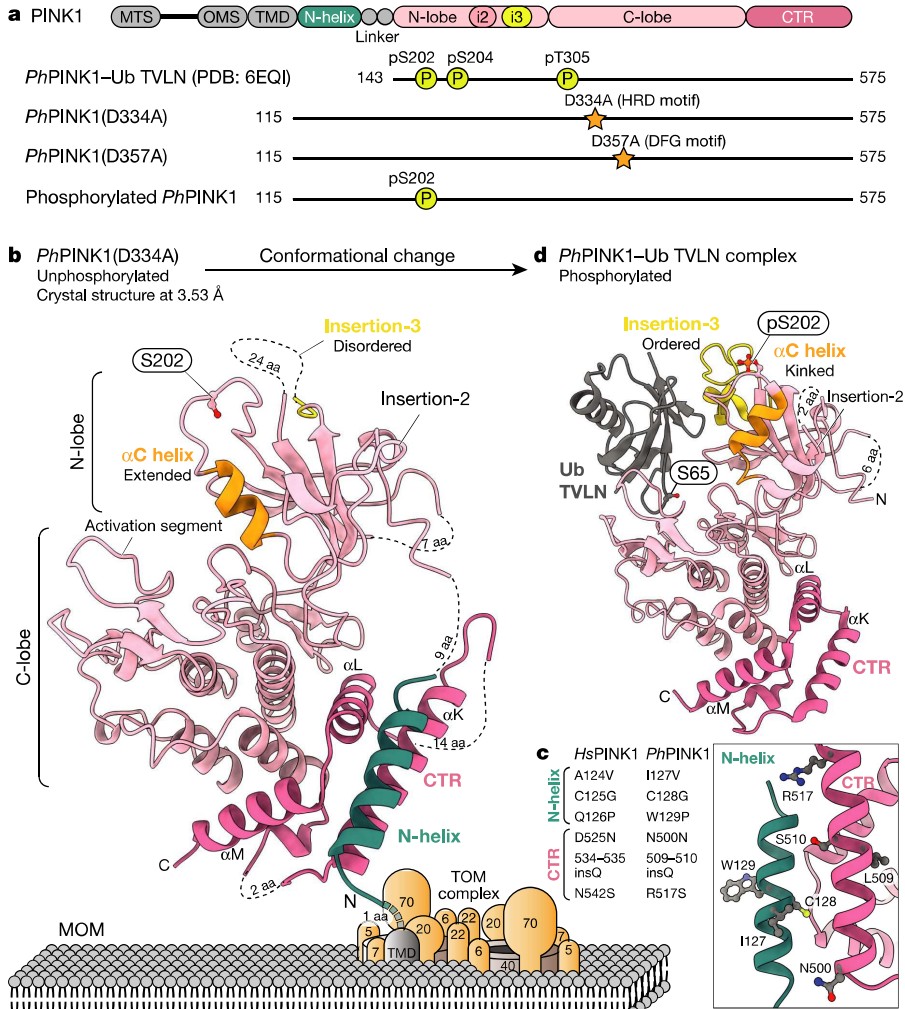

**Fig. 1 | Crystal structure of the cytosolic portion of *Ph*PINK1. a**, *Ph*PINK1 constructs that were used previously (Protein Data Bank (PDB): 6EQI)[10] and structurally characterized in this study. The mitochondrial-targeting sequence (MTS), outer mitochondrial membrane localization signal (OMS), TMD, N-helix and linker, insertion-2 (i2), insertion-3 (i3) and the CTR are indicated. **b**, The crystal structure of unphosphorylated *Ph*PINK1(D334A) (amino acids 115–575), with an extended αC helix and disordered insertion-3. The N-helix (teal) packs against the CTR and directly follows the predicted TMD that interacts with the TOM complex (not to scale) (Extended Data Figs. 1 and 2 and Supplementary Table 1). aa, amino acids. **c**, EOPD mutations in the N-helix and CTR affect the interface. Mutations according to refs. [2,27]. **d**, The previous structure of the *Ph*PINK1–Ub TVLN complex (PDB: 6EQI; Ub TVLN is shown in grey, without the nanobody[10]), with a kinked αC helix, phosphorylated (p) Ser202 and ordered insertion-3.

revealed a kinase fold that is most similar to *Tc*PINK1 structures[11,12] with an extended αC helix and a disordered insertion-3 (Fig. 1b and Extended Data Fig. 1).

A previously undescribed N-terminal region (amino acids 115–142) extends from the kinase N-lobe and adds a helix (amino acids 121–135, hereafter N-helix) to the kinase C-lobe to extend the C-terminal region (CTR) (Fig. 1b). A structurally reminiscent N-helix–CTR region in pseudokinases SgK223 and SgK269 (Extended Data Fig. 2a) forms a dimerization domain[25,26]. By contrast, the N-helix in *Ph*PINK1 generates a monomeric enzyme in solution (Extended Data Fig. 2b–d). The N-helix–CTR interface is a hotspot for EOPD mutations (Fig. 1c) and its structural integrity is crucial for PINK1 function[27]. Moreover, the N-helix immediately follows the predicted transmembrane domain (TMD, predicted amino acids 101–117; Thr119 is the first visible residue in our structure), suggesting a restrained action radius for PINK1 at the MOM (Fig. 1b).

The structural differences between unphosphorylated *Ph*PINK1 (Fig. 1b) and the *Ph*PINK1–Ub TVLN complex[10] (Fig. 1d) resolve the species disparity, but imply that there are large-scale conformational changes in the N-lobe during PINK1 activation.

## *Ph*PINK1 oligomerization enables cryo-EM

A serendipitous observation illuminated the event of *Ph*PINK1 autophosphorylation. *Ph*PINK1 with a different kinase-inactivating mutation, D357A in the kinase DFG motif (Extended Data Fig. 1e), eluted in two distinct peaks on size-exclusion chromatography (SEC) during purification (Fig. 2a and Extended Data Fig. 3a). The major peak corresponded to a molecular mass of 604 kDa as measured using SEC multi-angle light scattering (MALS) and suggested the presence of an oligomerized form of *Ph*PINK1 (molecular mass, 53 kDa). A minor peak of 66 kDa indicated transient dimer formation (Fig. 2a). Furthermore, melting curve analysis of *Ph*PINK1(D357A) revealed a high secondary melting temperature that was absent for monomeric *Ph*PINK1 variants (Extended Data Fig. 3b). Together, these data suggested the formation of a defined *Ph*PINK1 oligomer.

Indeed, negative-stain EM revealed that *Ph*PINK1(D357A) formed highly regular, easily discernible monodisperse particles (Extended Data Fig. 3c). Subsequent cryo-EM analysis resulted in a 2.48 Å density map of a dodecamer of *Ph*PINK1, in which two rings of three dimers form a bagel-like arrangement with $D_3$ symmetry (Fig. 2b, Methods,

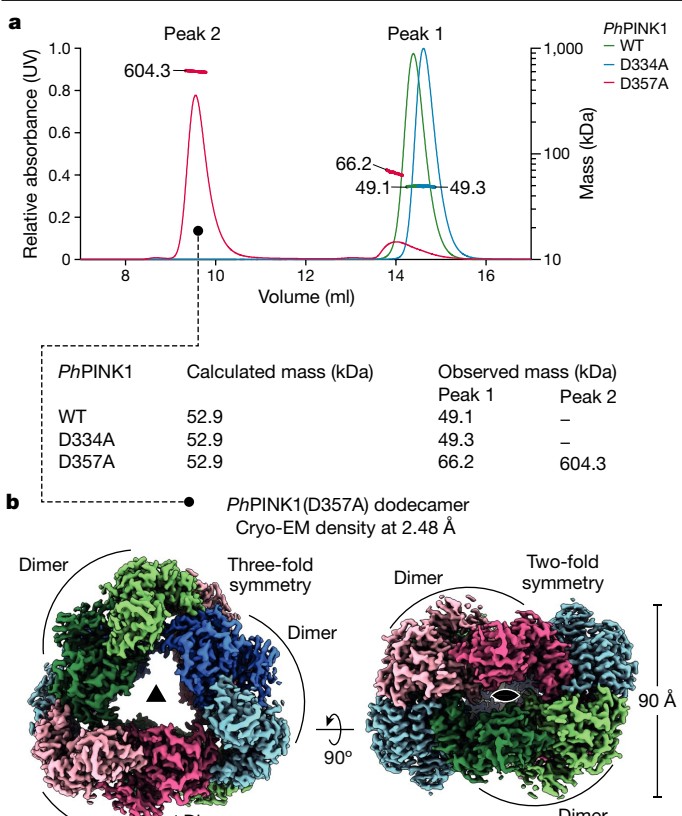

**a**

*Ph*PINK1
— WT
— D334A
— D357A

| *Ph*PINK1 | Calculated mass (kDa) | Observed mass (kDa) | |
|---|---|---|---|
| | | Peak 1 | Peak 2 |
| WT | 52.9 | 49.1 | – |
| D334A | 52.9 | 49.3 | – |
| D357A | 52.9 | 66.2 | 604.3 |

**b**

*Ph*PINK1(D357A) dodecamer
Cryo-EM density at 2.48 Å

**Fig. 2 | Oligomerization of a kinase inactive *Ph*PINK1 enables cryo-EM. a**, SEC–MALS analysis of *Ph*PINK1 (amino acids 115–575) variants (Fig. 1a). Absorbance was measured at a wavelength of 280 nm. Theoretical and observed molecular mass values are indicated. Each protein displayed identical behaviour in at least three SEC runs, and SEC–MALS experiments were performed twice. **b**, Cryo-EM density map for the *Ph*PINK1(D357A) dodecamer at 2.48 Å, indicating monomers in different colours and dimers in shades of the same colour. Left, top view with three-fold symmetry. Right, side view with two-fold symmetry (Extended Data Fig. 3 and Supplementary Table 2).

Extended Data Figs. 3 and 4 and Supplementary Table 2). A *Ph*PINK1 dimer is the smallest unit within the oligomer, and symmetry expansion and local refinement of a masked dimer increased the resolution to 2.35 Å (Extended Data Fig. 3d, f).

We speculated that wild-type (WT) *Ph*PINK1 may not oligomerize due to autophosphorylation[10] and, indeed, when WT *Ph*PINK1 was dephosphorylated with λ-phosphatase (λ-PP), it formed an oligomer; subsequent rephosphorylation of the oligomer destabilized it (Extended Data Fig. 4c; see below). Dodecamer formation seems to be *Ph*PINK1 specific, as equivalent constructs of *Tc*PINK1 did not form oligomers (Extended Data Fig. 4d), and oligomer-enabling interface residues are not conserved in *Tc*PINK1 and *Hs*PINK1 (Extended Data Fig. 4e).

## The structural basis for autophosphorylation

The *Ph*PINK1(D357A) dimer captures the enzyme in the process of *trans*-autophosphorylation, in which a loop of the kinase N-lobe of one monomer is placed into the substrate-binding site of the second monomer in a symmetric contact (Fig. 3a). For this contact to occur, the αC helices are required to be fully extended such that the adjacent Ser202 residue sits in the phosphate-accepting position (Fig. 3a–c). Ser202 forms a hydrogen bond with the catalytic base Asp334 of the HRD motif, typical of kinase-substrate interactions. Modelling ATP into the nucleotide-binding site places the γ-phosphate within 2.3 Å of the Ser202 hydroxyl, poised for phosphoryl transfer[10] (Fig. 3b, c and Extended Data Fig. 5a).

The dimer structure provides profound insights into PINK1 mechanism and regulation. The activation segment (amino acids 357–390) and the adjacent αEF–αF loop (amino acids 393–406) contribute numerous dimer contacts in a symmetric, complementary dimer interface (Fig. 3a (middle) and 3d). Interestingly, the activation segment of PINK1 adopts an identical conformation in all of the structures determined to date (Extended Data Fig. 5b–c).

The dimer structure orients each N-helix–CTR region such that both of the N termini face the MOM (Fig. 3a (bottom)). Strikingly, the distance between TMDs is compatible with placing a TOM complex dimer between the *Ph*PINK1 N termini (Extended Data Fig. 5d). Although the molecular details of this interaction require refinement, it is tempting to speculate that the TOM complex components[17,18,27,28] assist dimerization.

In the *Ph*PINK1 dimer structure, Cys169 at the turn of the kinase P-loop is located within 3.8 Å of its symmetric counterpart (Fig. 3e). Although Cys at this position is unusual in kinases, it is conserved in *Hs*PINK1, but is a Thr in *Tc*PINK1 (Fig. 3f; see below). As a consequence, $H_2O_2$ treatment and non-reducing gel electrophoresis resolves a disulphide-linked dimer of the oligomer-stabilizing *Ph*PINK1(D357A), but not of the WT enzyme, and the C169A mutation prevents oxidative dimerization (Extended Data Fig. 5f). In *Tc*PINK1, engineering an equivalent T172C mutation enables oxidative dimerization (Extended Data Fig. 5f), suggesting similar dimer formation in *Tc*PINK1 (ref. [29]).

We next extended our studies to human PINK1 by expressing *Hs*PINK1 variants in HeLa *PINK1*[−/−] cells. Treatment with oligomycin/antimycin A (OA)— to stabilize *Hs*PINK1—and $H_2O_2$ leads to a discernible disulphide-linked dimer of WT *Hs*PINK1 that is prevented by a C166A mutation in *Hs*PINK1 (equivalent to C169A in *Ph*PINK1) (Fig. 3g). Dimerization is seen readily with WT *Hs*PINK1 without the need of an additional dimer-stabilizing mutation, D384A (equivalent to D357A in *Ph*PINK1) (Fig. 3g).

Importantly, multiple EOPD mutations are located in the dimerization interface (Extended Data Fig. 5g). We used oxidative cross-linking on 11 mutants from patients with EOPD, as in Fig. 3g, in conjunction with autophosphorylation and phosphorylated-ubiquitin generation analysis in HeLa *PINK1*[−/−] cells. Our results show that, despite retaining Cys166, many EOPD mutants are defective in oxidative dimerization, and these mutants display no or distinct autophosphorylation and are deficient in ubiquitin phosphorylation (Extended Data Fig. 5h). We conclude that a similar dimer arrangement is present in *Hs*PINK1 in which it probably facilitates phosphorylation at Ser228 (Extended Data Fig. 6; further discussed below). Oxidative dimerization of *Hs*PINK1 could be useful for future studies.

## Phosphorylated Ser202 unlocks ubiquitin kinase

The *Ph*PINK1 dimer structure with Ser202 in the phospho-acceptor site explains the phosphorylation ability and requirements for PINK1 activation. Phos-tag SDS–PAGE kinase assays showed that unphosphorylated, monomeric *Ph*PINK1(D334A) was rapidly phosphorylated by WT *Ph*PINK1 (Extended Data Fig. 6a, b), and phosphorylation was abrogated by a S202A mutation (Extended Data Fig. 6b). Consistently, Ser202 was the only phosphorylation site of *Ph*PINK1 identified by mass spectrometry[30] (Extended Data Fig. 6c). Furthermore, *Ph*PINK1(D357A) oligomer is not phosphorylated by WT *Ph*PINK1 as its acceptor Ser202 is inaccessible in the oligomer (Extended Data Fig. 6d).

To directly test whether phosphorylation of Ser202 is sufficient to confer the phosphorylation activity of *Ph*PINK1 towards ubiquitin, we purified and dephosphorylated WT *Ph*PINK1 or *Ph*PINK1(S202A). Subsequent co-incubation of *Ph*PINK1 with both ubiquitin and ATP led to the rapid autophosphorylation of WT *Ph*PINK1 and ubiquitin phosphorylation, whereas *Ph*PINK1(S202A) did not autophosphorylate, nor did it phosphorylate ubiquitin (Extended Data Fig. 6e). These results confirm that, in vitro, Ser202 is the main autophosphorylation site of *Ph*PINK1, and that autophosphorylation at Ser202 is necessary and sufficient for *Ph*PINK1 to phosphorylate ubiquitin.

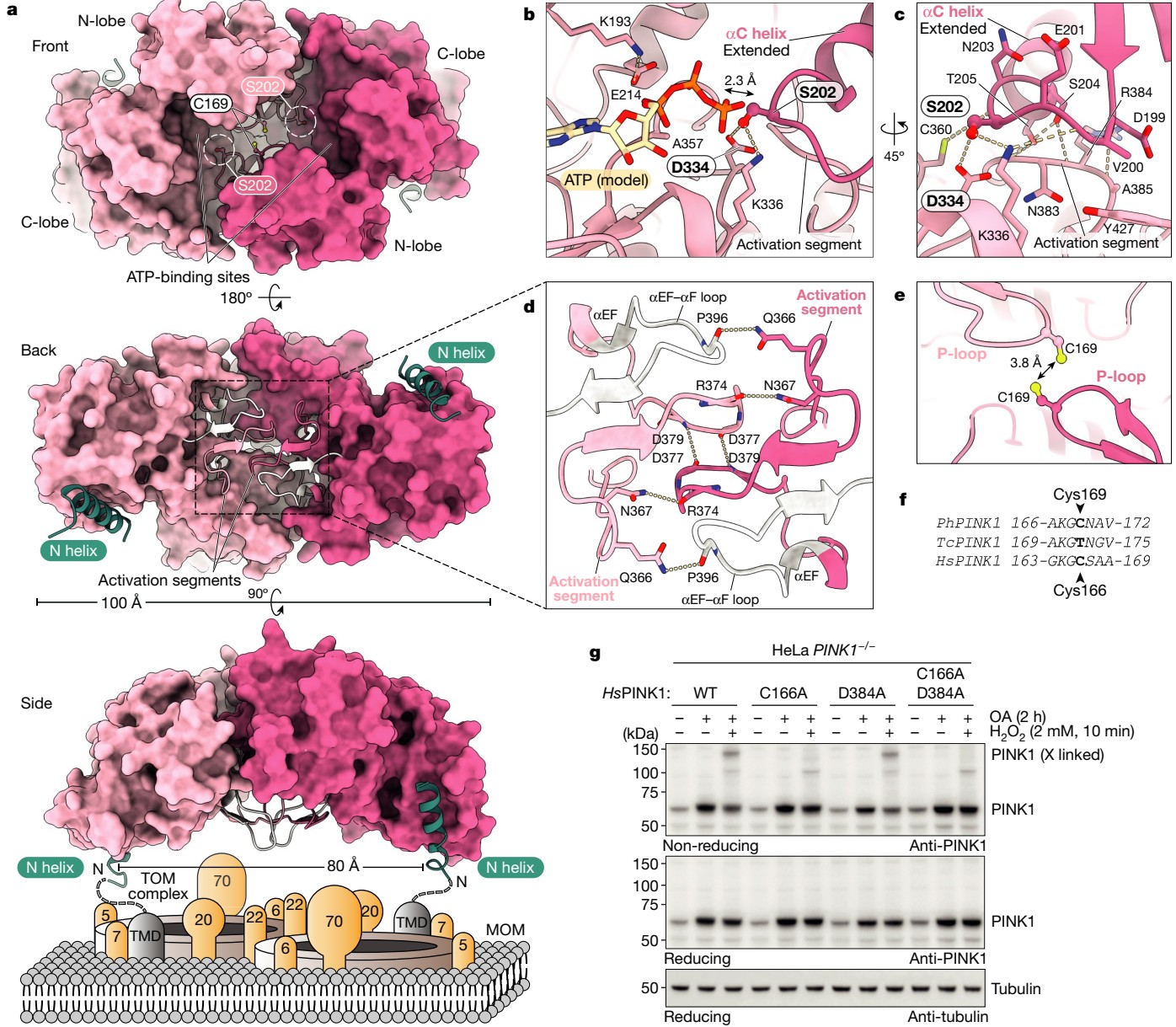

**Fig. 3 | The *Ph*PINK1 dimer before *trans*-autophosphorylation. a**, The structure of the *Ph*PINK1(D357A) dimer in surface representation. Top, front view of the dimer with empty ATP-binding sites. The Ser202-containing loop (Ser202 circled in white) reaches into the acceptor site of the opposing kinase domain. Middle, back view of the dimer, with complementary activation segments (cartoon, coloured). Bottom, side view of the dimer showing the N-helices, indicating how it may sit on the MOM interacting with a TOM complex. A molecular model of the *Ph*PINK1 dimer manually docked onto the TOM complex is shown in Extended Data Fig. 5d. **b–e**, Detailed views in stick representation. The dotted lines indicate hydrogen bonds. **b**, ATP-binding and *trans*-autophosphorylation interactions. ATP was modelled from PDB 2PHK (ref. [42]) as before[10]. **c**, Coordination of the Ser202-containing loop. **d**, Dimer interactions through activation segments and αEF–αF loops. **e**, The close proximity of P-loop Cys169 residues during dimer formation (Extended Data Fig. 5e). **f**, Conservation of *Ph*PINK1 Cys169 in *Hs*PINK1 (Cys166) but not *Tc*PINK1 (Thr172). **g**, Formation of a disulphide-linked *Hs*PINK1 dimer in HeLa cells that were treated with $H_2O_2$. *Hs*PINK1 was expressed in HeLa *PINK1*[−/−] cells and stabilized with OA treatment (Methods). $H_2O_2$ treatment leads to a disulphide-linked *Hs*PINK1 dimer band visualized on a non-reducing gel, which is absent with C166A mutation, suggesting *Hs*PINK1 also dimerizes through Cys166. A putative dimer-trapping mutation in *Hs*PINK1, D384A (D357A in *Ph*PINK1; compare with Fig. 2a), does not further stabilize the dimer. The experiments were performed in biological triplicate with identical results. The uncropped blots are provided in Supplementary Fig. 1.

Experiments expressing *Hs*PINK1 variants in *PINK1*[−/−] HeLa cells, and assessing both autophosphorylation and ubiquitin phosphorylation, confirmed that phosphorylation of Ser228 (equivalent to *Ph*PINK1 Ser202) is necessary and sufficient for *Hs*PINK1 to become a ubiquitin kinase (Extended Data Fig. 6f–i), consistent with previous reports[30–33]. We identify that Ser229 and Ser230 are potential additional PINK1 autophosphorylation sites, but phosphorylation at neither site triggers ubiquitin phosphorylation (Extended Data Fig. 6f–i).

## Capturing PINK1 conformational changes

We next examined whether phosphorylation of Ser202 leads to a conformation change. To resolve this question and the final step of *Ph*PINK1 activation from a canonical kinase to a ubiquitin kinase (Fig. 1), we generated a partially cross-linked WT *Ph*PINK1 oligomer and added $Mg^{2+}$/ATP to phosphorylate each kinase domain (Fig. 4a–c). In the final 3.07 Å cryo-EM map of the dimer (Fig. 4d, Extended Data Fig. 7 and

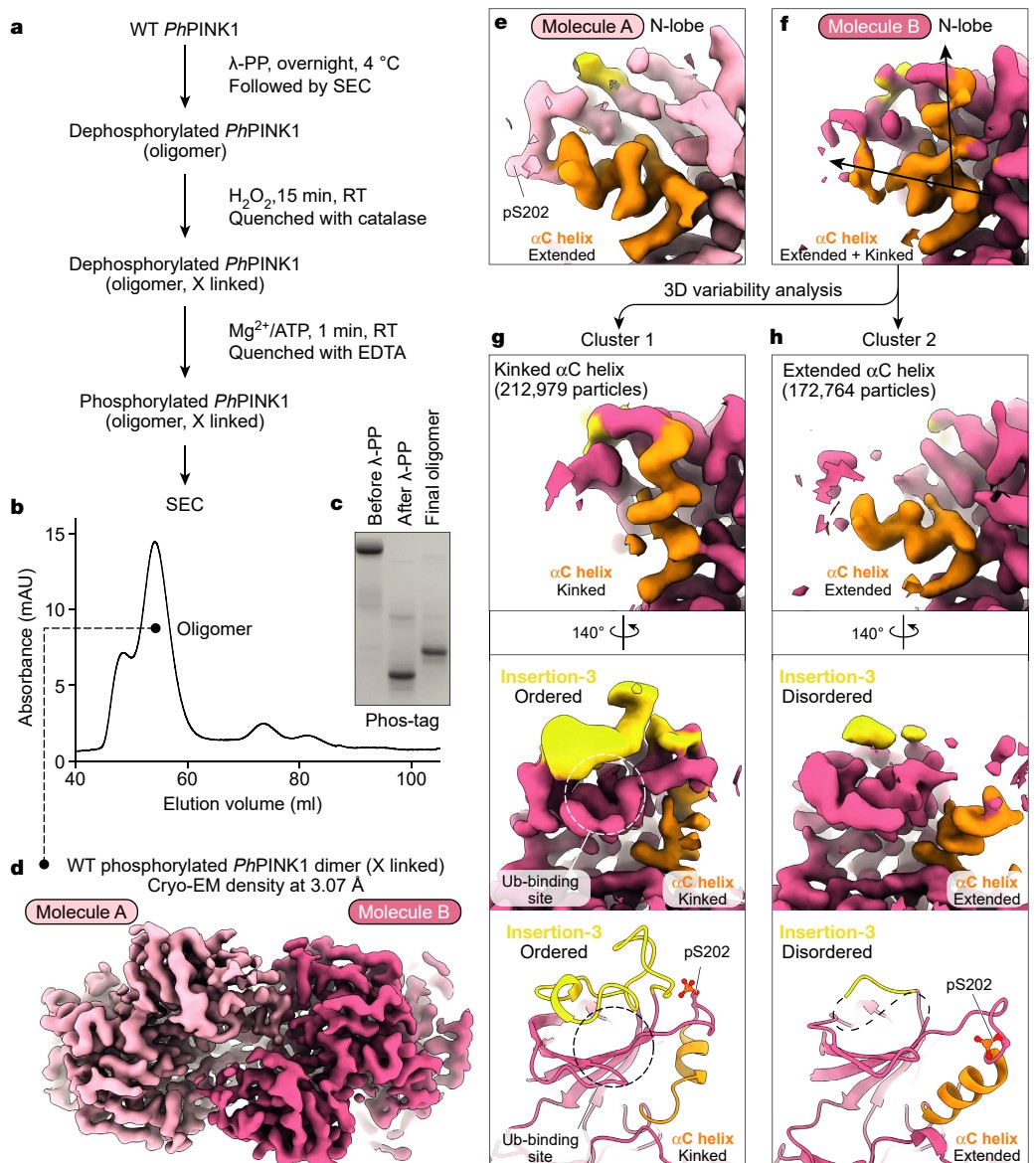

**Fig. 4 | The cryo-EM structure of Ser202-phosphorylated *Ph*PINK1 reveals a conformational change. a**, Flowchart for the generation of the phosphorylated and partially cross-linked WT *Ph*PINK1 dodecamer. RT, room temperature. **b**, Profile of the final SEC run. **c**, Phos-tag analysis of the *Ph*PINK1 species. The final oligomer fraction comprises homogenously phosphorylated *Ph*PINK1 and was used for cryo-EM analysis. The experimental workflow shown in **a**–**c** was performed once in this exact configuration. The uncropped gel is provided in Supplementary Fig. 1. **d**, Cryo-EM density map of the Ser202-phosphorylated WT *Ph*PINK1 dimer at 3.07 Å. A break in symmetry in

the N-lobe is visible (Methods, Extended Data Fig. 7 and Supplementary Table 2). **e**, EM density for the N-lobe of molecule A shows an extended αC helix (orange) with the phosphorylated Ser202 at the tip. **f**, The EM density for the N-lobe of molecule B shows a less-ordered state of the αC helix seemingly in transition. **g**, **h**, 3D variability analysis enabled the clustering of distinct states of the N-lobe in molecule B. **g**, In the first cluster, the αC helix (orange) is kinked, and extra density can be modelled by a poly-Ala model of insertion-3 (yellow). **h**, The second cluster resembles molecule A with an extended αC helix and disordered insertion-3.

Supplementary Table 2), the kinase C-lobe and the oligomer-forming N-helix–CTR region were highly similar to the original, unphosphorylated structure (Fig. 3). The N-lobe of the kinase showed structural differences, including a break in symmetry leading to different states of the αC helix in the individual kinase domains (Fig. 4d–h).

In one conformation (molecule A), the αC helix is extended as in the original dimer, with clear density for the phosphorylated Ser202 that remains in the acceptor site of the phosphorylating domain (Fig. 4e and Extended Data Fig. 7e). The ATP-binding site is empty (the nucleotide was probably washed out in the final SEC step), enabling the phosphate to sit in the acceptor site (Extended Data Fig. 7e).

In the second conformation (molecule B), the αC helix and N-lobe appear to be in transition. The αC helix in some particles adopts an extended conformation, while it is kinked in others (Fig. 4f–h). Using 3D variability analysis in cryoSPARC[34] we were able to visualize individual conformations and cluster them into two distinct αC conformations (Fig. 4g, h). Importantly, in the conformation with a kinked αC helix, insertion-3 density appears and could be interpreted with a poly-Ala model for this region analogous to the previous *Ph*PINK1–Ub TVLN complex[10] (Fig. 4g).

PINK1 ensures that each kinase domain is phosphorylated[18,30,33], explained now by our data (Fig. 4 and Extended Data Fig. 4c).

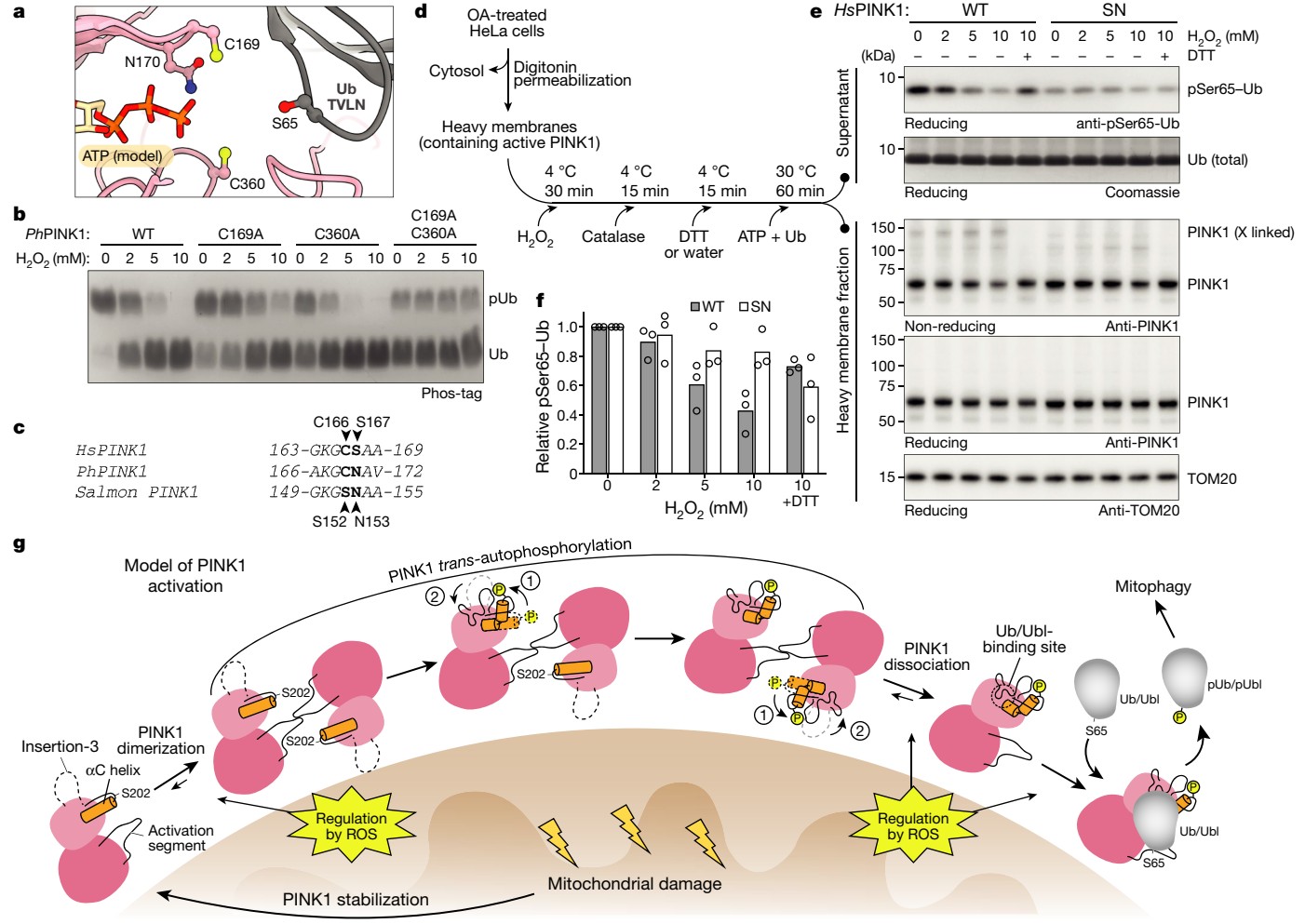

**Fig. 5 | Regulation of PINK1 by oxidation and the model of PINK1 activation.**
**a**, Cys169–which is involved in dimerization (Fig. 3)–and Cys360 line the ATP-binding pocket of *Ph*PINK1 and are also close to the substrate ubiquitin (PDB: 6EQI)[10]. ATP was modelled as in Fig. 3b. **b**, Phos-tag analysis of *Ph*PINK1-mediated ubiquitin phosphorylation, with increasing concentrations of $H_2O_2$. Mutations of Cys169 and Cys360 render *Ph*PINK1 less active but also unresponsive to $H_2O_2$. The experiments were performed in biological triplicate with identical results. The uncropped gel is provided in Supplementary Fig. 1.
**c**, Conservation of Cys169 and its context in *Ph*PINK1, *Hs*PINK1 and salmon (*Salmo salar*) PINK1 (Extended Data Fig. 9a). **d**, The workflow for the experiment in **e**. Details are provided in the Methods. Heavy membranes isolated from OA-treated HeLa *PINK1*$^{-/-}$ cells expressing WT *Hs*PINK1 or *Hs*PINK1(C166S/S167N) (SN, mutating the P-loop to a fish-like sequence to generate active Cys166-lacking *Hs*PINK1) were pretreated with increasing

concentrations of $H_2O_2$ (as indicated in **e**) to oxidize membrane-associated active *Hs*PINK1, before incubation with recombinant ubiquitin and ATP. DTT, dithiothreitol. **e**, Western blotting of the samples generated in **d**, indicating reversible inactivation of *Hs*PINK1 by $H_2O_2$, which was not observed using the Cys166-lacking SN mutant. The experiments were performed in biological triplicate. Uncropped blots are provided in Supplementary Fig. 1. **f**, Quantification of pSer65-Ub band intensities from experiments in **e** and its repeats (*n* = 3; Supplementary Fig. 2). Intensities were adjusted on the basis of PINK1 levels (reducing gel) and normalized to the 0 mM $H_2O_2$ condition for each *Hs*PINK1 variant. Band intensities were quantified using ImageLab (Bio-Rad, v.6.1). **g**, The model for PINK1 activation on the surface of depolarized mitochondria. We expect that αC helix kinking (1) precedes ordering of insertion-3 (2) to form the ubiquitin-binding site. ROS, reactive oxygen species.

We conceptualized dimer stability as a function of intact Ser202–Asp334 contacts (Extended Data Fig. 8), and investigated different scenarios by assessing oligomer formation. Two intact Ser202–Asp334 contacts enable dimerization and oligomerization of unphosphorylated *Ph*PINK1, whereas unphosphorylated *Ph*PINK1(D334A) mutant or Ser202-phosphorylated *Ph*PINK1 is monomeric, as both contacts are disrupted (Extended Data Fig. 8). Hetero-oligomer formation observed from mixing unphosphorylated *Ph*PINK1(D334A) with Ser202-phosphorylated *Ph*PINK1 showed that one intact Ser202–Asp334 contact is sufficient to enable oligomer formation (Extended Data Fig. 8). We conclude that PINK1 dimers remain stable until each kinase domain is phosphorylated.

## Regulation of PINK1 by Cys oxidation

Dimer structures and cross-linking experiments highlighted a possibility for a new mode of PINK1 regulation through oxidation. The cross-linkable Cys169 is one of two conserved and reactive[35] Cys residues in the ATP-binding cleft of PINK1 (Fig. 5a and Extended Data Fig. 9a). Biochemical experiments revealed that oxidation with $H_2O_2$ inhibits ubiquitin kinase activity of phosphorylated, monomeric WT *Ph*PINK1 (Fig. 5b). *Ph*PINK1(C169A) (Cys166 in *Hs*PINK1), *Ph*PINK1(C360A) (Cys387 in *Hs*PINK1) or a double mutant, rendered the kinase less active, but interestingly also rendered it unresponsive to $H_2O_2$ in vitro (Fig. 5b). *Tc*PINK1 containing Thr172 instead of Cys169 is

less responsive to $H_2O_2$ inhibition, similar to *Ph*PINK1(C169A) (compare Extended Data Fig. 9b with Fig. 5b). Importantly, *Ph*PINK1 inhibition by $H_2O_2$ is reversed by dithiothreitol, highlighting the possibility of a regulatory switch (Extended Data Fig. 9c).

The relevance of *Hs*PINK1 Cys166 or Cys387 has not been studied. Consistent with in vitro experiments with *Ph*PINK1, *Hs*PINK1(C166A) or *Hs*PINK1(C387A) had a reduced ability to generate phosphorylated ubiquitin compared with WT *Hs*PINK1 when expressed in *PINK1*[−/−] HeLa cells (Extended Data Fig. 9d). Ectopic expression of YFP–Parkin in HeLa cells lacking endogenous Parkin leads to seemingly higher overall levels of phosphorylated ubiquitin that are reduced with *Hs*PINK1(C166A) or *Hs*PINK1(C387A) (Extended Data Fig. 9e). Consistently, recruitment of YFP–Parkin to the mitochondria is delayed in cells expressing *Hs*PINK1(C166A) or *Hs*PINK1(C387A) compared with in cells expressing WT *Hs*PINK1 (Extended Data Fig. 9f, g and Supplementary Video 1).

Although these results revealed the importance of the Cys residues lining the ATP-binding site of *Hs*PINK1, the limited ubiquitin kinase activity of *Hs*PINK1 Cys mutants (Extended Data Fig. 9d–g) precluded studies on the regulation of *Hs*PINK1 by oxidation. Interestingly, many fish species contain a Ser–Asn motif instead of *Hs*PINK1 Cys166–Ser167 (Fig. 5c and Extended Data Fig. 9a), and indeed S167N mutation rescues the activity of the impaired *Hs*PINK1(C166S) mutant (Extended Data Fig. 9h). The active *Hs*PINK1 variant lacking Cys166, *Hs*PINK1(C166S/S167N) (SN), enabled us to test whether *Hs*PINK1 can also be reversibly oxidized by enriching for mitochondria from OA-treated HeLa cells expressing *Hs*PINK1 variants, subjecting them to oxidation and assessing the phosphorylation of recombinant ubiquitin (Fig. 5d–f). WT *Hs*PINK1 was reversibly inactivated by $H_2O_2$ treatment, whereas the *Hs*PINK1 SN mutant was unresponsive to $H_2O_2$, consistent with experiments with recombinant *Ph*PINK1. Collectively, our data reveal that Cys166 is a site of reversible oxidation in *Hs*PINK1, which would enable reactive oxygen species to regulate PINK1 activity (Fig. 5g).

## Implications for human PINK1

This completes our model of PINK1 activation and regulation (Fig. 5g). PINK1 stabilized on mitochondria is an active protein kinase even in its unphosphorylated state. Dimerization, possibly with the assistance of the TOM complex, enables *trans*-autophosphorylation at *Ph*PINK1 Ser202 (Ser228 in *Hs*PINK1). Our structural work strongly supports the notion that phosphorylation triggers conformational changes in the N-lobe, including kinking of the αC helix and organization of insertion-3. The dimer is destabilized after both copies of PINK1 are phosphorylated, enabling phosphorylated PINK1 to act as a monomeric ubiquitin kinase and Parkin Ubl kinase to initiate mitophagy (Fig. 5g). We further identified an oxidation switch in Cys169 (Cys166 in *Hs*PINK1) that prevents PINK1 activation by potentially preventing dissociation of the autophosphorylated PINK1 dimer and/or hindering ubiquitin access to PINK1 in the phosphorylated, monomeric conformation. A conceptually similar redox switch exists in fructosamine-3-kinases[36] (Extended Data Fig. 9i). Considering that PINK1 is located at one of the hotspots of reactive oxygen species production in the cell, this new potential regulatory mechanism warrants further investigation.

Structure predictions from AlphaFold2 (refs. [37,38]) for *Hs*PINK1 (Extended Data Fig. 10) provide further support for our model of PINK1 activation. AlphaFold2 predicts the kinase in a ubiquitin-binding-competent conformation with a kinked αC helix and ordered insertion-3, suggesting that there is ubiquitin kinase activity even in the absence of PINK1 phosphorylation (Extended Data Fig. 10a). This is questionable as Ser228 phosphorylation is a prerequisite for PINK1 ubiquitin kinase activity (Extended Data Fig. 6). We used ColabFold[39] to predict a dimeric *Hs*PINK1 structure (Extended Data Fig. 10c). The result was almost indistinguishable from the dimer arrangement that we experimentally determined in *Ph*PINK1 (Extended Data Fig. 10d), and our oxidative cross-linking experiments are consistent with the predicted dimer (Fig. 3g).

As *Hs*PINK1 autophosphorylates at Ser228 (Extended Data Fig. 6) and the dimer forms analogously to *Ph*PINK1, we anticipate that unphosphorylated *Hs*PINK1 has an extended αC helix that places Ser228 into the active site of the dimeric molecule to facilitate autophosphorylation. The predicted AlphaFold2 model displays a kinked αC helix and phosphorylated-Ser228-induced organization of insertion-3 and is probably the final phosphorylated state of the ubiquitin kinase PINK1.

Thus, we conclude that the activation mechanism of PINK1 derived here applies across species. Important additional implications include (1) our demonstration that PINK1 is an active kinase without phosphorylation, reopening the investigation for PINK1 substrates and roles before autophosphorylation[40]; (2) the orientational restraints that PINK1 experiences at the MOM, which would limit its activity radius to MOM-proximal substrates; and (3) the observed regulation by oxidation, which warrants further studies. Our mechanistic insights will probably help efforts to use PINK1 as a drug target to stimulate mitophagy and treat PD[41].

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

## Methods

### Cloning

*Ph*PINK1 and *Tc*PINK1 constructs for bacterial expression were cloned into the pOPINK vector[43] using In-Fusion Cloning (Takara Bio) incorporating an N-terminal GST tag and 3C cleavage site. Mutagenesis was performed using the Q5 site-directed mutagenesis kit (NEB).

### Protein expression and purification

GST-tagged *Ph*PINK1 and *Tc*PINK1 constructs were transfected into *Escherichia coli* Rosetta2 (DE3) pLacI cells (Novagen) and cells were grown at 37 °C in 2× YT medium until an optical density at 600 nm ($OD_{600}$) of 0.6–0.8 was reached. Protein expression was induced by the addition of 200 µM IPTG and cultures were incubated overnight at 18 °C. Cells were collected by centrifugation at 5,000$g$ for 15 min at 4 °C and frozen at −80 °C. Cells were thawed and lysed by sonication in purification buffer (25 mM Tris (pH 8.5), 300 mM NaCl, 5% (v/v) glycerol, 10 mM DTT) supplemented with EDTA-free protease inhibitor cocktail tablets (Roche), lysozyme and DNase I. Lysates were clarified by centrifugation at 44,000$g$ for 30 min at 4 °C and the supernatant was incubated with Glutathione Sepharose 4B resin (Cytiva). After washing with purification buffer, the resin was incubated with GST–3C PreScission Protease overnight to cleave the GST tag. The cleaved protein was concentrated and purified by SEC using a HiLoad 16/600 Superdex 200 pg column (Cytiva) in SEC buffer (25 mM Tris (pH 8.5), 150 mM NaCl, 5% (v/v) glycerol, 10 mM DTT). The fractions containing pure protein were pooled, concentrated, flash-frozen in liquid nitrogen and stored at −80 °C. *Ph*PINK1 (amino acids 143–575) and GST–*Ph*PINK1 (amino acids 115–575) were purified as described previously[8,10,21]. For *Tc*PINK1 mutants, additional NaCl was added after SEC to a final concentration of 300 mM NaCl to minimize precipitation during concentration.

His-tagged λ-PP was expressed as *Ph*PINK1 but in BL21(DE3) cells. Cells were lysed by sonication in binding buffer (50 mM Tris (pH 7.4), 500 mM NaCl, 10% (v/v) glycerol) supplemented with EDTA-free protease inhibitor cocktail tablets (Roche), lysozyme, DNase I, and 1 mM PMSF. Lysates were clarified by centrifugation at 44,000$g$ for 30 min at 4 °C and the supernatant was incubated with HisPur Cobalt Resin (Thermo Fisher Scientific). Resin was washed extensively with binding buffer and subsequently eluted with elution buffer (50 mM Tris (pH 7.4), 10% (v/v) glycerol, 250 mM imidazole). Eluted protein was then purified by SEC using a HiLoad 16/600 Superdex 75 pg column (Cytiva) in 50 mM Tris (pH 7.4), 10% (v/v) glycerol followed by anion exchange chromatography using a Resource Q 6 ml column (Cytiva). The fractions containing pure protein were pooled and concentrated. Glycerol was added to the protein to a final concentration of 50% (v/v) glycerol before flash-freezing in liquid nitrogen.

### Crystallization of *Ph*PINK1(D334A)

Crystallization screens were performed at the CSIRO C3 Crystallisation Centre. *Ph*PINK1(D334A) (amino acids 115–575) was crystallized at 1.5 mg ml$^{-1}$ by sitting-drop vapour diffusion against 20% (w/v) polyethylene glycol 3350, 0.2 M triammonium citrate, 0.1 M ammonium sulphate, 0.01 M magnesium chloride from 1:1 protein to mother liquor ratio, in 300 nl drops at 8 °C. Needle-like crystals appeared within a week and grew over the following two weeks. Crystals were cryo-protected in mother liquor diluted with 100% (v/v) ethylene glycol to achieve a final concentration of 25% (v/v) ethylene glycol before vitrification in liquid nitrogen.

### Crystallographic data collection, phasing and refinement

Diffraction data were collected at the Australian Synchrotron (ANSTO) using the MX2 beamline ($\lambda$ = 0.9537 Å, 100 K)[44] and processed using XDSme (v.0.6.5.2)[45]. Data were merged using AIMLESS (v.0.5.21)[46] implemented in CCP4i (v.7.0.001)[47] and molecular replacement was performed using PHASER (v.2.8.3)[48] implemented in Phenix (v.1.19.2-4158)[49] using *Ph*PINK1 from the *Ph*PINK1–Ub TVLN complex (PDB: 6EQI)[10] as the search model. Model building was performed in Coot (v.0.9)[50] and underwent multiple rounds of refinement in Phenix. The N-helix was built de novo. The final model has excellent geometry, with final Ramachandran statistics: 95.09% favoured, 4.65% allowed and 0.26% outliers. Several regions within *Ph*PINK1 could not be modelled due to disorder; these included amino acids 115–118 (N terminus), 138–146 (N-helix linker to the N-lobe), 182–188 (insertion-2), 260–283 (insertion-3), 493–494 and 526–539 (loops in CTR), and 574–575 (C terminus). Data collection and refinement statistics are provided in Supplementary Table 1.

### SEC–MALS

Size-exclusion chromatography multi-angle light scattering (SEC–MALS) experiments were performed using a Superdex 200 Increase 10/300 GL column (Cytiva) coupled with DAWN HELEOS II light scattering detector and Optilab T-rEX refractive index detector (Wyatt Technology). The system was equilibrated in 25 mM Tris (pH 8.5), 150 mM NaCl, 5% (v/v) glycerol, 2 mM TCEP and calibrated using bovine serum albumin (2 mg ml$^{-1}$) before analysis of experimental samples. For each experiment, 100 µl of purified protein (1 mg ml$^{-1}$) was injected onto the column and eluted at a flow rate of 0.5 ml min$^{-1}$. Experimental data were collected and processed using ASTRA (Wyatt Technology, v.7.3.19).

### Thermal denaturation assay

Assays were performed with 4 µM *Ph*PINK1 and 5× SYPRO Orange (Invitrogen) in 25 mM Tris (pH 8.5), 150 mM NaCl, 10 mM DTT at a total volume of 25 µl. Melting curves were recorded in duplicate using a Rotor-Gene Q (Qiagen) with a temperature ramp of 1 °C min$^{-1}$ from 25 °C to 80 °C. Data were collected using the Rotor-Gene Q Series Software (v.2.3.1) and analysed using GraphPad Prism (v.9.0.0). Melting curves were fitted with Boltzmann sigmoidal functions to calculate the melting temperatures.

### Assessment of WT *Ph*PINK1 oligomerization and preparation of monomeric phosphorylated *Ph*PINK1

WT *Ph*PINK1 (amino acids 115–575, autophosphorylated) at 15 mM was dephosphorylated overnight at 4 °C with 7.5 µM λ-PP in SEC buffer supplemented with 2 mM MnCl$_2$, and oligomerization was assessed by SEC using a HiLoad 16/600 Superdex 200 pg column in SEC buffer. To assess the effect of *Ph*PINK1 autophosphorylation on oligomer formation, 10 mM MgCl$_2$ and 10 mM ATP was added to dephosphorylated *Ph*PINK1 for 1 min at room temperature to initiate autophosphorylation. The kinase reaction was quenched with 20 mM EDTA, and the oligomeric state of phosphorylated *Ph*PINK1 was assessed by SEC using a HiLoad 16/600 Superdex 200 pg column in SEC buffer. To prepare monomeric phosphorylated *Ph*PINK1, the fractions corresponding to monomeric protein from the SEC run were pooled, concentrated, flash-frozen in liquid nitrogen and stored at −80 °C.

### Preparation of *Ph*PINK1 and phosphorylated *Ph*PINK1 oligomer for EM analysis

*Ph*PINK1(D357A) (amino acids 115–575) was purified as described above but with the following modifications. An additional anion exchange step using a Mono Q 5/50 GL column (Cytiva) was included before a final SEC run and, during SEC, the protein was buffer exchanged into either SEC buffer (for negative stain EM) or glycerol-free SEC buffer (for cryo-EM). In both cases, *Ph*PINK1(D357A) elutes largely as an oligomer close to but not overlapping with the void volume of a HiLoad 16/600 Superdex 200 pg column. SDS–PAGE analysis of individual fractions was used and oligomer-containing fractions were pooled if more than 95% pure.

To generate the phosphorylated *Ph*PINK1 oligomer, WT *Ph*PINK1 (amino acids 115–575) was first dephosphorylated as described above. Dephosphorylated *Ph*PINK1 was purified by SEC using a HiLoad 16/600

Superdex 200 pg column in DTT-free SEC buffer. The fractions containing oligomeric *Ph*PINK1 were pooled and immediately treated with 2 mM $H_2O_2$ for 15 min at room temperature to cross-link the dimer and stabilize the oligomer, and the reaction was quenched with 10 U ml$^{-1}$ catalase (Sigma-Aldrich). Homogeneous phosphorylation at Ser202 was achieved by the addition of 10 mM $MgCl_2$ and 10 mM ATP for 1 min at room temperature followed by the addition of 20 mM EDTA to inactivate the kinase and residual λ-PP. Oligomeric phosphorylated *Ph*PINK1 was purified in a final SEC run on a HiLoad 16/600 Superdex 200 pg column in glycerol- and DTT-free SEC buffer (25 mM Tris (pH 8.5), 150 mM NaCl). The fractions containing the oligomer were pooled, concentrated, flash-frozen in liquid nitrogen and stored at −80 °C.

### Negative-stain EM

Negative stain EM data collection was performed at the Ian Holmes Imaging Centre at the Bio21 Molecular Science and Biotechnology Institute, University of Melbourne. Samples were diluted to 0.005–0.01 mg ml$^{-1}$ and applied to a glow-discharged carbon-coated Cu grid (200 mesh). After 60 s, the solution was blotted off and the grid was stained in 0.8% (w/v) uranyl formate solution for 30 s. Excess solution was blotted off and was followed by two washes in water. The grids were imaged at room temperature using the Talos L120C electron microscope at a magnification of ×52,000 and a defocus value of around −1 μm with a pixel size of 2.44 Å. Particle picking, extraction and 2D classification were performed using RELION (v.3.1)[51].

### Cryo-EM sample preparation and data acquisition

Cryo-EM data collection was performed at the Ian Holmes Imaging Centre at the Bio21 Molecular Science and Biotechnology Institute, University of Melbourne. Grids were prepared by dispensing 4 μl of *Ph*PINK1(D357A) (2 mg ml$^{-1}$) or phosphorylated *Ph*PINK1 (1.3 mg ml$^{-1}$) onto a glow-discharged UltrAuFoil R1.2/1.3 (300 mesh) or Quantifoil R1.2/1.3 grid (200 mesh), respectively, at 4 °C and 100% humidity. Grids were blotted for 4 s with a nominal blot force of −1 before plunging into liquid ethane using a Vitrobot Mark IV (Thermo Fisher Scientific). Data for *Ph*PINK1(D357A) were collected on a Thermo Scientific Titan Krios G4 microscope equipped with a Gatan K3 detector and Biocontinuum energy filter at 300 keV. The acquisition was performed in EFTEM NanoProbe mode, nominal magnification ×105,000, zero-loss slit 10 eV in correlated double-sampling mode (formerly 'super-resolution mode', 0.4165 Å px$^{-1}$), with a total exposure of 50 e Å$^{-2}$ over 40 frames. A total of 644 image stacks was collected. The data for phosphorylated *Ph*PINK1 were collected on the Talos Arctica microscope equipped with a Gatan K2 detector and Bioquantum energy filter at 200 keV. The acquisition was performed in EFTEM NanoProbe mode, nominal magnification ×165,000, zero-loss slit 10 eV and electron-counting mode (0.78 Å px$^{-1}$), with a total exposure of 50 e Å$^{-2}$ over 40 frames. A total of 1,717 image stacks was collected. All data were collected using EPU software (Thermo Fisher Scientific, v.2.9).

### Cryo-EM image processing and model building

All data processing was performed using cryoSPARC (v.3.2.0)[52].

**_Ph_PINK1(D357A) dataset.** Image stacks were corrected for beam-induced motion using patch motion, Fourier cropped to 0.833 Å px$^{-1}$ and used for CTF estimation using patch CTF algorithm in cryoSPARC[52]. To create templates for particle picking, the micrographs were initially picked using blob picker, followed by particle extraction and 2D classification. Representative classes were used for template picking. Particles (310,371) were picked, extracted using a 380 px box size and processed for 2D classification. The best 235,948 particles were used to create an ab initio model. The 31,080 particles from the 2D classes, not resembling *Ph*PINK1, were used to create a 'junk' ab initio model used as a trap for further data clean-up. Good particles were further cleaned using several rounds of heterogeneous refinement

procedures first using the good and junk ab initio models followed by increasingly higher-resolution templates. The final set consisted of 216,021 particles and reached 2.48 Å using homogeneous refinement protocol with imposed $D_3$ symmetry. To perform local refinement of the *Ph*PINK1 dimer, the symmetry was expanded to $C_1$, yielding 1,295,406 particles. The *Ph*PINK1 dimer was modelled into the dodecamer map (see below), converted into a volume using UCSF Chimera[53], low passed to 12 Å and used to create a soft padded mask in EMAN2 ([54]). This mask was used for local refinement of the *Ph*PINK1 dimer, yielding a 2.35 Å map of the *Ph*PINK1 dimer.

**Phosphorylated _Ph_PINK1 dataset.** The initial data processing steps were similar to those for *Ph*PINK1(D357A). In brief, template picking yielded 205,887 particles and particle extraction was performed with a 440 px box size at 0.78 Å px$^{-1}$. 2D classification yielded 139,028 particles, and heterogeneous refinement reduced the dataset to 89,061 particles. Homogeneous refinement protocol with imposed $D_3$ symmetry resulted in a 3.11 Å map. Symmetry expansion and local refinement yielded a 3.07 Å map of the phosphorylated *Ph*PINK1 dimer from 543,366 particles. To separate individual conformations of the αC helix in molecule B, we performed 3D variability analysis[34] in cluster mode with a mask around insertion-2, insertion-3 and the αC helix. Local refinement of cluster 1 (212,979 particles) using the dimer mask yielded a 3.25 Å map with a predominantly kinked αC helix. Local refinement of cluster 2 (172,764 particles) using the dimer mask yielded a 3.28 Å map with a predominantly extended αC helix. Cluster 3 (148,623 particles) represented *Ph*PINK1 states that could not be easily identified. All of the maps were processed for local-resolution estimation and local-resolution-based filtering using internal cryoSPARC (v.3.2.0) algorithms[52].

**Model building and refinement.** The crystal structure of *Ph*PINK1 from the *Ph*PINK1–Ub TVLN complex (PDB: 6EQI)[10] was used as the initial model and was docked using UCSF Chimera (v.1.14)[53] into to density corresponding to a monomer within the *Ph*PINK1(D357A) dodecamer. Manual rebuilding of the model was performed in Coot (v.0.9)[50] and the additional N-helix included within the construct was built de novo. The model underwent multiple rounds of refinement using real space refine in Phenix (v.1.19.2-4158)[49]. To generate a model of the dimer, the model was rebuilt and refined in one monomer, then duplicated and realigned into density of the opposing monomer, followed by further rounds of rebuilding and refinement. Multiplication of the *Ph*PINK1 dimer into the entire dodecamer map enabled generation of a dodecamer model. For dodecamer refinement, non-crystallography symmetry (NCS) restraints were used during refinement using real space refine, using two NCS groups corresponding to the two chains of the dimer. Owing to the low-resolution density corresponding to insertion-3 in the 'kinked αC' dimer (Fig. 4g and Extended Data Fig. 7c), an atomic model of insertion-3 could not be built de novo. As the shape of the density was reminiscent of the ordered insertion-3 from the *Ph*PINK1–Ub TVLN complex[10], insertion-3 (amino acids 259–280) was taken from the *Ph*PINK1–Ub TVLN complex, rigid-body fitted into the density in Coot (v.0.9), stubbed at the Cβ carbons and merged with the rest of the model. This model was then passed once through real space refine in Phenix.

Data collection and refinement statistics are provided in Supplementary Table 2. All of the models were validated using MolProbity (v.4.5.1)[55]. Structures were visualized and figures were generated in UCSF ChimeraX (v.1.1.1)[56].

### In vitro $H_2O_2$ cross-linking assays

In vitro disulphide-linkage assays were performed at 22 °C by incubating 1.5 μM *Ph*PINK1 or *Tc*PINK1 with 2 mM $H_2O_2$ in 25 mM Tris (pH 7.4), 150 mM NaCl. At the indicated timepoints, the reaction was quenched in NuPAGE LDS sample buffer (Invitrogen), run on reducing

or non-reducing NuPAGE 4–12% Bis-Tris gels (Invitrogen) and stained with InstantBlue Coomassie Protein Stain (Abcam).

## In vitro kinase and oxidation assays

All $Ph$PINK1 autophosphorylation assays were performed in phosphorylation buffer (25 mM Tris (pH 7.4), 150 mM NaCl, 10 mM $MgCl_2$, 1 mM DTT) with 1.5 μM GST–$Ph$PINK1 (amino acids 115–575, WT, autophosphorylated) and 15 μM $Ph$PINK1 (amino acids 115–575) D334A or D357A (substrate). Reactions were started by the addition of 10 mM ATP and incubated at 22 °C for the indicated times.

For the simultaneous $Ph$PINK1 autophosphorylation and ubiquitin phosphorylation assay in Extended Data Fig. 6e, dephosphorylated $Ph$PINK1 (amino acids 119–575, oligomerization deficient) was first prepared by incubating 15 μM $Ph$PINK1 (amino acids 119–575) WT or S202A overnight at 4 °C with 7.5 μM λ-PP in SEC buffer supplemented with 2 mM $MnCl_2$, then purified by SEC. Kinase reactions were carried out as described above using 1.5 μM dephosphorylated $Ph$PINK1 (amino acids 119–575) and 15 μM ubiquitin.

Oxidation-coupled ubiquitin phosphorylation assays were performed in DTT-free phosphorylation buffer using 1.5 μM $Ph$PINK1 (amino acids 115–575) and 15 μM ubiquitin. $Ph$PINK1 was incubated with the indicated concentrations of $H_2O_2$ for 15 min at 4 °C, and the kinase reaction was initiated by the addition of 10 mM ATP and 15 μM ubiquitin and incubated at 22 °C for 3 h. To assess reversible $Ph$PINK1 oxidation, the indicated concentrations of $H_2O_2$ were added and incubated for 1 h at 4 °C, followed by the addition of 10 U $ml^{-1}$ catalase (Sigma-Aldrich) for 15 min to quench $H_2O_2$. The samples were divided and 10 mM DTT was added to one half to reverse $Ph$PINK1 oxidation. The kinase reaction was initiated by the addition of 10 mM ATP and 15 μM ubiquitin and incubated at 22 °C for 3 h.

All kinase reactions were quenched in SDS sample buffer (66 mM Tris (pH 6.8), 2% (w/v) SDS, 10% (v/v) glycerol, 0.005% (w/v) bromophenol blue) and run on custom-made 7.5% (for $Ph$PINK1 autophosphorylation) or 17.5% (for ubiquitin phosphorylation) Phos-tag gels (reducing) containing 50 μM Phos-tag Acrylamide AAL-107 (Wako) and 100 μM $MnCl_2$. Gels were stained with InstantBlue Coomassie Protein Stain (Abcam). Note that, in Phos-tag gels, phosphorylated proteins run markedly slower than unphosphorylated proteins, and no longer just according to their molecular mass. Therefore, size markers are not provided for the Phos-tag gels.

## $Ph$PINK1 phosphosite identification by mass spectrometry

$Ph$PINK1(D334A) phosphorylation was performed as described above, but with 10 mM DTT. The samples were run on SDS–PAGE and the band corresponding to $Ph$PINK1(D334A) was excised and destained twice with 50 mM ammonium bicarbonate/50% (v/v) acetonitrile. After dehydration with 100% (v/v) acetonitrile, the samples were reduced (10 mM TCEP for 30 min), alkylated (40 mM chloroacetamide for 30 min) and digested overnight (15 ng $μl^{-1}$ of TPCK-treated trypsin at 37 °C). Peptides were extracted twice with 60% (v/v) acetonitrile/0.1% (v/v) formic acid and analysed on a timsTOFII pro mass spectrometer (Bruker) with PASEF-MS acquisition. Peptides were separated using a 90 min gradient (solvent A, 0.1% (v/v) formic acid; solvent B, 99.9% (v/v) acetonitrile/0.1% (v/v) formic acid) on a C18 analytical column (Aurora Series Emitter Column, AUR2-25075C18A 25 cm × 75 μm × 1.6 μm, IonOpticks). Data were searched using MaxQuant (v.1.6.17.0) at a 1% false-discovery rate, with oxidation and phosphorylation as variable modifications. The final MS2 spectra were reproduced in Skyline Daily (v.21.1.1.198)[57].

## Analytical SEC to assess $Ph$PINK1 hetero-oligomerization

Analytical SEC was performed using a Superdex 200 Increase 3.2/300 column (Cytiva) equilibrated in SEC buffer. $Ph$PINK1 (amino acids 115–575) D334A, D357A and monomeric phosphorylated $Ph$PINK1 (amino acids 115–575, prepared as described above) were each, or in combination, diluted to 2 mg $ml^{-1}$ (per protein) in SEC buffer and incubated

overnight at 4 °C. 50 μl protein was loaded onto the column per run and eluted at 0.04 ml $min^{-1}$ flow rate.

## Cell culture and constructs

HeLa $PINK1^{-/-}$ cells were a gift from M. Lazarou (Monash University) and were authenticated at the Garvan Molecular Genetics facility using short tandem repeat profiling. Cells were cultured in DMEM supplemented with 10% (v/v) FBS (Gibco or Sigma-Aldrich), penicillin–streptomycin, and maintained at 37 °C and 5% $CO_2$. Cells were also screened routinely for mycoplasma contamination using the MycoAlert Mycoplasma Detection Kit (Lonza). To limit the level of $Hs$PINK1 overexpression, the pcDNA5/FRT/TO plasmid was modified using the Q5 site-directed mutagenesis kit (NEB) to generate a 539 bp deletion in the CMV promoter (CMVd3)[58], hereafter referred to as pcDNA5$^{d3}$. The full-length, WT $Hs$PINK1 sequence was inserted into the BamHI site of pcDNA5$^{d3}$ using In-Fusion Cloning (Takara Bio) and used for transient transfections. For stable $Hs$PINK1 expression, the $Hs$PINK1 sequence was inserted into the BamHI and NheI of the lentiviral pFU PGK Hygro (pFUH) plasmid using InFusion Cloning. All of the $Hs$PINK1 mutants were generated using the Q5 site-directed mutagenesis kit (NEB).

## Generation of stable cell lines

Stable YFP–Parkin and $Hs$PINK1-expressing HeLa $PINK1^{-/-}$ cell lines were generated using retroviral transduction of a pBMN-YFP-Parkin construct (gift from R. Youle; Addgene plasmid, 59416)[59] followed by lentiviral transduction of pFUH-$Hs$PINK1 WT or mutant constructs. For imaging, TOM20-Halo was incorporated by retroviral transduction of a pMIH-TOMM20-Halo construct (gift from B. Kile; Addgene plasmid, 111626)[60]. All cells were selected by fluorescence sorting or by antibiotic selection.

## Lattice light-sheet microscopy

Before imaging, cells were stained overnight using the JF646 HaloTag ligand according to the manufacturer's instructions (Promega), then treated with 10 μM oligomycin and 4 μM antimycin A (OA) to depolarize mitochondria. Time-lapse live-cell data were acquired using a Lattice Light Sheet 7 (Zeiss, pre-serial). Light sheets (488 nm and 633 nm) of length 30 μm with a thickness of 1 μm were created at the sample plane using a ×13.3/0.44 NA objective. Fluorescence emission was collected via a ×44.83/1 NA detection objective. Aberration correction was set to a value of 182 to minimize aberrations as determined by imaging the Point Spread Function using 170 μm fluorescent microspheres at the coverslip of a glass-bottom chamber slide. Resolution was determined to be 454 nm (lateral) and 782 nm (axial). Data were collected with a range of frame rates of 16–20 ms and a $z$-step of 300 nm. Light was collected through a multi-band stop, LBF 405/488/561/633, filter. Images were collected at 37 °C and 5% $CO_2$.

## Lattice light-sheet microscopy data analysis

Images taken using the Zeiss Lattice Light Sheet were processed in ZEN (Zeiss, v.3.5) using Zeiss's deskew and deconvolution modules using a constrained iterative algorithm and 20 iterations. Maximum-intensity projections of the lattice videos were generated using Fiji (ImageJ1.53k). To assess YFP–Parkin translocation, YFP–Parkin foci were manually counted. The time at which each cell began to show YFP–Parkin foci was recorded and the data were used to graph the cumulative fraction of cells that displayed YFP–Parkin foci over time. A linear fit was applied to the cumulative curves from the range of 20–80% of the translocated cells and used to calculate the time (min) at which 50% of cells exhibited YFP–Parkin translocation. A two-sided, two-sample Kolmogorov–Smirnov test was used to compare the cumulative distribution of the two curves. A MATLAB script (Supplementary Data) was used to perform the Kolmogorov–Smirnov test. GraphPad Prism (v.9.0.0) was used to generate the graphs.

## Transient transfection and western blotting

Cells were seeded in six-well plates one day before Lipofectamine 3000-mediated transfection of 1.5 μg pcDNA5[d3]-HsPINK1 WT or mutants. Then, 24 h after transfection, cells were treated with 10 μM oligomycin and 4 μM antimycin A (OA) for 2 h to depolarize mitochondria and induce HsPINK1 accumulation. To induce the formation of disulphide-linked HsPINK1 dimer, 2 mM $H_2O_2$ was added to the culture medium for 10 min before cell lysis. For stable YFP–Parkin/HsPINK1-expressing cell lines, cells were seeded two days before OA treatment and cell lysis. Cells were lysed directly in SDS sample buffer and run on reducing (or non-reducing for assessment of disulphide-linked PINK1 species) NuPAGE 4–12% Bis-Tris gels (Invitrogen). For Phos-tag western blots, the samples were run on custom-made 7.5% Phos-tag gels (reducing) containing 50 μM Phos-tag Acrylamide AAL-107 and 100 μM $MnCl_2$, and the gels were washed three times for 10 min in 10 mM EDTA followed by 10 min in water before transfer. Protein was transferred to PVDF membranes using the Trans-Blot Turbo Transfer System (Bio-Rad). Membranes were blocked in 5% (w/v) milk powder in Tris-buffered saline containing 0.1% (v/v) Tween-20 (TBS-T) for 1 h, then incubated in primary antibodies (containing 3% (w/v) bovine serum albumin and 0.02% (w/v) sodium azide) overnight at 4 °C. The primary antibodies used were as follows: rabbit anti-PINK1 D8G3 1:1,000 (Cell Signaling Technology, 6946, 5), mouse anti-Parkin Prk8 1:1,000 (Cell Signaling Technology, 4211, 7), rabbit anti-phosphorylated ubiquitin (Ser65) 1:1,000 (Millipore, ABS1513-I, 3117322), rabbit anti-TOM20 FL-145 1:1,000 (Santa Cruz Biotechnology, sc-11415, D1613). Membranes were washed with TBS-T and incubated in goat anti-rabbit HRP-conjugated secondary antibodies (1:5,000, SouthernBiotech, 4010-05, A4311-TF99D) or goat anti-mouse HRP-conjugated secondary antibodies (1:5,000, SouthernBiotech, 1030-05, E2518-Z929D) for 1 h at room temperature before washing with TBS-T and detection using the ChemiDoc (Bio-Rad) after HRP substrate incubation. For tubulin blots, the membranes were incubated in hFAB rhodamine anti-tubulin antibodies (1:5,000; Bio-Rad, 12004165) for 1 h at room temperature or overnight at 4 °C followed by washing in TBS-T and detection using the ChemiDoc (Bio-Rad).

## Subcellular fractionation and ubiquitin phosphorylation assay

HeLa $PINK1^{-/-}$ cells ($1 \times 10^6$) were seeded in 10 cm dishes. The next day, cells were transiently transfected with 5 μg pcDNA5[d3]-HsPINK1 WT or C166S/S167N (SN) and, the next day, treated with OA for 2 h to accumulate HsPINK1. Cells were permeabilized in 1 ml fractionation buffer (20 mM HEPES (pH 7.4), 250 mM sucrose, 50 mM KCl, 2.5 mM $MgCl_2$) supplemented with 0.025% (w/v) digitonin, protease inhibitor cocktail tablet (Roche) and PhosSTOP (Roche) for 20 min at 4 °C. Heavy membranes were pelleted by centrifugation for 5 min at 14,000$g$ at 4 °C, followed by washing with fractionation buffer. Heavy membrane pellets were resuspended in 100 μl fractionation buffer and divided into 50 μl fractions. Each fraction was incubated with the indicated concentrations of $H_2O_2$ with gentle agitation for 30 min at 4 °C. Catalase (10 U ml$^{-1}$; Sigma-Aldrich) was added for 15 min to quench $H_2O_2$, and 10 mM DTT was added to select fractions for 15 min to study reversible oxidation. To initiate the kinase reaction, 10 mM ATP and 15 μM ubiquitin was added and the mixture was incubated 1 h at 30 °C with agitation. Heavy membranes (containing HsPINK1) and supernatant (containing Ub) were separated by centrifugation and samples of both were prepared in SDS sample buffer. Western blotting was performed as described above. ImageLab (Bio-Rad, v.6.1) was used to quantify band intensities for the phosphorylated monoubiquitin band, from three independent experiments (Fig. 5e and Supplementary Fig. 2).

## AlphaFold structure prediction

The structure of HsPINK1 as predicted by AlphaFold2 was obtained from the AlphaFold Protein Structure Database (https://alphafold.ebi.ac.uk/)[37,38]. To predict the structure of Parkin Ubl-bound HsPINK1 and dimerized HsPINK1, we used the ColabFold Google Colab notebook called AlphaFold2_complexes[39]. The predicted model with the highest lDDT score is shown.

## Reporting summary

Further information on research design is available in the Nature Research Reporting Summary linked to this paper.

## Data availability

The coordinates and crystallographic structure factors for PhPINK1 (D334A) have been deposited at the PDB under accession code 7T3X, and EM models and maps under accession codes 7T4M (PhPINK1(D357A) dodecamer; Electron Microscopy Data Bank (EMDB): EMD-25680), 7T4N (PhPINK1(D357A) dimer; EMDB: EMD-25681), 7T4L (WT phosphorylated PhPINK1 dimer with extended αC helix in chain B; EMDB: EMD-25679) and 7T4K (WT phosphorylated PhPINK1 dimer with kinked αC helix in chain B; EMDB: EMD-25678). We also deposited the map for WT phosphorylated PhPINK1 dimer before 3D variability analysis (EMDB: EMD-25677). Uncropped versions of all gels and blots are provided in Supplementary Fig. 1. Source data are provided with this paper.

## Code availability

The MATLAB script that was used to perform the Kolmogorov–Smirnov test to analyse lattice light-sheet imaging data is provided in the Supplementary Data.

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

**Acknowledgements** We thank the members of the Ubiquitin Signalling Division at WEHI and I. Lucet (WEHI) and M. Lazarou (Monash) for reagents, advice and comments on the manuscript; J. Newman (CSIRO C3 Crystallisation Centre) for help with crystallization; and the staff at the WEHI Information Technology Services and the WEHI Research Computing Platform for providing facilities and support. This work was funded by WEHI, an NHMRC Investigator Grant (GNT1178122 to D.K.), the Michael J. Fox Foundation and Shake-It-Up Australia (to D.K. and G.D.). A.G. is a CSL Centenary Fellow. G.D. is supported by a fellowship from the Bodhi Education

Fund and NHMRC Ideas grant (GNT2004446). Z.Y.G. is supported by an Australian Government Research Training Program Scholarship. Research was further supported by an NHMRC Independent Research Institutes Infrastructure Support Scheme grant (361646) and Victorian State Government Operational Infrastructure Support grant. This research was undertaken in part using the MX2 beamline at the Australian Synchrotron, part of ANSTO, and made use of the Australian Cancer Research Foundation (ACRF) detector. We also acknowledge use of the Titan Krios and other facilities at the Ian Holmes Imaging Centre at the Bio21 Molecular Science and Biotechnology Institute.

**Author contributions** D.K. conceived the project. Z.Y.G. designed and performed all experiments. S.C. and G.D. contributed to the design of cell biology experiments. D.K. and T.R.C. contributed to crystallography. S.A.C. performed mass spectrometry experiments. T.R.C. performed SEC–MALS experiments. S.C., M.J.M., N.D.G. and K.L.R. performed and analysed lattice light sheet microscopy experiments. A.F.S. contributed ideas, advice and preliminary data. A.L. and A.G. performed EM data collection, A.G. performed data processing and map calculations and Z.Y.G. performed model building. D.K. and Z.Y.G. analysed the data and wrote the manuscript with contributions from all of the authors.

**Competing interests** D.K. serves on the Scientific Advisory Board of BioTheryX Inc. The other authors declare no competing interests.

## Additional information
**Correspondence and requests for materials** should be addressed to David Komander.

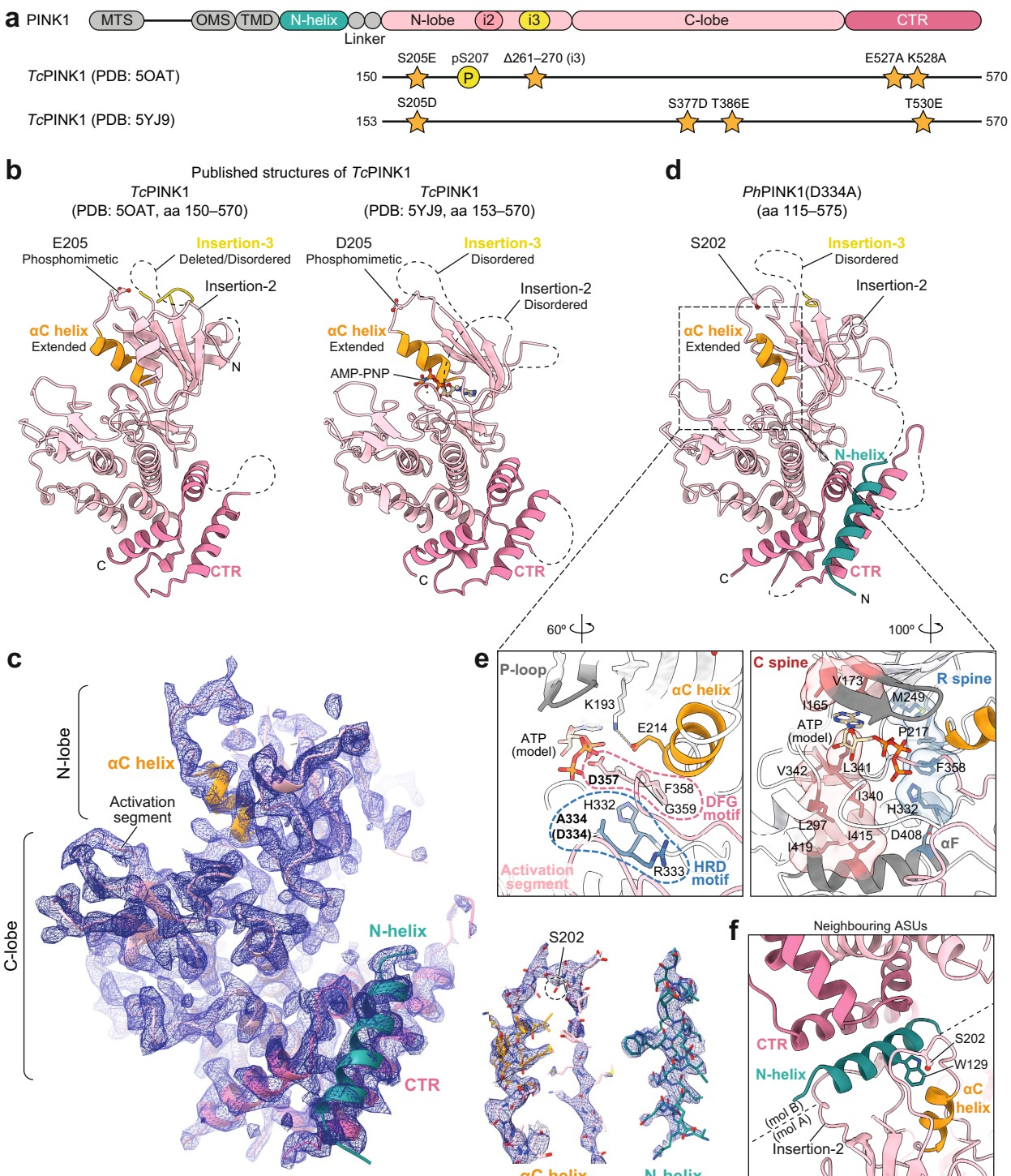

**Extended Data Fig. 1 | Crystal structures of unphosphorylated PINK1.**
**a**, Constructs used in crystal structures of *Tc*PINK1 (PDB: 5OAT[11] and 5YJ9[12]). Phosphomimetics substituted for Ser205 (equivalent to Ser202 in *Ph*PINK1) and other residues, although one structure (PDB: 5OAT) still showed phosphorylation at Ser207 (equivalent to Ser204 in *Ph*PINK1, and Ser230 in *Hs*PINK1, see below). Constructs were further engineered as indicated and described[11,12]. **b**, Depiction of *Tc*PINK1 crystal structures, with key features indicated. **c**, 2|F$_o$|−|F$_c$| electron density contoured at 1.5σ, covering the *Ph*PINK1(D334A) molecule in the asymmetric unit. Right, detail for regions of interest including the αC helix and Ser202-containing loop and the N-helix. **d**, Our crystal structure of unphosphorylated *Ph*PINK1(D334A) (compare Fig. 1) in the same orientation as in **b**, for comparison. **e**, Two different zoomed views

of the ATP-binding site of unphosphorylated *Ph*PINK1, with an ATP molecule modelled in from PDB 2PHK (ref. [42]) as before[10]. Left, highlighting key features common to most active kinases, including the P-loop (grey), αC helix (orange) and activation segment (pink). The DFG and HRD motifs (dotted outlines) which include Asp357 involved in coordination of Mg$^{2+}$, and the substrate-binding Asp334 involved in phosphoryl transfer (and mutated to Ala in the crystal structure) are indicated. Glu214 and Lys193 form a crucial salt bridge to coordinate ATP. Right, assembled catalytic (C) and regulatory (R) spines[24] in red and blue, respectively, indicative of an active kinase conformation. **f**, Crystal contact involving N-helix residue Trp129, which binds to the N-lobe of a symmetry related molecule. ASU, asymmetric unit.

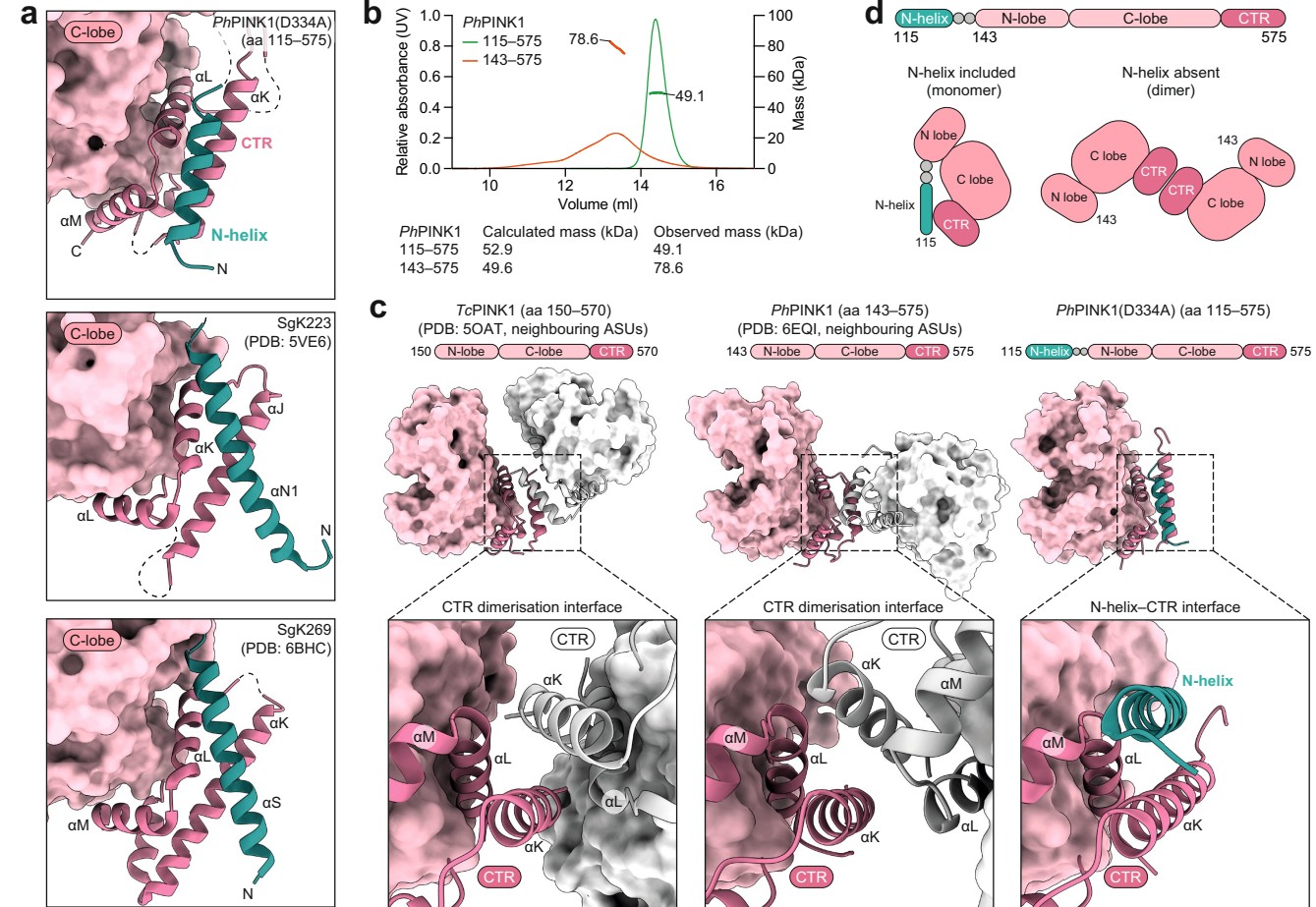

**Extended Data Fig. 2 | The N-helix and its role in keeping PINK1 monomeric.**
**a**, The pseudokinases SgK223 (PDB: 5VE6) and SgK269 (PDB: 6BHC) were structurally characterised only recently[25,26], and contain a CTR similar to PINK1 that is complemented by an N-helix, in a highly analogous fashion now shown for PINK1. While conceptually similar, the organization of the N-helix on the CTR is however distinct. SgK223 and SgK269 utilize a cross-architecture to provide a dimerization interface, whereas the orientation in PINK1 is parallel to the αK helix of the CTR. **b**, SEC–MALS analysis of *Ph*PINK1 with (amino acids 115–575) or without (amino acids 143–575) the N-helix. The shorter *Ph*PINK1 construct tends to form less well-defined dimers, whereas the longer variant is

a monomer. Experiments were performed three times with identical results. **c**, Previous *Tc*PINK1 and *Ph*PINK1 structures dimerized in the crystal lattice through the CTR, reflecting an available interaction surface since the N-helix was missing. *Tc*PINK1 structures dimerized identically (left, only shown for PDB 5OAT[11]). The relative orientation of *Ph*PINK1 molecules in PDB 6EQI[10] was different to *Tc*PINK1 (middle). Right, *Ph*PINK1(D334A) structure including the N-helix, for comparison. **d**, Schematic of the N-helix–CTR interaction, and situation in previous PINK1 structures, as identified in structures and in SEC–MALS.

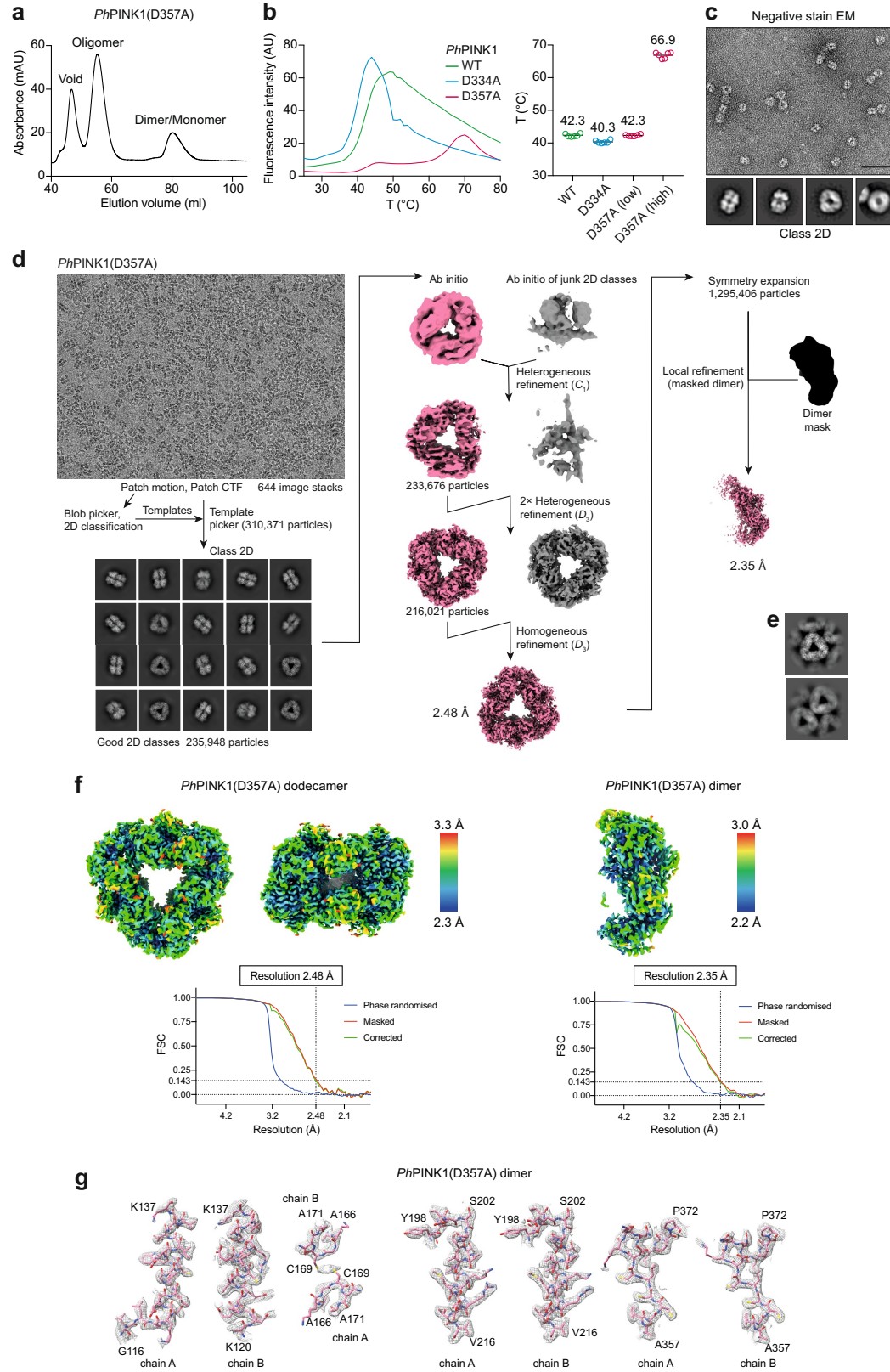

**Extended Data Fig. 3** | See next page for caption.

**Extended Data Fig. 3 | *Ph*PINK1(D357A) oligomerization enables EM studies. a**, Elution profile of the *Ph*PINK1(D357A) mutant during purification by SEC. Data shown is from a representative experiment of three runs. **b**, Thermal denaturation studies of purified *Ph*PINK1 (amino acids 115–575) variants. The crystallized D334A mutant has a melting curve profile and melting temperature similar to WT *Ph*PINK1. *Ph*PINK1(D357A) shows an unusual profile with a high secondary melting temperature. Technical duplicates were measured in three independent experiments. Average melting temperatures are indicated. **c**, Negative stain EM analysis of *Ph*PINK1(D357A) revealed a highly ordered oligomer suitable for cryo-EM studies. Minimal processing was performed in RELION (v.3.1)[51], and a subset of the resulting 2D classes is depicted as insets. Scale bar, 50 nm. Negative staining for this sample was performed once, before advancing to cryo-EM analysis. **d**, Flowchart for cryo-EM analysis as described in Methods. **e**, 2D classification of particles reveal a 2D-crystalline arrangement of *Ph*PINK1(D357A) oligomers in some areas of the grid. **f**, Final cryo-EM density maps (coloured by local resolution) for the *Ph*PINK1(D357A) dodecamer (left) at 2.48 Å and the extracted dimer (right) at 2.35 Å. **g**, Examples of map quality for the final 2.35 Å density covering the *Ph*PINK1(D357A) dimer.

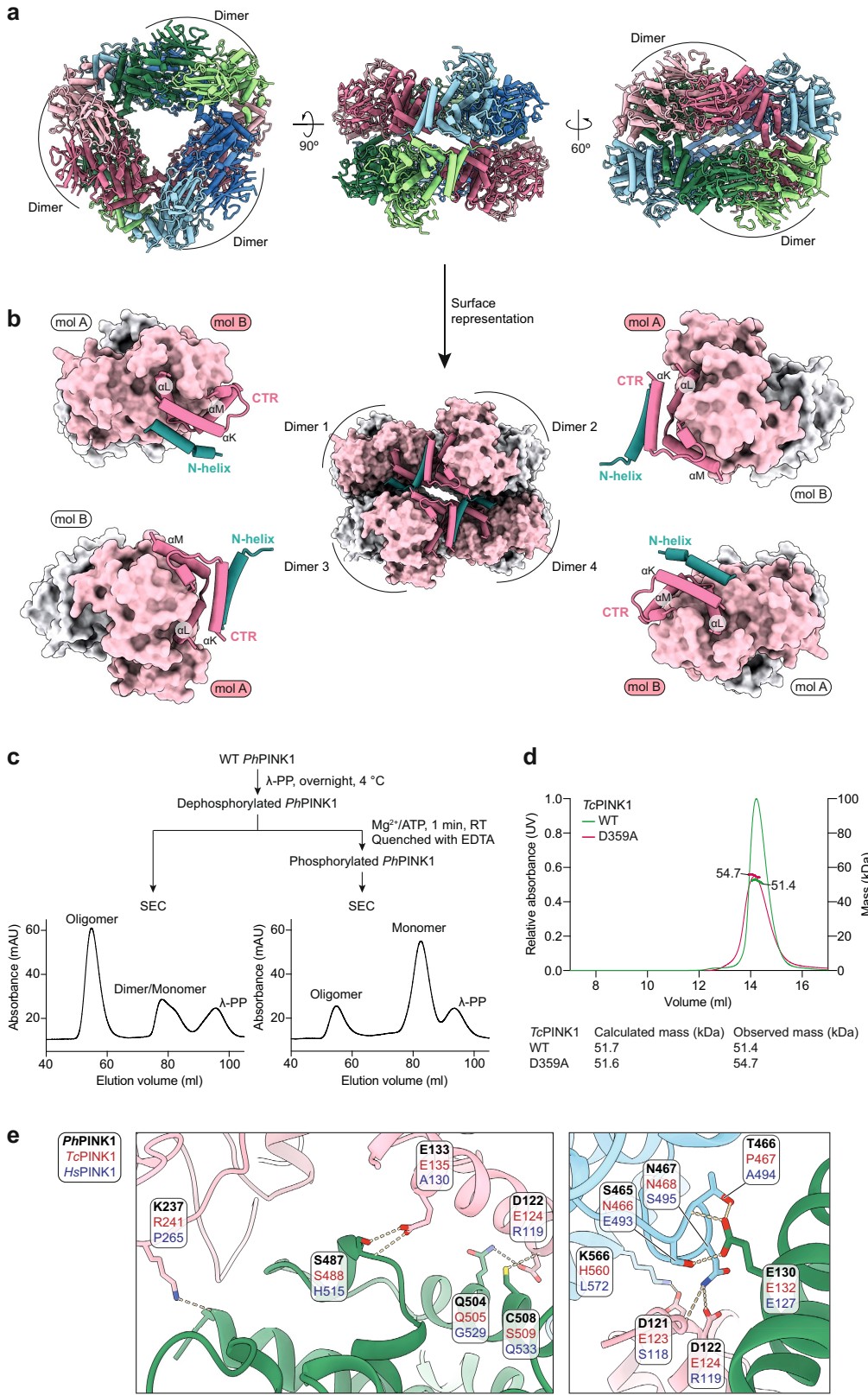

**Extended Data Fig. 4** | See next page for caption.

**Extended Data Fig. 4 | Further analysis of the *Ph*PINK1 oligomeric state.** **a**, Model built into the *Ph*PINK1(D357A) dodecamer cryo-EM density as in Fig. 2b. **b**, Side view of the oligomer showing how the N-helix–CTR area facilitates contacts to connect four dimers. **c**, We were concerned that *Ph*PINK1 oligomerization may only happen with a specific mutant protein and tested whether the oligomer would also form with unphosphorylated WT *Ph*PINK1. WT *Ph*PINK1 is phosphorylated at multiple sites when purified from *E. coli* (see[10]), and was hence dephosphorylated by λ-PP and repurified by SEC as indicated (see Methods). Like the inactive D357A mutant, a prominent oligomer, as well as a dimer/monomer equilibrium is apparent on SEC. Alternatively, the WT protein was first dephosphorylated by λ-PP, then rephosphorylated by adding Mg$^{2+}$/ATP for 1 min, and the kinase and λ-PP were inactivated by EDTA (see Methods). Phosphorylation destabilized the oligomer to a predominant monomeric species, explaining why WT *Ph*PINK1 does not form the oligomer, and suggesting that autophosphorylation resolves the oligomer and dimer (see below). These experiments were part of the purification process and were performed three times with identical results. **d**, SEC–MALS analysis of *Tc*PINK1 variants, to show that corresponding *Tc*PINK1 constructs (amino acids 117–570) do not show oligomeric behaviour. *Tc*PINK1 purifications revealing monomeric behaviour were performed at least twice, and a SEC–MALS experiment was performed once. **e**, Residues mediating hydrogen bonds (dotted lines) in the oligomer interfaces. Lack of conservation of oligomer contacts between *Ph*PINK1, *Tc*PINK1 and *Hs*PINK1 likely explain why *Tc*PINK1 does not form an oligomer, and why we do not expect an identical oligomer in *Hs*PINK1.

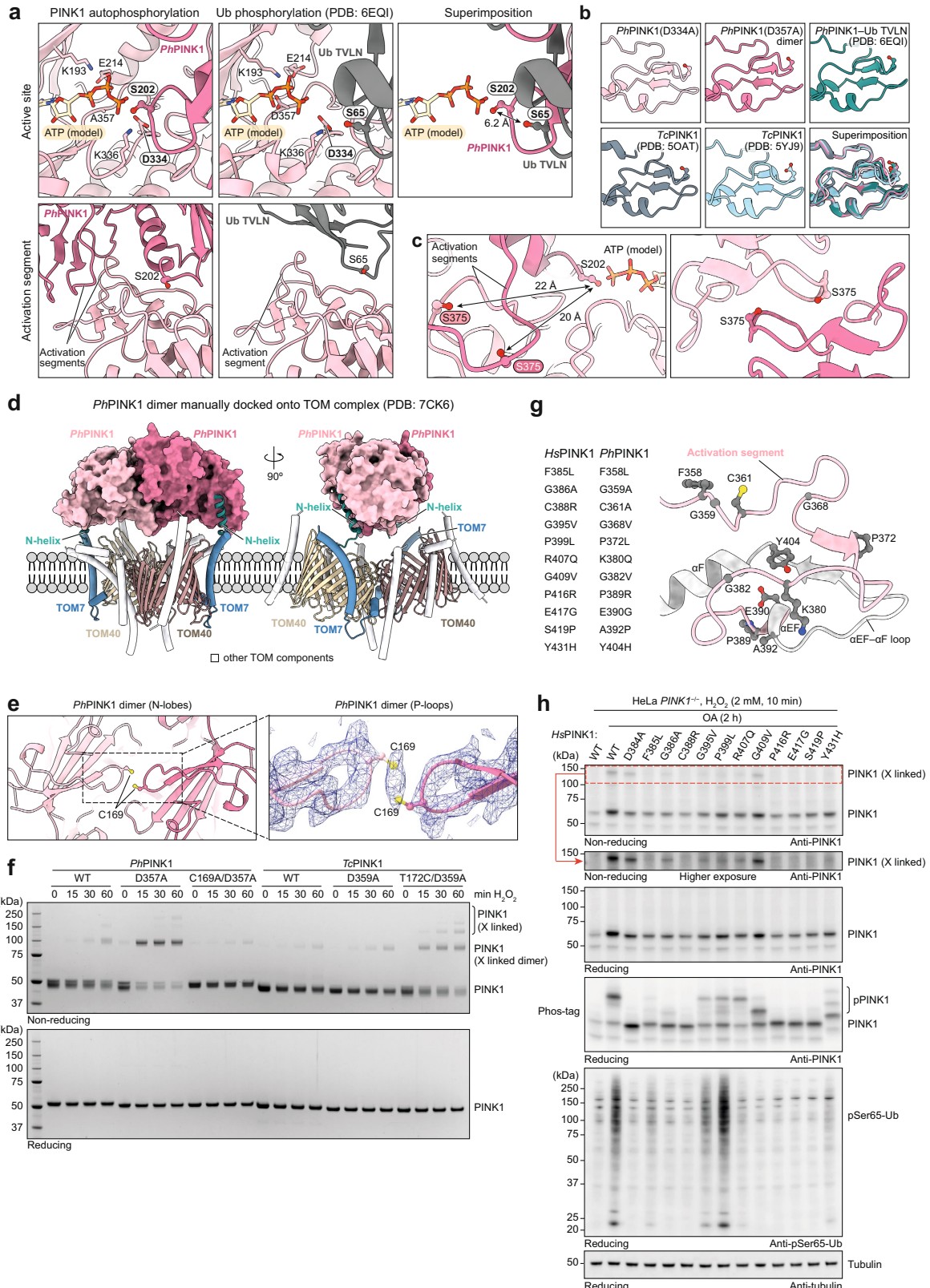

**Extended Data Fig. 5** | See next page for caption.

**Extended Data Fig. 5 | Structural and biochemical analysis of the *Ph*PINK1 dimer. a**, Comparison between *Ph*PINK1 autophosphorylation and ubiquitin phosphorylation resolved previously (PDB: 6EQI)[10], with relevant details in the active site (top row) and in the activation segment (bottom row). The right panel shows substrate disposition in relation to a modelled ATP molecule as in Fig. 3b. The dimeric *Ph*PINK1 autophosphorylation complex appears to place Ser202 in an ideal phospho-accepting position. **b**, Comparison of activation segment structures in all published PINK1 structures, revealing high structural similarity. Ser375 (*Ph*PINK1)/Ser377 (*Tc*PINK1) is shown in ball-and-stick representation; this residue was mutated to Asp in one prior structure (see Extended Data Fig. 1a). **c**, *Ph*PINK1 Ser375 corresponds to Ser402 in *Hs*PINK1, which is a reported phosphorylation site[31,33], but is located within the activation segment and out of reach of the substrate-binding site within dimeric *Ph*PINK1. Our structure does not reveal how autophosphorylation at this residue could be facilitated in *cis* or *trans* (left), nor how phosphorylation through e.g. an upstream kinase would contribute to PINK1 activity or function, since phosphorylation would likely disrupt the dimer (right). ATP was modelled as in Fig. 3b. **d**, Manual docking of dimeric *Ph*PINK1 onto a cryo-EM structure of dimeric human TOM complex (PDB: 7CK6)[61]. The *Ph*PINK1 dimer was oriented with its two N-helices (spanning ~80 Å) aligning with the two TOM7 subunits of the TOM complex dimer. TOM7 has been reported as essential for PINK1 stabilization on the TOM complex[27,28]. Note that some TOM components have considerable cytosolic domains that would need to be accommodated in addition to a PINK1 dimer. **e**, Unidentified density in the 2.35 Å cryo-EM map connects Cys169 in the dimer. **f**, Time course of *Ph*PINK1 and *Tc*PINK1 disulphide formation upon treatment with 2 mM $H_2O_2$, resolved on a non-reducing SDS–PAGE gel. The dimer/oligomer stabilizing *Ph*PINK1 D357A mutation (Fig. 3) enables fast disulphide formation that is averted by an additional C169A mutation. *Tc*PINK1 WT or D359A (equivalent to *Ph*PINK1(D357A)) mutant do not show fast cross-linking behaviour observed in *Ph*PINK1. Engineering of an additional Thr172 to Cys mutation (*Tc*PINK1(T172C/D359A)) results in rapid emergence of cross-linked *Tc*PINK1 dimers upon oxidation. Experiments were performed in biological triplicate with identical results. See Supplementary Fig. 1 for uncropped gels. **g**, EOPD mutations in the activation segment and αEF–αF loop. Mutations according to[10]. **h**, *Hs*PINK1 EOPD mutants listed in **g** were expressed in HeLa *PINK1*^(−/−) cells, stabilized with OA and treated with $H_2O_2$ to assess PINK1 dimerization, autophosphorylation and ubiquitin phosphorylation activity (see Methods). The control D384A mutant (mutation of the DFG motif, equivalent to D357A in *Ph*PINK1) was included as an inactive mutant. As anticipated, it is able to dimerize (oxidative cross-link formed) but unable to autophosphorylate or generate phosphorylated ubiquitin. Oxidative dimerization enables separation of pure catalytic mutants and dimerization deficient mutants. Experiments were performed in biological triplicate with identical results. See Supplementary Fig. 1 for uncropped blots.

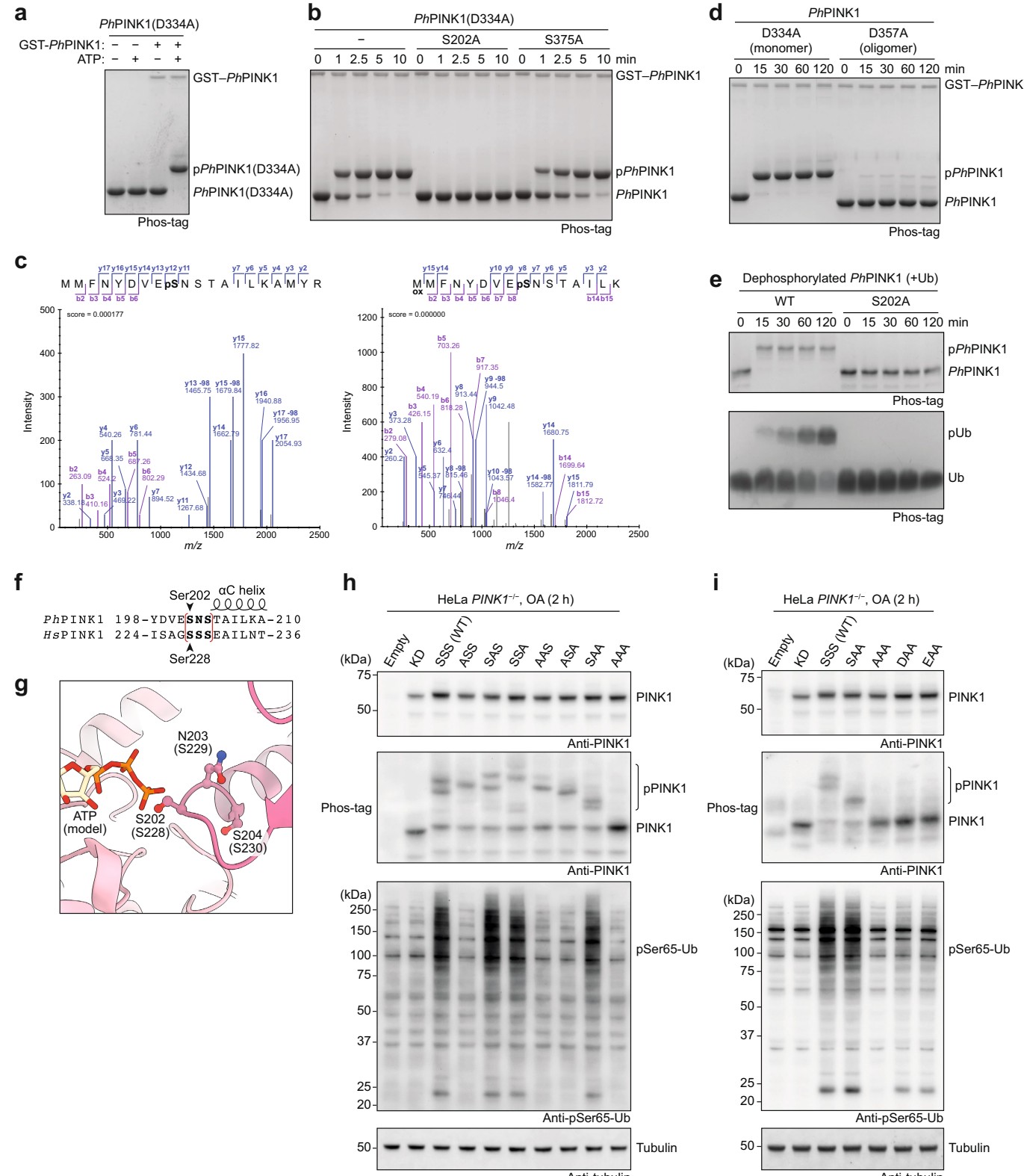

**Extended Data Fig. 6** | See next page for caption.

**Extended Data Fig. 6 | Analysis of autophosphorylation sites in *Ph*PINK1 and *Hs*PINK1. a**, Phos-tag analysis of inactive, monomeric *Ph*PINK1(D334A), which does not autophosphorylate when incubated with ATP. Phosphorylation by WT GST-tagged *Ph*PINK1 for 2 h leads to a shift of the entire protein consistent with a single phosphorylation site, and a small amount (1–2%) shifts to a higher species indicating additional autophosphorylation events after prolonged incubation. A representative gel of three independent experiments is shown. See Supplementary Fig. 1 for uncropped gel. **b**, As in **a** but shown in a time course experiment. Mutating Ser202 to Ala abrogates the observed gel shift, indicating that Ser202 is the sole site of phosphorylation. Mutation of Ser375 (equivalent to *Hs*PINK1 Ser402) to Ala does not impede phosphorylation. A representative gel of three independent experiments is shown. See Supplementary Fig. 1 for uncropped gel. **c**, Mass spectrometry confirms the detection of a phosphate at Ser202 of *Ph*PINK1(D334A) after phosphorylation with WT GST–*Ph*PINK1. See Methods. **d**, Time course experiment as in **b**, with monomeric *Ph*PINK1(D334A) or oligomeric *Ph*PINK1(D357A). While the monomer can be phosphorylated by WT GST–*Ph*PINK1, the oligomer is not efficiently phosphorylated since Ser202 is buried in the stable dimer/oligomer structure. This experiment was performed twice with identical results. See Supplementary Fig. 1 for uncropped gel. **e**, Time course experiment using dephosphorylated WT *Ph*PINK1 (amino acids 119–575, which we found to be oligomerization impaired) and *Ph*PINK1 (amino acids 119–575) S202A, and with ubiquitin in the reaction. See Methods for details. Phosphorylation of *Ph*PINK1 at Ser202 is essential for ubiquitin phosphorylation, as both phosphorylation events are completely abrogated if Ser202 cannot be phosphorylated. Representative gels of three independent experiments are shown. See Supplementary Fig. 1 for uncropped gels. **f–i**, Analysis of autophosphorylation sites in *Hs*PINK1. **f**, In *Hs*PINK1, the Ser202 equivalent residue is Ser228, which is a known, important autophosphorylation site. In *Ph*PINK1, the Ser202 site is followed by Asn203 and Ser204, and Ser204 can be phosphorylated when *Ph*PINK1 is expressed in *E. coli*[10]. Also, the Ser204 equivalent residue in *Tc*PINK1 (Ser207) was phosphorylated in a previous crystal structure (Extended Data Fig. 1a).

In *Hs*PINK1, Ser228 is followed by Ser229 and Ser230, and we investigated whether phosphorylation of these residues occurs and whether it contributes to *Hs*PINK1 activity. (**g**) Structural detail of the Ser202-containing loop in the active site of the *Ph*PINK1 autophosphorylation dimer. Equivalent *Hs*PINK1 residues are in brackets. ATP was modelled as in Fig. 3b. Only Ser202 is in the phospho-acceptor site, but the other residues may occupy the site and be phosphorylated subsequently, if the end of the αC helix slightly unravels; since we expect conformational changes in this region (see below), this seemed feasible. **h**, **i**, *Hs*PINK1 variants with combinations in the Ser228-containing loop as indicated (empty, vector control; KD, kinase dead, a triple mutant K219A, D362A (HRD motif), D384A (DFG motif); SSS (WT) refers to the WT sequence with Ser228, Ser229, Ser230; ASS refers to Ala228, Ser229, Ser230; etc.) were expressed in HeLa *PINK1*[−/−] cells, treated with OA for 2 h, and subjected to western blotting (see Methods). **h**, A Phos-tag gel probed with anti-PINK1 antibody reveals that KD and AAA mutants remain unphosphorylated, while the WT *Hs*PINK1 (SSS) shows multiple phosphorylation states, indicating more than one phosphorylation event in this loop. All three Ser residues appear to be phosphorylatable. These results were correlated with appearance of Ser65-phosphorylated ubiquitin in the same cell lysates, revealed by a ubiquitin Ser65-phosphospecific antibody. WT *Hs*PINK1 leads to a strong signal for phosphorylated ubiquitin, while Ala mutation in Ser228 does not lead to a phosphorylated ubiquitin signal. In contrast, if only Ser228 is present (e.g. SAA mutant) the phosphorylated ubiquitin signal is indistinguishable from that of WT *Hs*PINK1. **i**, As in **h**, but with further substitution of Ser228. Phosphomimetic residues Asp228 or Glu228, followed by Ala229 and Ala230, lead to a small increase in phosphorylated ubiquitin, in the absence of autophosphorylation. The results in **h** and **i** indicate that also in *Hs*PINK1, Ser228 phosphorylation is essential to turn PINK1 into a ubiquitin kinase, and a phosphomimetic is a weak substitute. Other residues in this area may also become phosphorylated but do not trigger ubiquitin phosphorylation. Experiments shown in **h** and **i** were performed in biological triplicate with identical results. See Supplementary Fig. 1 for uncropped blots.

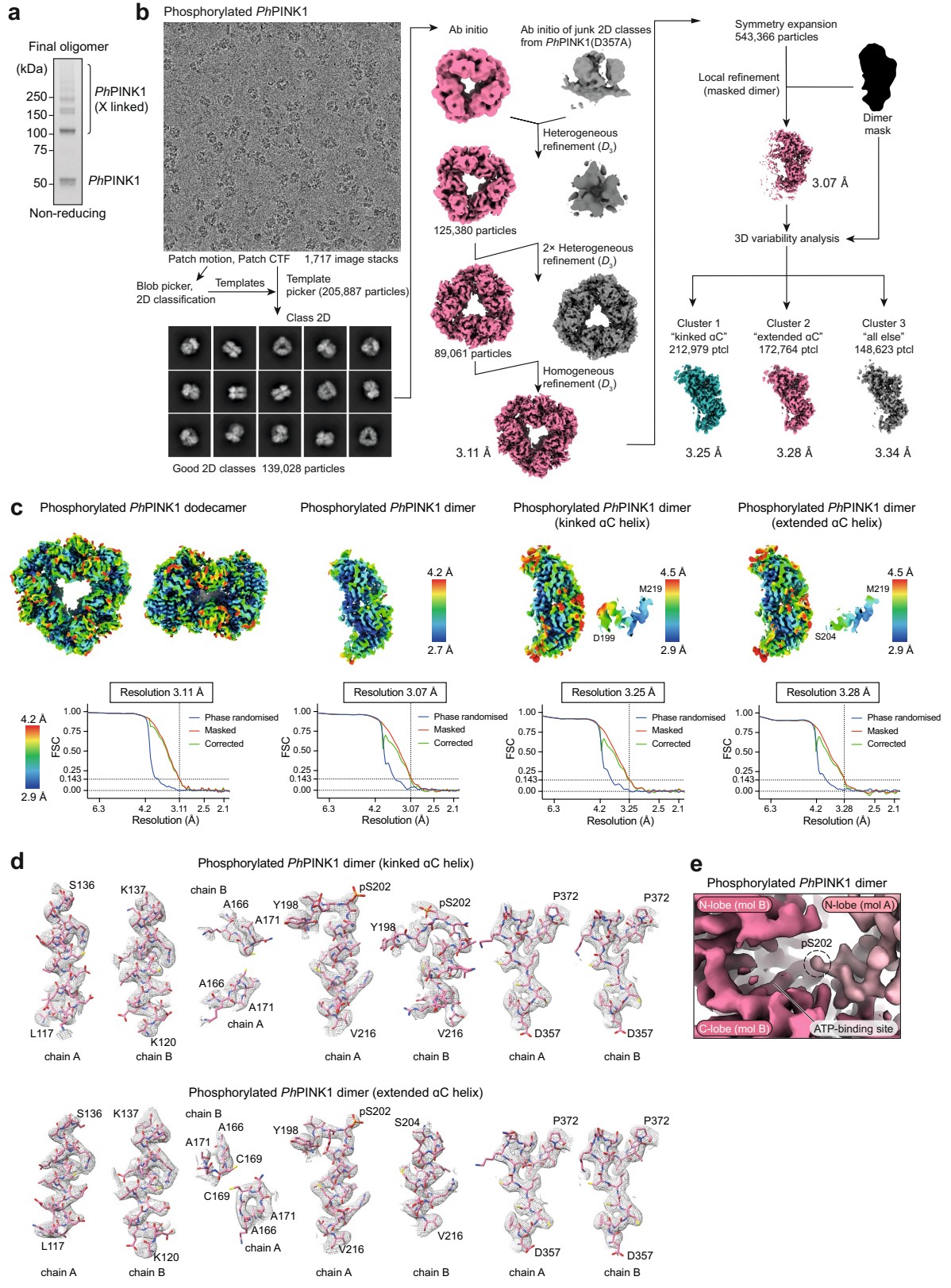

**Extended Data Fig. 7 | Structure of a cross-linked, phosphorylated *Ph*PINK1 reveals asymmetry and conformational changes in the N-lobe. a**, Final phosphorylated and cross-linked *Ph*PINK1 oligomer on a non-reducing SDS–PAGE gel. See Supplementary Fig. 1 for uncropped gel. **b**, Flowchart for cryo-EM analysis, as in Extended Data Fig. 3. **c**, Local resolution maps and resolution calculations. **d**, Map quality for selected regions. **e**, EM density for the phosphorylated Ser202 of molecule A, in the substrate-binding site of molecule B.

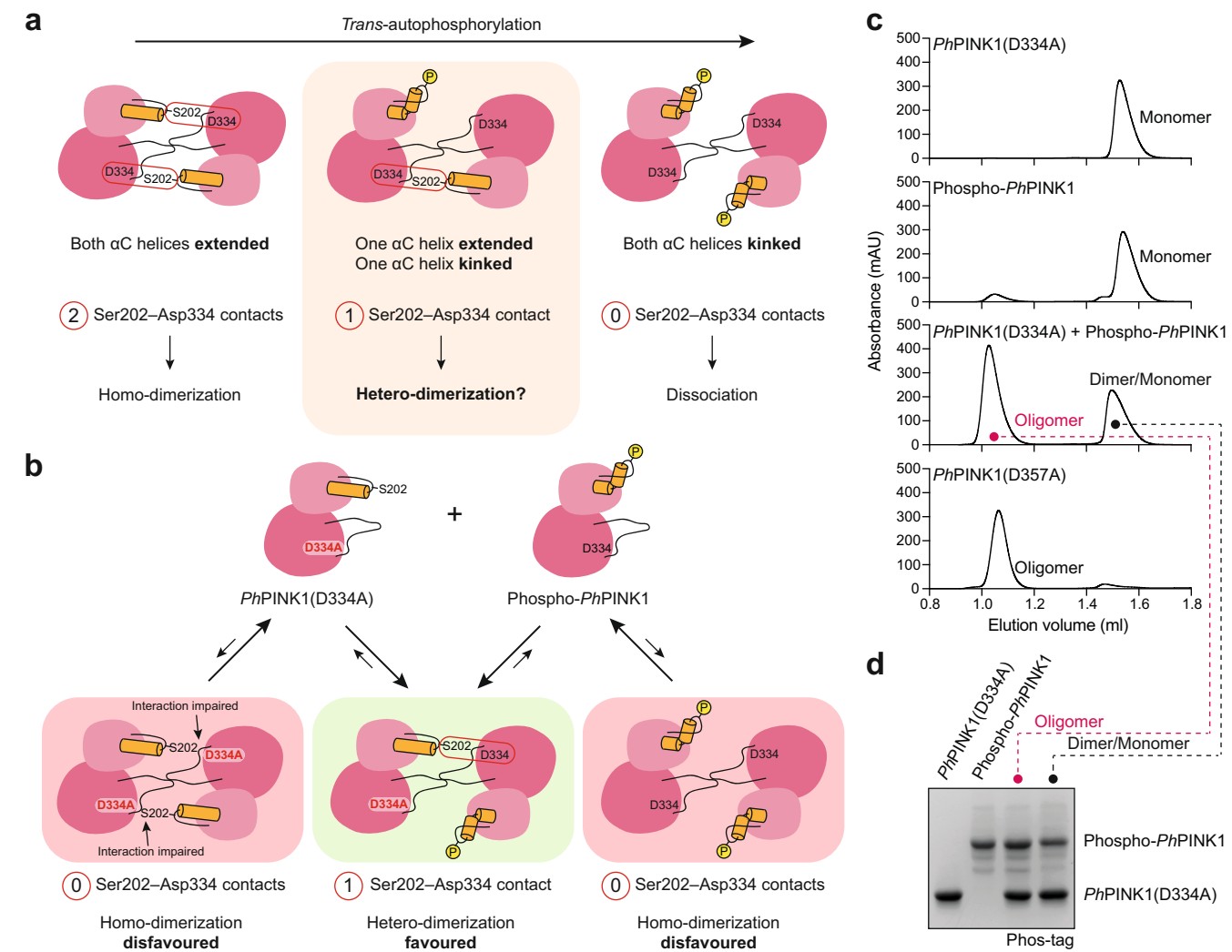

**Extended Data Fig. 8 | Assessment of *Ph*PINK1 dimer stability by measuring oligomer formation. a**, Our studies had revealed that disruption of the kinase–substrate interaction had a detrimental effect for dimer formation. We wondered whether the underlying dimerization mechanism would facilitate the reported effect that each copy of PINK1 is phosphorylated during mitophagy[18,30,33]. Based on our biochemical data (Fig. 2a and Extended Data Fig. 4c) we conceptualized PINK1 dimer interactions as a function of the number of intact Ser202–Asp334 contacts. Ser202 phosphorylation would resolve the Ser202–Asp334 contacts individually, leading to one contact point after the first phosphorylation event, and zero contact points when both PINK1 molecules are phosphorylated. It was unclear whether the dimer is stable after the first phosphorylation event, when only one Ser202–Asp334 contact point exists within the dimer. **b**, We explored whether we would be able to generate *Ph*PINK1 oligomers with an unphosphorylated *Ph*PINK1(D334A) mutant and Ser202-phosphorylated *Ph*PINK1 (phospho-*Ph*PINK1), as indicated, with the knowledge that homo-oligomerization is disfavoured (zero Ser202–Asp334

contact points, highlighted in red). Formation of a hetero-oligomer between unphosphorylated *Ph*PINK1(D334A) and phospho-*Ph*PINK1 would indicate that the dimer is stable with just one Ser202–Asp334 contact (highlighted in green). **c**, SEC analysis of the mutants confirms that unphosphorylated *Ph*PINK1(D334A) or phospho-*Ph*PINK1 do not form oligomers, and hence, zero Ser202–Asp334 contact points are not sufficient. In contrast, enabling one contact point, as per the green scenario in **b** where variants are mixed, enables oligomerization of *Ph*PINK1. Each individual protein showed identical results in every purification run (*n* = 3), and the mixing experiments were performed in biological triplicate. **d**, Phos-tag analysis confirms that the oligomer species is a 1:1 mixture of unphosphorylated *Ph*PINK1(D334A) and phospho-*Ph*PINK1. Experiments were performed in biological triplicate. See Supplementary Fig. 1 for uncropped gel. Together, these experiments show that the dynamic equilibrium favours dimer formation until each copy of PINK1 has been phosphorylated, ensuring full phosphorylation.

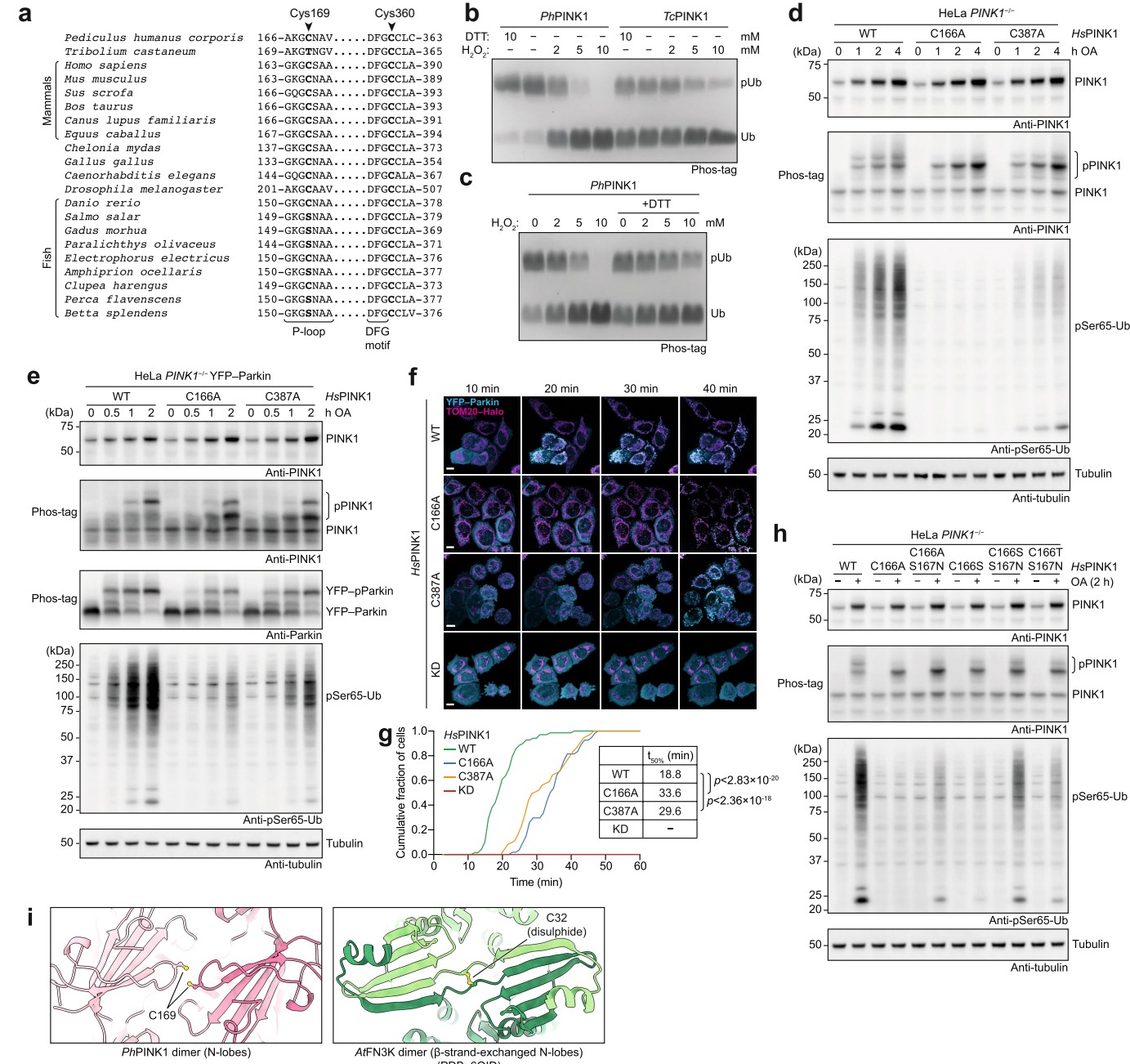

**Extended Data Fig. 9** | See next page for caption.

**Extended Data Fig. 9 | Regulation of PINK1 activity by oxidation.**
**a**, Extended sequence alignment indicating that Cys169 and Cys360 are well conserved in PINK1, and invariant in mammalian PINK1. Cys169 is a Thr in *Tc*PINK1, and a Ser in many fish species. **b**, Comparison of *Ph*PINK1 and *Tc*PINK1, for their ability to be regulated by oxidation, shown in Phos-tag ubiquitin phosphorylation assays (see Fig. 5b). While *Ph*PINK1 activity is abrogated with $H_2O_2$, *Tc*PINK1 with Thr172 in the P-loop, remains active. The observed reduction in *Tc*PINK1 activity could be a result of oxidation of the conserved Cys362 (Cys360 in *Ph*PINK1) in the active site. Experiments were performed in biological triplicate. See Supplementary Fig. 1 for uncropped gel. **c**, Inhibition of *Ph*PINK1 ubiquitin phosphorylation activity can be reversed with DTT, suggesting reversible regulatory oxidation. See Methods. Experiments were performed in biological triplicate. See Supplementary Fig. 1 for uncropped gel. **d**, Time course assessment of *Hs*PINK1 Cys–Ala mutants transiently expressed in HeLa *PINK1*$^{-/-}$ cells. OA treatment leads to accumulation of *Hs*PINK1 and slightly altered autophosphorylation for both C166A and C387A. The *Hs*PINK1(C166A) mutant was almost completely deficient in ubiquitin phosphorylation, while *Hs*PINK1(C387A) showed highly reduced but still detectable phosphorylated ubiquitin levels. Experiments were performed in biological triplicate. See Supplementary Fig. 1 for uncropped blots. **e**, Time course of *Hs*PINK1 mutants as in **d**, but using stable *Hs*PINK1 expression in the presence of YFP–Parkin. Presence of YFP–Parkin seemingly increases levels of phosphorylated ubiquitin for all *Hs*PINK1 variants as compared to **d**, yet overall, phosphorylated ubiquitin and phosphorylated Parkin levels remain strongly diminished with *Hs*PINK1(C166A), and to a lesser degree with *Hs*PINK1(C387A). Experiments were performed in biological triplicate. See Supplementary Fig. 1 for uncropped blots. **f**, Translocation of YFP–Parkin (cyan) to mitochondria (magenta, TOM20–Halo) in HeLa *PINK1*$^{-/-}$ stably expressing *Hs*PINK1 Cys–Ala variants upon OA treatment, imaged using lattice light sheet microscopy. Maximum intensity projections are shown for four different timepoints. YFP–Parkin translocation is delayed in cells expressing either *Hs*PINK1(C166A) or *Hs*PINK1(C387A) relative to WT h*Hs*PINK1. A kinase dead (KD) *Hs*PINK1 variant was included as a control. Images are representative of three independent experiments. Scale bar is 10 μm. See Supplementary Video 1. **g**, Quantification of YFP–Parkin translocation in **f**. The cumulative fraction of cells exhibiting YFP–Parkin translocation is shown over time. Approximately 70 cells were counted per cell line. Each curve was fitted to determine the time for 50% of the cells to feature translocation. Significant differences between curves were determined using a two-sample Kolmogorov–Smirnov test. A MATLAB script and Source Data are available as Supplementary Material. Exact *p* values are: WT–C166A: $p < 2.83 \times 10^{-20}$; WT–C387A: $p < 2.36 \times 10^{-18}$. **h**, *Hs*PINK1 mutants expressed in HeLa *PINK1*$^{-/-}$ cells and treated with OA for 2 h. Activity of *Hs*PINK1(C166A) and *Hs*PINK1(C166S) can be partially restored by an additional S167N mutation, mimicking the sequence observed in many fish species. A *Tc*PINK1-like sequence introduced into *Hs*PINK1, C166T/S167N, is less active than the *Hs*PINK1(C166S/S167N) mutant. The additional Asn in the P-loop of the *Hs*PINK1(C166S/S167N) mutant recovers activity of *Hs*PINK1(C166S), suggesting that both residues are also important in ubiquitin/Ubl substrate interactions, and not merely involved in dimerization. Experiments were performed in biological triplicate. See Supplementary Fig. 1 for uncropped blots. **i**, A redox active switch in fructosamine-3-kinases (PDB: 6OID) is conceptually similar, utilizing a Cys at an identical position (Cys32) for Cys-mediated cross-linking and regulation of kinase activity by oxidation[36], however the overall orientation of kinase domains is dissimilar.

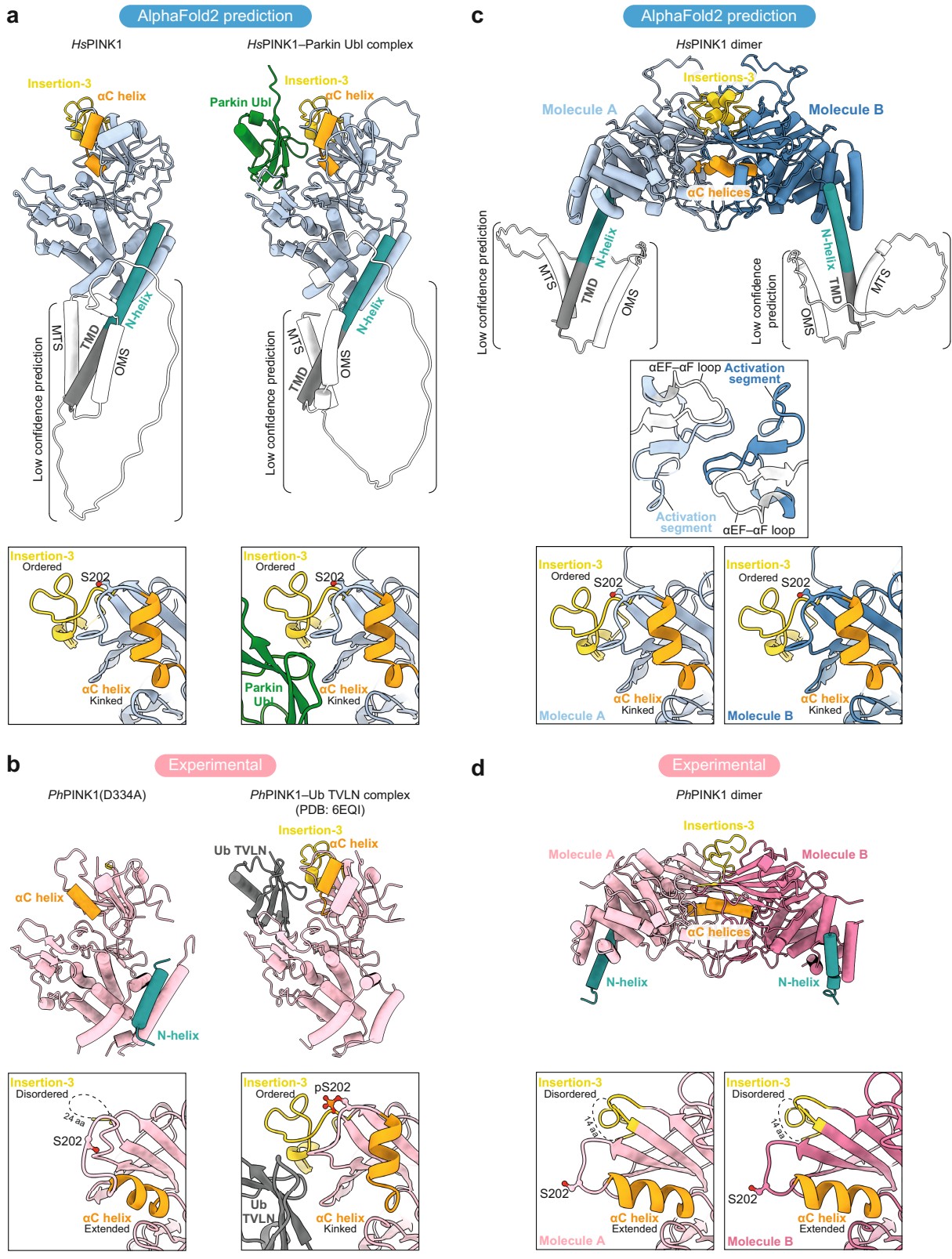

**a** AlphaFold2 prediction

*Hs*PINK1

Insertion-3
αC helix
MTS
TMD
OMS
N-helix
Low confidence prediction

Insertion-3
Ordered
S202
αC helix
Kinked

*Hs*PINK1–Parkin Ubl complex

Parkin Ubl
Insertion-3
αC helix
MTS
OMS
TMD
N-helix
Low confidence prediction

Insertion-3
Ordered
S202
Parkin Ubl
αC helix
Kinked

**b** Experimental

*Ph*PINK1(D334A)

αC helix
N-helix

Insertion-3
Disordered
24 aa
S202
αC helix
Extended

*Ph*PINK1–Ub TVLN complex
(PDB: 6EQI)

Insertion-3
αC helix
Ub TVLN
N-helix

Insertion-3
Ordered
pS202
Ub TVLN
αC helix
Kinked

**c** AlphaFold2 prediction

*Hs*PINK1 dimer

Insertions-3
Molecule A
Molecule B
αC helices
MTS
N-helix
TMD
OMS
Low confidence prediction
N-helix
TMD
OMS
MTS
Low confidence prediction

αEF–αF loop
Activation segment
Activation segment
αEF–αF loop

Insertion-3
Ordered
S202
αC helix
Kinked
Molecule A

Insertion-3
Ordered
S202
αC helix
Kinked
Molecule B

**d** Experimental

*Ph*PINK1 dimer

Insertions-3
Molecule A
Molecule B
αC helices
N-helix
N-helix

Insertion-3
Disordered
14 aa
S202
αC helix
Extended
Molecule A

Insertion-3
Disordered
14 aa
S202
αC helix
Extended
Molecule B

**Extended Data Fig. 10** | See next page for caption.

**Extended Data Fig. 10 | Analysis of the *Hs*PINK1 model as predicted by AlphaFold2[37,38].** (**a**, **b**) Predicting the structure of *Hs*PINK1 via AlphaFold2 (**a**, left), resulted in a model remarkably similar to *Ph*PINK1 from the *Ph*PINK1–Ub TVLN complex[10] (**b**, right), and features a kinked αC helix, ordered insertion-3, and an extended N-helix that binds and extends from the CTR directly into the membrane. Consistently, predicting a complex between *Hs*PINK1 with the ubiquitin-like (Ubl) domain of human Parkin (**a**, right), places the Ubl domain at *Hs*PINK1 analogously to the *Ph*PINK1–Ub TVLN complex (**b**, right). Both predictions are dissimilar to unphosphorylated *Ph*PINK1 (**b**, left) and *Tc*PINK1 (Extended Data Fig. 1). Insets show the detail of the N-lobe with a kinked αC helix and an ordered insertion-3. The prediction is somewhat surprising, as it suggests *Hs*PINK1 to be an active ubiquitin kinase even without Ser228 phosphorylation, which contradicts biochemical analysis. Since AlphaFold2 does not yet predict the impact of post-translational modifications, we interpret the prediction such that it is possible and even likely, that *Hs*PINK1 can adopt a ubiquitin-phosphorylation-competent conformation consistent with our previous *Ph*PINK1–Ub TVLN complex structure[10]. **c**, **d**, We next used AlphaFold2 to predict a dimer of *Hs*PINK1. **c**, Strikingly, AlphaFold2 predicts a symmetric dimer with a dimer interface identical to the one shown for *Ph*PINK1 (compare with **d** and Fig. 3). In fact, we have already validated this arrangement of *Hs*PINK1 molecules via our Cys166 cross-linking experiments in Fig. 3g. However, in the predicted dimer of *Hs*PINK1, the αC helix is kinked and Ser228 does not contact the second molecule. This is different from our conclusions in Extended Data Fig. 8, but again may be a result of not incorporating the effect of Ser228 phosphorylation. We therefore anticipate that unphosphorylated *Hs*PINK1 can also adopt a conformation with an extended αC helix that places Ser228 into the active site of the dimeric molecule to facilitate autophosphorylation prior to forming the depicted conformation. AlphaFold2 predictions hence support the notion that the activation model proposed in Fig. 5g applies to *Hs*PINK1.

# nature research

# Reporting Summary

Nature Research wishes to improve the reproducibility of the work that we publish. This form provides structure for consistency and transparency in reporting. For further information on Nature Research policies, see our Editorial Policies and the Editorial Policy Checklist.

## Statistics

For all statistical analyses, confirm that the following items are present in the figure legend, table legend, main text, or Methods section.

| n/a | Confirmed | |
|---|---|---|
| ☐ | ☒ | The exact sample size (*n*) for each experimental group/condition, given as a discrete number and unit of measurement |
| ☐ | ☒ | A statement on whether measurements were taken from distinct samples or whether the same sample was measured repeatedly |
| ☐ | ☒ | The statistical test(s) used AND whether they are one- or two-sided *Only common tests should be described solely by name; describe more complex techniques in the Methods section.* |
| ☒ | ☐ | A description of all covariates tested |
| ☒ | ☐ | A description of any assumptions or corrections, such as tests of normality and adjustment for multiple comparisons |
| ☒ | ☐ | A full description of the statistical parameters including central tendency (e.g. means) or other basic estimates (e.g. regression coefficient) AND variation (e.g. standard deviation) or associated estimates of uncertainty (e.g. confidence intervals) |
| ☐ | ☒ | For null hypothesis testing, the test statistic (e.g. *F*, *t*, *r*) with confidence intervals, effect sizes, degrees of freedom and *P* value noted *Give P values as exact values whenever suitable.* |
| ☒ | ☐ | For Bayesian analysis, information on the choice of priors and Markov chain Monte Carlo settings |
| ☒ | ☐ | For hierarchical and complex designs, identification of the appropriate level for tests and full reporting of outcomes |
| ☒ | ☐ | Estimates of effect sizes (e.g. Cohen's *d*, Pearson's *r*), indicating how they were calculated |

*Our web collection on statistics for biologists contains articles on many of the points above.*

## Software and code

Policy information about availability of computer code

| Data collection | ÄKTA Pure (Cytiva), DAWN light scattering detector (Wyatt Technology), Optilab T-rEX Refractive Index Detector (Wyatt Technology), Rotor-Gene Q (Qiagen), timsTOFII Pro Mass Spectrometer (Bruker), ChemiDoc (Bio-Rad), Talos L120C TEM (Thermo Scientific), Talos Arctica TEM (Thermo Scientific); Titan Krios G4 TEM (Thermo Scientific); Lattice Light Sheet (LLS) 7 (ZEISS - pre-serial) |
|---|---|
| Data analysis | Crystallography: XDSme (0.6.5.2), CCP4i (7.0.001), POINTLESS (1.10.18), AIMLESS (0.5.21), Phenix (1.19.2-4158-000), Phaser (2.8.3), MolProbity (4.5.1), Coot (0.9), UCSF ChimeraX (1.1.1) Cryo-EM: EPU software (2.9), Relion (3.1), cryoSPARC (3.2.0), UCSF Chimera (1.14), UCSF ChimeraX (1.1.1), Coot (0.9), EMAN (2.3.1) Biochemistry: GraphPad Prism (9.0.0), Image Lab (Bio-Rad, 6.1), ASTRA software (Wyatt technology, 7.3.1.9), Rotor-Gene Q Series Software (2.3.1), MaxQuant (1.6.17.0), Skyline Daily (21.1.1.198), UNICORN 7 (Cytiva, 7.5 and 7.6) Imaging: ZEN (ZEISS, 3.5), Fiji (ImageJ, 1.53k), Matlab (R2019b) |

For manuscripts utilizing custom algorithms or software that are central to the research but not yet described in published literature, software must be made available to editors and reviewers. We strongly encourage code deposition in a community repository (e.g. GitHub). See the Nature Research guidelines for submitting code & software for further information.

## Data

Policy information about availability of data

All manuscripts must include a data availability statement. This statement should provide the following information, where applicable:
- Accession codes, unique identifiers, or web links for publicly available datasets
- A list of figures that have associated raw data
- A description of any restrictions on data availability

Coordinates and crystallographic structure factors for PhPINK1 D334A have been deposited with the protein data bank under accession codes 7T3X, and EM models and maps under accession codes 7T4M (PhPINK1 D357A dodecamer, EMD-25680), 7T4N (PhPINK1 D357A dimer, EMD-25681), 7T4L (with extended αC helix in chain B, EMD-25679), and 7T4K (wild-type phosphorylated PhPINK1 dimer with kinked αC helix in chain B, EMD-25678). We also deposited the map for wild-type phosphorylated PhPINK1 dimer prior to 3D variability analysis (EMD-25677). Uncropped versions of all gels and blots are provided in Supplementary Figure 1. Source data for Western blot quantification (Fig. 5f), melting curve measurements (Extended Data Fig. 3b) and quantifying imaging results (Extended Data Fig. 9g) are provided, the latter with corresponding Matlab script.

# Field-specific reporting

Please select the one below that is the best fit for your research. If you are not sure, read the appropriate sections before making your selection.

☒ Life sciences        ☐ Behavioural & social sciences        ☐ Ecological, evolutionary & environmental sciences

For a reference copy of the document with all sections, see nature.com/documents/nr-reporting-summary-flat.pdf

# Life sciences study design

All studies must disclose on these points even when the disclosure is negative.

| | |
|---|---|
| Sample size | No sample size calculation was performed. The sample size (n) depicted in biochemical assays reflects the number of independently measured experiments. In cell biological experiments sample size (n) indicates the number of independent replicates performed. The number of technical and independent biological experiment is stated in each figure legend. Each experiment was successfully replicated at least three times. |
| Data exclusions | No data were excluded from analyses |
| Replication | Biochemical data including gel-based assays were reproduced with technical replicates and in independent experiments, with the number n of independent experiments stated in each figure legend. For gel-based assays, experiments were performed independently on different days. Uncropped gels are shown in Supplementary Information. A minimum of three repeats was performed for each experiment. All attempts at replication were successful. |
| Randomization | not applicable, as only a small number of samples were handled in each experiment. |
| Blinding | not applicable, as only a small number of samples were handled in each experiment. |

# Reporting for specific materials, systems and methods

We require information from authors about some types of materials, experimental systems and methods used in many studies. Here, indicate whether each material, system or method listed is relevant to your study. If you are not sure if a list item applies to your research, read the appropriate section before selecting a response.

## Materials & experimental systems

| n/a | Involved in the study |
|---|---|
| ☐ | ☒ Antibodies |
| ☐ | ☒ Eukaryotic cell lines |
| ☒ | ☐ Palaeontology and archaeology |
| ☒ | ☐ Animals and other organisms |
| ☒ | ☐ Human research participants |
| ☒ | ☐ Clinical data |
| ☒ | ☐ Dual use research of concern |

## Methods

| n/a | Involved in the study |
|---|---|
| ☒ | ☐ ChIP-seq |
| ☒ | ☐ Flow cytometry |
| ☒ | ☐ MRI-based neuroimaging |

## Antibodies

| | |
|---|---|
| Antibodies used | rabbit anti-PINK1 D8G3 (Cell Signaling Technology, #6946, lot 5); mouse anti-Parkin Prk8 (Cell Signaling Technology, #4211, lot 7), rabbit anti-phospho-ubiquitin (Ser65) (Millipore #ABS1513-I, lot 3117322); rabbit anti-TOM20 FL-145 (Santa Cruz Biotechnology |

sc-11415, lot D1613); goat anti-rabbit (SouthernBiotech, #4010-05, lot A4311-TF99D); goat anti-mouse (SouthernBiotech, #1030-05, lot E2518-Z929D); anti-tubulin hFAB rhodamine (Bio-Rad, #12004165, batch 64385864).

| Validation | All primary antibodies have been validated for Western blotting in human cell lysates, by the manufacturer. Each primary antibody has featured in >50 publications as listed on the manufacturers websites.

PINK1 (https://www.cellsignal.com/products/primary-antibodies/pink1-d8g3-rabbit-mab/6946)
Parkin (https://www.cellsignal.com/products/primary-antibodies/parkin-prk8-mouse-mab/4211)
Phospho-ubiquitin (https://www.merckmillipore.com/AU/en/product/Anti-phospho-Ubiquitin-Ser65,MM_NF-ABS1513-I)
TOM20 (https://www.scbt.com/p/tom20-antibody-fl-145)
Tubulin (https://www.bio-rad.com/en-au/sku/12004165-hfab-rhodamine-anti-tubulin-primary-antibody-200-ul?ID=12004165) |

## Eukaryotic cell lines

Policy information about cell lines

| Cell line source(s) | HeLa human PINK1 -/- cells (gift from Michael Lazarou, Monash University) |
| Authentication | Cell lines displayed expected cell morphologies and have been validated by the Garvan Molecular Genetics facility. |
| Mycoplasma contamination | Cells were tested negative for Mycoplasma. Cells were screened routinely for mycoplasma contamination using the MycoAlert Mycoplasma Detection Kit (Lonza) as per manufacturer's instructions. |
| Commonly misidentified lines (See ICLAC register) | No misidentified cell lines have been used |

