## [Peer Review File · Nature]

Manuscript Title: Activation mechanism of PINK1

Reviewer Comments & Author Rebuttals

Reviewer Reports on the Initial Version:

Ref #1

This manuscript describes the structure of phPINK1 in new states. They add an N-terminal helical structure of PINK1 that was not previously known (except on BioRxiv for tcPINK1), they determine the structure of nonphosphorylated dimers of phPINK1, they identify that hydrogen peroxide induces disulfide dimers of ph and human PINK1, and they explore the conformational change that occurs upon phosphorylation. The disulfide dimer issue is only seen under oxidizing conditions and shown not to occur with hPINK1 under an in-cell activation condition. The cryoEM structure of a crosslinked oligomer containing a mixture of inactive and active homodimers is most interesting. This manuscript makes valuable contributions to the understanding of PINK1 activation. The work appears well done and the text is very clearly written.

Specific Points

- 1) The scale of the TOM complex in Fig. 1b is very far off that of PINK1. Although this may be OK in Fig. 1b, in Fig. 3b it is more of a problem as it depicts the positioning of PINK1 on opposite sides of the TOM dimer – whereas if to scale the PINK1 dimer may well be on proximal sides of the TOM dimer. It may be worth scaling them more appropriately in Fig. 3a to offer clearer insight into this interesting aspect.
- 2) In Fig. 4(d,e), to corroborate that PINK1 forms a disulfide dimer it would be good to add a reducing gel of the samples treated under the same conditions, ideally with the same samples used for nonreducing gels.
- 3) Fig. 4 – does the C166A mutation affect hPINK1 mitophagy activity?
- 4) Extended Fig. 9 is quite an elegant experiment. However, it would be important to add to panel d one more lane showing the monomeric gel filtration peak from the mixed wt and D357A PINK1 using a tagged form of wt phPINK1. If the results show that only fully phosphorylated tagged PINK1 appears in the monomer, it supports the model in panel b. If the monomer contains a mixture of phosphorylated and non-phosphorylated wtPINK1, the model needs to be revised or the text toned down to reflect the weakening of the data in support of the model.
- 5) On line 258 starts out with “highly accurate” followed later with “this is questionable”. I suggest deleting the words highly accurate.
- 6) The final model in Fig. 5f is a fairly obvious way to explain the mutual phosphorylation of the two components of the dimer – it is hard to think of another model other than the subunits of a dimer precisely simultaneously phosphorylate one another. And the later - rather unlikely - possibility is not ruled out as they studied a crosslinked oligomer that may behave differently than WT PINK1 dimers bound to the TOM complex.
- 7) The cystine dimer aspect of the manuscript from Fig. 4 is not incorporated into the model in Fig. 5f (other than cryptically in the two black loops). I feel Fig. 4 detracts from the manuscript as it is 1) not shown to occur without peroxide treatment or to occur during PINK1 induced mitophagy and 2) it is not shown that the disulfide is reduced (or oxidized) in Extended Fig. 9 or in context of the final dimer dissociation model in Fig. 5f.

Ref #2

Here the authors have extended their structural analysis of the mechanisms through which the PINK1 mitochondrial kinase, which is mutationally inactivated in one form of early onset Parkinson's Disease (PD), is activated in response to mitochondrial damage. They have previously reported crystal structures of the insect *Pediculus humanus* (Ph) catalytic domain in a phosphorylated state bound to ubiquitin in its C-terminally retracted conformation, showing how Ub contacts the PINK1 N-terminal lobe via a unique insertion and presents Ub Ser65 to the active site promoting its phosphorylation, and also how autophosphorylation in the N lobe stabilizes the

unique PINK1 catalytic domain insertions. Here, they describe a crystal structure of an unphosphorylated kinase-dead D334A PhPINK1 catalytic domain, which reveals a 38 aa N-terminal helix extending from the N-lobe, not observed in the structure of the phosphorylated PINK1 catalytic domain, that interfaces with the C-terminal CTR helix, and holds the PhPINK1 monomer in the inactive state. The functional importance of this new interface is underscored by the presence of a number of early onset PD mutation sites. This new structure also suggests that the N-lobe of PINK1 has to undergo significant conformational changes during PINK1 activation. Structural analysis of a D357A kinase-dead mutant PhPINK1 catalytic domain led to the unexpected discovery that the PhPINK1 catalytic domain can dimerize and oligomerize, and cryo-EM analysis of D357A PhPINK1 revealed a dodecameric structure, comprised of a two-ring stack each containing three dimers. This structure is disrupted when PhPINK1 is allowed to autophosphorylate, consistent with phosphorylation preventing oligomerization. Next, the authors used the D357A PhPINK1 dimer structure to elucidate the structural basis for autophosphorylation, showing that the OH-group of S202, which is autophosphorylated, in one monomer lies within 2.2 Å of the gamma phosphate of ATP in the other monomer. Interestingly, in the inactive dimer structure, the N-helix:CTR region is oriented so that both N-termini face the mitochondrial outer membrane (MOM). To establish that S202 phosphorylation occurs in trans, they showed that WT PhPINK1 catalytic domain can phosphorylate D334A PhPINK1 in vitro on S202. Importantly, they showed that a S202A mutant PhPINK1 can neither autophosphorylate nor phosphorylate Ub, establishing the importance of S202 phosphorylation in PhPINK1 activation. They also showed that the phosphorylation of both S202 residues within a dimer is required for dissociation into monomers. Through their structural analysis, they also uncovered an additional mode of PINK1 regulation resulting from the formation of a disulfide bond under oxidative conditions between Cys169 in one monomer with Cys169 in the second monomer. The Ub kinase activity of a C169A mutant PhPINK1 was less sensitive than WT PhPINK1 to H₂O₂ treatment, consistent with Cys169 being an oxidation target. Ala mutation of Cys360 in the ATP-binding pocket, which lies 15 Å away from Cys169, by itself did not render PhPINK1 less sensitive to H₂O₂ but a C169/360A double mutant was significantly more resistant to H₂O₂ treatment, consistent with Cys169 and Cys360 oxidation being functional oxidation sensors, but for different reasons. Consistently, the D357A mutant PhPINK1 catalytic domain, which is predominantly dimeric, showed a much greater propensity to form disulfide-bonded dimers than the WT PhPINK1, and dimers of human PINK1 could also be observed when hPINK1 was treated with H₂O₂. In a final step, they partially crosslinked dimers and oligomers by H₂O₂ treatment of unphosphorylated PhPINK1 catalytic domain, and then allowed them to undergo autophosphorylation, before submitting them to cryo-EM structural analysis. By analysis of different molecular classes, they obtained two distinct types of structure - a symmetric PhPINK1 dimer trapped during the process of trans-autophosphorylation, with both molecules phosphorylated at S202. However, the second PhPINK1 molecule in each dimer adopted two different conformations, with one structure in a partially unfolded state suggesting that this molecule of phospho-PhPINK1 had been trapped in the process of undergoing a conformational change to become an active Ub kinase. Finally, they used the new AlphaFold and ColabFold structure prediction algorithms to predict structures of hPINK1 monomers and dimers. The monomer structure predicted was very similar to that previously reported for the hPINK1-Parkin-Ubl complex, and the predicted hPINK1 dimer had a dimer interface almost identical to that reported here for PhPINK1, suggesting that the lessons learned from the PhPINK1 structural analysis described here are highly relevant to the mechanisms of activation and regulation of hPINK1.

This is a very convincing and well written story, which provides new insights into the mechanism of activation of the PINK1 kinase in repose to mitochondrial damage. The cryo-EM structure of the PINK1 dimer trapped in the act of transphosphorylation of Ser222 in one monomer by the other, is particularly exciting, and the dimer structure with both termini pointing towards the membrane implies that the organization of PINK1 on the surface of mitochondria, and in particular in the vicinity of the TOM transport pore will be important in its activation in response to damage and its ability to phosphorylate Ub at S65. In addition, the new evidence that PINK1 can be inhibited by oxidation of two Cys residues, suggests that PINK1 may be negatively regulated in the cell by oxidative stress, which given the key role of the mitochondrion in generating ROS could be important.

Points: 1. Figure 2A/page 4: It is hard to see colors of the lines in the 66K peak – the three peaks

could be labeled more clearly. The D334A Ph PINK1 mutant cannot autophosphorylate either, but does not form dimers or oligomers. Is this because the S202-D334 interaction is missing?

2. Figure 2B/page 4: The dodecameric PhPINK1 structure is intriguing, even if it is unique to PhPINK1. The authors state that TcPINK1 was not found to oligomerize, but it is not clear if the contact residues that stabilize the dodecamer are present in TcPINK1, or, in one sense more importantly, in hPINK1. The overall dimensions of this dodecameric structure are not given, and if this structure could form in the cell how would these dimensions relate to those of the dimeric TOM complexes? With regard to the hexameric symmetry of this PhPINK1 complex, it is interesting to note that the PGAM5 phosphatase, another MOM resident protein, was recently shown to be able to adopt a hexameric structure in vitro (PMID: 30705304).

3. Page 4: Did the authors test if an equivalent mutant form of hPINK1 can form oligomers? Can higher order oligomers be revealed by crosslinking studies, which might stabilize unstable oligomers?

4. Page 5 top: This section could be written more clearly to indicate that S202 in one monomer is interacting with substrate binding site of the second monomer to facilitate trans-phosphorylation by the other monomer, and also to indicate that S202 is in a loop in the N-lobe

5. Figure 3B: Perhaps, the S202 and D334 residues/labels could be highlighted in some way so that it is easier for the reader to appreciate their interaction.

6. Figure E6/page 6: Can more than one of these three Ser in hPINK1 be phosphorylated simultaneously, or is their phosphorylation mutually exclusive? There is some discussion of this in the legend, but it would be interesting to know whether either S229 or S230 could be accommodated in the active site of the other monomer once S228 is phosphorylated, and whether having multiple phosphorylated Ser might lead to "stronger" activation.

8. Pages 7 and 9: As the authors indicate, the fact that the C169A and C360A PhPINK1 mutants are still at least partially sensitive to H₂O₂ (in Figure 4C, PhPINK1 seems to be as sensitive as WT PhPINK1 to H₂O₂) suggests oxidation of either Cys to sulfenic acid or a higher order oxidation state is inhibitory, but particularly for Cys169 one would like to know what dictates the formation of a disulfide bond versus sulfenic acid. Can PINK1 dimers be detected in cells treated with H₂O₂?

7. Page 7, line 213: From Figure E8 it appears that these partially crosslinked WT PhPINK1 oligomers were a result of H₂O₂ oxidation. If this is the case, it should be indicated in the text, and also why this oligomer is not simply a dimer, which what the data in Figure 4 suggests it should be.

8. Page 7: Since ROS generated by the ETC and exiting from the mitochondrion will be present at very high concentrations locally, one might expect these Cys to be constitutively oxidized, unless GSH levels are also very high in the vicinity of the MOM.

10. Figure 3A/Figure 5F/page 8: With regard to the authors' modeling of the PINK1 activation process, it is not clear whether the TOM dimer complex in the lower panel of Figure 3A is drawn to scale. There are Cryo-EM structures of the human TOM complex (125 Å across) that could be included and possibly docked onto the PhPINK1 dimer (or even dodecamer) to see whether this model is realistic.

Minor point: Page 7, line 204: The authors need to indicate that OA (okadaic acid) is a PP2A inhibitor, and spell out why it was used.

Ref #3

This manuscript uses X-ray crystallography and cryo-EM to study regulation of head lice PINK1 kinase. It identifies a) the location of the N-terminal helix, b) an unexpected dimer in the non-phosphorylated state that explains why Ser202 (Ser228 in human) is the main regulatory phosphorylation c) a redox switch mediated by C169 and d) the structural consequences of Ser202 auto-phosphorylation, nicely validated by AlphaFold2 prediction. The combination of the dimer

structure and the N-terminal helix location suggests how PINK1 oriented relative to TOM complex on the membrane. The relevance of the dimerization, the phosphorylation at Ser228 and the occurrence of the disulfide bound dimer are validated in human PINK1 in HeLa PINK1 -/- cells by expression of PINK1 variants.

Much of this work is enabled by an unexpected dodecamer found in the D357A mutant of PhPINK1, which enabled the high resolution cryo-EM analysis.

It is an elegant study and the experiments are well presented, with gorgeous figures. The main finding in my opinion is the identification of the PINK1 dimer, as this made it possible to bring mechanistic clarity to the activation by Ser202 phosphorylation and the PINK1 location relative to the mitochondrial membranem. It also suggested the presence of a potential redox switch. Such a redox switch at that location on mitochondria is indeed evocative.

Comments

It remains to be seen what the functional consequences of the redox switch are, as validation was limited to showing that it can occur in an overexpression system. It is clear that the authors attempted to limit overexpression artefacts, but it would be good to add a control to show how their expression compares to WT expression.

The occurrence of the dodecamer is of course fortuitous as it allowed high resolution cryo-EM analysis. The success of that analysis shows this is a very well-behaved dodecamer, and then one really wonders if it could have a functional role after all. The manuscript rather easily dismisses that option, it would at least be necessary to discuss the reasons why and possibly the processing of the PINK1 protein in more detail

The validation of the dimer in the human cells is primarily through its disulfide formation, although not strictly necessary, some other form of validating the existence of the dimer would be nice. With respect to the H₂O₂ experiment, what is the band that is induced at 100 kD by H₂O₂? Why is TcPINK1 affected by H₂O₂ if it lacks ser 202

For the last structure it would be good if local resolution in the region that changes between upon phosphorylation could be made explicit

There is a nice table for refinement of the X-ray structure but nothing for the cryo-EM structures, please add an equivalent table 1.

The attempt to keep to length restrictions seems to have really hurt precision and clarity in many places; the main issues that were most distracting are

1) Details of the specific experiment: all cryo-EM is presumably performed with D357A, but this is only noted in a few places and omitted elsewhere;

2) Some kinase specific terms could use better introduction in a general journal 'the HRD' , C and R spines, 'a second kinase inactivating mutant' the phosphorylation sites, acceptor site

3) The Parkinson introduction has some errors probably due to brevity: it suggests that all EOPD have known mutations, also suggests that only EOPD has no disease-modifying treatment. In fact as far as I know it is only one of the EOPD mutations for which there is hope of a disease modifying treatment (GBA mutations), whereas idiopathic Parkinson is not different from EOPD (except sometimes faster in progression)

4) It would be helpful to indicate more precisely which structure is what, e.g. by giving pdb files in the schematic model of figure 5F, explaining what PDB is meant in line 232 where 'the PhPINK1:Ub complex' suddenly pops up, uniprot filenumbers for the alphafold predictions,

5) Precision in general,

a. Line 62: divergent Ser/Thr protein kinase: divergent relative to what?

b. Line 64: that contain only two insertions : relative to what? I guess this should be 2 of these four, but then you immediately wonder in which lobe;

c. line 66: adopt typical folds typical of what? typical kinase ? or specific to these?

d. Line 102: A second kinase inactivating mutation second relative to what?

e. Line 130 Modelling ATP places the γ -phosphate: its gamma-phosphate

f. Line 131: resembling the PhPINK1:Ub: grammar wrong, not clear what resembling refers to

g. Line 188: this inhibition could partially be restored with C169A mutation (Fig. 4c). I guess they

mean prevented....

h. Line 213: a partially crosslinked wild-type PhPINK1 oligomer: a few more words to explain would be useful

i. Line 238 there are two intact contacts, which are subsequently resolved by phosphorylation at Ser202 and aC helix kinking : it is not clear what is meant here

j. Line 248: phosphorylated ser 202 leads to : I'm not sure this is inference or proven

k. Line 261: extra words, remove in the : "even in the without hPINK1 phosphorylation."

Author Rebuttals to Initial Comments:

We would like to thank the Reviewers for their expert views and highly constructive criticism. In the following, we detail how we addressed each of their points, in most cases with additional experimentation. In conclusion, we believe we have substantiated the manuscript further.

We have slightly reordered the manuscript and now discuss the process of PINK1 activation first, before we discuss regulation of PINK1 by oxidation in a new **Fig 5**. With new experiments as suggested by the reviewers, we have now reaffirmed the oxidative regulation of PINK1, including of human PINK1 on mitochondria. For example, we now show that Cys166 has a strong impact on phosphoubiquitin generation and mitophagy as suggested by Reviewer 1, and we show that regulation of PINK1 activity is reversible for *Ph*PINK1 in vitro, and also for hPINK1 expressed in cells (new **Fig 5**). In the latter, we introduce and exploit another new insight: Cys166 is conserved in most vertebrates, with the exception of fish species; substitution of the P-loop of hPINK1 with the analogous sequence from fish, circumvents reversible regulation of PINK1 activity by H₂O₂. We are now highly confident that PINK1 has a reversible oxidative switch that requires further investigations and appropriate animal models in future work.

We have further extended our biochemical and structural work by providing a more sophisticated discussion of distances in the complexes and a better modelling of the TOM complex. We also addressed multiple points regarding clarity of figures and text, which are mentioned below.

We believe that our revised manuscript addresses all concerns raised by the Reviewers, and we are grateful for their suggestions as they significantly improved the manuscript in both content and style.

Reviewer #1 (comments to authors)

This manuscript describes the structure of phPINK1 in new states. They add an N-terminal helical structure of PINK1 that was not previously known (except on BioRxiv for tcPINK1), they determine the structure of nonphosphorylated dimers of phPINK1, they identify that hydrogen peroxide induces disulfide dimers of ph and human PINK1, and they explore the conformational change that occurs upon phosphorylation. The disulfide dimer issue is only seen under oxidizing conditions and shown not to occur with hPINK1 under an in-cell activation condition. The cryoEM structure of a crosslinked oligomer containing a mixture of inactive and active homodimers is most interesting. This manuscript makes valuable contributions to the understanding of PINK1 activation. The work appears well done and the text is very clearly written.

Specific Points

1) The scale of the TOM complex in Fig. 1b is very far off that of PINK1. Although this may be OK in Fig. 1b, in Fig. 3b it is more of a problem as it depicts the positioning of PINK1 on opposite sides of the TOM dimer – whereas if to scale the PINK1 dimer may well be on proximal sides of the TOM dimer. It may be worth scaling them more appropriately in Fig. 3a to offer clearer insight into this interesting aspect.

We agree with the Reviewer that this is a very interesting aspect and finding of our manuscript. Indeed, while merely illustrative in Fig. 1, we aimed to keep the scale of the TOM complex in Fig. 3 as appropriate as possible based on the current data. Several TOM complex structures from different species, determined by cryo-EM, are now available, and show distinctive oligomeric states of the TOM complex. A prevalent model is a TOM complex dimer, in which two TOM40 pores are juxtaposed and surrounded by TOM complex subunits TOM22, TOM5, TOM6 and TOM7.

Our PINK1 dimer structures offer a clear restraint as to how far the N-termini are separated from each other as they extend into the membrane-spanning region that presumably anchors PINK1 in the mitochondrial membrane. This distance is approximately 80 Å, which would be consistent with anchoring PINK1 either proximal or opposite sides of the TOM complex dimer, depending on how the PINK1 transmembrane domain is angled relative to the adjacent N-helix (although AlphaFold2 does predict the TMD and N-helix to form a single continuous helix).

We provide a more sophisticated model of this complex in **Extended Data Fig. 5d**, where we orient the N-helices to line up with TOM7, a TOM complex subunit important for PINK1 stabilisation on the TOM complex during mitophagy (Hasson et al, Nature 2013, Sekine et al, Mol Cell 2019). We note that this modelling is highly similar to the model suggested by Trempe and colleagues for TcPINK1, available on BioRxiv. However, we feel that these insights require further careful structural validation, which is a task for subsequent studies. Considering this, we felt that a complex depiction in the Figure 3 would be too speculative, and so we would refrain from including it in the main Figures (we are happy to take advice to the contrary from Reviewers and Editor on this matter).

2) In Fig. 4(d,e), to corroborate that PINK1 forms a disulfide dimer it would be good to add a reducing gel of the samples treated under the same conditions, ideally with the same samples used for nonreducing gels.

We have now added reducing gel electrophoresis of matched samples to confirm reduction of the disulphide-linked dimer. These experiments are shown in **Fig. 3g** and **Extended Data Fig. 5f,h**.

3) Fig. 4 – does the C166A mutation affect hPINK1 mitophagy activity?

This was an important question, which we have addressed with multiple lines of experimentation, now discussed in a new Fig 5 / Extended Data Fig 9.

Firstly, we now show that a human (h)PINK1 C166A mutant, when expressed in PINK1^{-/-} HeLa cells and induced by OA treatment to a similar degree as wild-type hPINK1, failed to catalyse significant ubiquitin phosphorylation compared with wild-type PINK1 (new **Extended Data Fig 9d**). Further, and consistent with our observations with recombinant PhPINK1, mutation of the juxtaposed cysteine in hPINK1 (C387A) mutant was likewise less active, but was not completely deficient (new **Extended Data Fig 9d**).

To assess the ability of these PINK1 variants to recruit Parkin, we co-expressed PINK1 and YFP-Parkin in HeLa PINK1^{-/-} cells that lack endogenous Parkin. Here, despite the cells now having a feed-forward amplification of phospho-ubiquitin due to Parkin expression, the Cys mutants were still largely defective as a ubiquitin kinase. Correspondingly, in cells expressing PINK1 C166A, Parkin translocation to mitochondria was delayed when assessed using real-time microscopy (new **Extended Data Fig 9f,g**).

Therefore, our data suggest for the first time that Cys166 on human PINK1, and to a lesser degree, Cys387, are important for PINK1 function in mitophagy. In addition, our data now go significantly beyond this, showing that C166 reversibly dictates hPINK1 activity in response to oxidation (H₂O₂ treatment). Since oxidative regulation was reversible with DTT, we can exclude that oxidation involves formation of Cys sulphinic or Cys sulphonic acid. Rather, PINK1 is more likely to form a stable Cys sulphenic acid intermediate (an S-OH reversible modification). We have discussed this oxidation control in the manuscript text.

4) Extended Fig. 9 is quite an elegant experiment. However, it would be important to add to panel d one more lane showing the monomeric gel filtration peak from the mixed wt and D357A PINK1 using a tagged form of wt phPINK1. If the results show that only fully phosphorylated tagged PINK1 appears in the monomer, it supports the model in panel b. If the monomer contains a mixture of phosphorylated and non-phosphorylated wtPINK1, the model needs to be revised or the text toned down to reflect the weakening of the data in support of the model.

We thank the reviewer for their comments and agree that these were interesting and important experiments. This experiment (now **Extended Data Fig. 8**) assesses whether dimerisation of PINK1 can occur between a phosphorylated PINK1 and unphosphorylated PINK1. We have already shown in **Extended Data Fig. 4c** that dephosphorylated PhPINK1 forms dimers/oligomers, and that this oligomer disassembles PhPINK1 phosphorylation by addition of ATP. One wonders whether this PINK1 oligomer would remain intact if phosphorylation only occurred on one molecule of the dimer and the other molecule remained unphosphorylated. If so, this would mean that the PINK1 dimer would remain intact until both molecules of the dimer are phosphorylated, ensuring complete phosphorylation of both molecules in the PINK1 dimer.

In vitro phosphorylation of PhPINK1 D334A by wild-type GST-PhPINK1 (autophosphorylated) already suggests that phosphorylated PINK1 can dimerise and phosphorylate unphosphorylated PINK1. Furthermore, the fact that the activation segment, which provides a major dimerization interface, remains identical in all structures of PINK1 to date, suggests that the activation segment may facilitate PINK1 dimerisation independent of its phosphorylation status.

Dimerisation occurs via three major points of interaction: the two symmetric interaction points between the Ser202-containing loop and the active site of the other molecule, and the activation segment. As activation segment conformation remains unchanged upon phosphorylation of Ser202, it likely does not contribute to dimer dissociation.

The interaction Ser202-containing loop of one molecule and the active site of the other involves an important Ser202-Asp334 hydrogen bond. Mutation of either Ser202 or Asp334 to Ala disrupts dimerization. Therefore, we have called this interaction the Ser202-Asp334 contact. Due to dimer symmetry, two Ser202-Asp334 contacts exist. Phosphorylation of Ser202 in one molecule will disrupt one Ser202-Asp334 contact. Phosphorylation of Ser202 of the second molecule will disrupt the second Ser202-Asp334 contact.

The question is whether just one Ser202-Asp334 contact in the dimer (i.e. when only one molecule is phosphorylated) is sufficient to keep the dimer intact.

We tested this idea as outlined in **Extended Data Fig. 8b**. Mixing phosphorylated *PhPINK1* with unphosphorylated *PhPINK1* and checking for oligomerisation (as a readout of dimerisation) by SEC would almost certainly lead to dimerisation between unphosphorylated *PhPINK1* alone, since we would expect the interaction between two unphosphorylated *PhPINK1* molecules to be stronger than between unphosphorylated and phosphorylated *PhPINK1* (two Ser202-Asp334 contacts as opposed to one Ser202-Asp334 contact).

To prevent dimerisation of unphosphorylated *PhPINK1* alone, while still allowing dimerisation between phosphorylated and unphosphorylated *PhPINK1*, we used *PhPINK1* D334A as unphosphorylated *PhPINK1*. D334A mutation prevents dimerisation between *PhPINK1* D334A molecules (D334A disrupts Ser202-Asp334 contacts), while still allowing dimerization between phosphorylated *PhPINK1* and *PhPINK1* D334A (one Ser202-Asp334 contact is intact, see diagram in **Extended Data Fig. 8b**).

Hence the oligomer that is formed when phosphorylated *PhPINK1* and *PhPINK1* D334A are mixed in **Extended Data Fig. 8c** must be a result of dimerisation between phosphorylated *PhPINK1* and *PhPINK1* D334A, and this was confirmed by running the oligomer on a Phos-tag gel in **Extended Data Fig. 8d**, confirming dimerisation can occur between phosphorylated *PhPINK1* and unphosphorylated *PhPINK1*.

The fact that a monomer peak is still present when both species are mixed simply suggests that the equilibrium does not go to completion towards an oligomeric state. This is expected since only one Ser202-Asp334 contact exists, so interaction between the dimer is not as strong as having both contacts present. Losing both Ser202-Asp334 contacts, leads to even lower levels of oligomerisation (see **Extended Data Fig. 8c**, *upper two SEC curves*). Hence, the composition of the monomer peak was expected to contain both species of *PhPINK1* as we now show as per the Reviewer's request. This is expected and does not affect our conclusions of the experiment.

(The reviewer suggests that we should use tagged *PhPINK1*, likely to separate the two mixed species on the gel. However, we use a Phos-tag gel which separates *PhPINK1* based on phosphorylation status through mobility shift. The two *PhPINK1* species of distinct phosphorylation status can be separated using a Phos-tag gel, hence a tag to generate a size

difference is not necessary. Furthermore, addition of a tag would likely disrupt oligomerisation.)

We did find that phosphorylated PINK1 still dimerises to a small extent, and indeed the new **Fig 5e** (wild-type PINK1, active from OA treated mitochondria) still shows a discernible cross-linked dimer band. This band could originate from stabilised yet unphosphorylated hPINK1 (~80% of PINK1 is phosphorylated as per Phos-tag analysis after OA treatment, see e.g. **Extended Data Fig 5h**, WT + OA), but could also refer to phosphorylated PINK1 dimers in equilibrium with a monomeric species.

We have now adjusted our Model in **Fig. 5** to indicate this equilibrium, as per the reviewers suggestion. We note that the dimeric form of PINK1 must resolve into a monomer, to allow access of ubiquitin to the same substrate binding site. We believe that access to a substrate, ubiquitin or Parkin, will further shift the equilibrium of PINK1 towards the monomeric ubiquitin kinase state.

5) On line 258 starts out with “highly accurate” followed later with “this is questionable”. I suggest deleting the words highly accurate.

We agree and have adjusted the text.

6) The final model in Fig. 5f is a fairly obvious way to explain the mutual phosphorylation of the two components of the dimer – it is hard to think of another model other than the subunits of a dimer precisely simultaneously phosphorylate one another. And the later - rather unlikely - possibility is not ruled out as they studied a crosslinked oligomer that may behave differently than WT PINK1 dimers bound to the TOM complex.

We agree. We did consider a few options to depict the idea that the process will unlikely stop unless both molecules are phosphorylated, but felt that the sequential path is likely sufficient. We have updated the model to take into account our new data in the proposed activation mechanism.

7) The cystine dimer aspect of the manuscript from Fig. 4 is not incorporated into the model in Fig. 5f (other than cryptically in the two black loops). I feel Fig. 4 detracts from the manuscript as it is 1) not shown to occur without peroxide treatment or to occur during PINK1 induced mitophagy and 2) it is not shown that the disulfide is reduced (or oxidized) in Extended Fig. 9 or in context of the final dimer dissociation model in Fig. 5f.

In the submitted manuscript we specifically omitted the cystine dimer from our model as the focus of the Cys cross-linkable dimer in Fig 4 was to confirm dimerisation mechanism in PhPINK1, human PINK1 in cells, and in an engineered form of TcPINK1. In light of suggestions from reviewers, we have now explored oxidative regulation further to show that i) hPINK1 when expressed in cells stimulated to undergo mitophagy is likewise regulated by oxidation and ii) oxidative control of both PhPINK1 and hPINK1 is reversible. We agree with the reviewer that these findings warrant separation and so we have now

made the focus of Fig 4 the validation of dimer formation and expanded on oxidative regulation in a revised Fig 5.

To address the role of oxidation in PINK1, we showed that mutation of PhPINK1 Cys169 or Cys360 does two things: both mutations render PINK1 less active (reduced phosphorylation of ubiquitin in the absence of any oxidation) and that both mutations, either individually or when combined, no longer respond to the inhibitory effect of hydrogen peroxide (now Fig 5b). As a result, we proposed that PINK1 may be regulated by oxidation. We now expand on these studies in the revised Fig 5, to show:

1. Human PINK1 C166A (and C387A) are likewise impaired at catalysing ubiquitin phosphorylation when expressed in cells (PINK1^{-/-} HeLa),
2. Consequently, hPINK1 C166A (and C387A) were impaired in their ability to stimulate downstream mitophagy signalling through Parkin recruitment to mitochondria. This is the first description of these residues playing a regulatory role in PINK1.

It is also true that we are lacking evidence that PINK1 is oxidised lacking exogenous H₂O₂ (commonly used on cells to mimic oxidative stress). Hence, following the Reviewer's suggestion, we screened multiple mitophagy inducing stimuli and tested for cross-linkable dimers in the absence of exogenous H₂O₂. We found that GTPP alone (a mitochondrial HSP90 inhibitor, that leads to partial depolarisation) or BCNU (a glutathione reductase inhibitor) together with OA led to detectable cross-linked PINK1 dimers (**Response Fig. 1**). However, we also found these compounds to be toxic to cells and so are reluctant to conclude anything from these experiments at this stage, but include them here for the reviewer's reference.

Response Fig. 1. Alternative stimuli to observe PINK1 dimer formation. Note that cells have died as indicated by loss of tubulin signal.

We now show that hPINK1 is also subject to control by reversible oxidation.

As discussed in the revised text, *Tc*PINK1 does not contain a Cys at position 169, but instead contains a Thr. Similarly, numerous fish species have a Ser at the equivalent position. In human PINK1, a Cys166 to Ala mutation or a Cys166 to Ser mutation are severely impaired in phosphorylating ubiquitin (new **Extended Data Fig 9h**). Importantly, however, activity can be restored to almost wild-type levels when the neighbouring Ser167 is concomitantly mutated to Asn (thereby mimicking the Ser-Asn of fish PINK1). This is likely due to the Cys166 directly contacting ubiquitin after the conformational change in the N-lobe has formed the ubiquitin binding site. A Ser-Ser mutant likely removes this important contact point and as a result, binds ubiquitin less efficiently. Reinstating Asn in the loop at 167 likely reconstitutes the substrate contact, thereby rescuing PINK1 ubiquitin kinase activity.

Irrespective of the molecular mechanism, by exploiting this active human PINK1 SN variant, which lacks Cys at position 166, we could now explore the role of Cys166 in the oxidative control of human PINK1 expressed in cells. We find that, like *PhPINK1*, hPINK1 can be regulated by oxidation via Cys166 and that this regulation is fully reversible (New **Figure 5d-f**).

Together, in revised Fig 3, we first show and exploit Cystine dimer formation, induced by H₂O₂, to validate the dimer model in *PhPINK1* (indeed enabling the second cryo-EM structure), and extend these findings to now show that the same is true for hPINK1 in cells. Then in revised Fig 5, we revisit these Cys variants to show that this site is part of a new system to enable reversible oxidative regulation of PINK1. Whilst we agree that the physiological relevance of this oxidative regulation of PINK1 requires further study, we feel it is important to alert the field of this new interesting mechanism for controlling PINK1 activity.

Referee #2 (Remarks to the Author):

Here the authors have extended their structural analysis of the mechanisms through which the PINK1 mitochondrial kinase, which is mutationally inactivated in one form of early onset Parkinson's Disease (PD), is activated in response to mitochondrial damage. They have previously reported crystal structures of the insect *Pediculus humanus* (Ph) catalytic domain in a phosphorylated state bound to ubiquitin in its C-terminally retracted conformation, showing how Ub contacts the PINK1 N-terminal lobe via a unique insertion and presents Ub Ser65 to the active site promoting its phosphorylation, and also how autophosphorylation in the N lobe stabilizes the unique PINK1 catalytic domain insertions. Here, they describe a crystal structure of an unphosphorylated kinase-dead D334A *PhPINK1* catalytic domain, which reveals a 38 aa N-terminal helix extending from the N-lobe, not observed in the structure of the phosphorylated PINK1 catalytic domain, that interfaces with the C-terminal CTR helix, and holds the *PhPINK1* monomer in the inactive state. The functional importance of this new interface is underscored by the presence of a number of early onset PD mutation sites. This new structure also suggests that the N-lobe of PINK1 has to undergo significant conformational changes during PINK1 activation.

Structural analysis of a D357A kinase-dead mutant *PhPINK1* catalytic domain led to the unexpected discovery that the *PhPINK1* catalytic domain can dimerize and oligomerize, and cryo-EM analysis of D357A *PhPINK1* revealed a dodecameric structure, comprised of a two-ring stack each containing three dimers. This structure is disrupted when *PhPINK1* is allowed to autophosphorylate, consistent with phosphorylation preventing oligomerization. Next, the authors used the D357A *PhPINK1* dimer structure to elucidate the structural basis for autophosphorylation, showing that the OH-group of S202, which is autophosphorylated, in one monomer lies within 2.2 Å of the gamma phosphate of ATP in the other monomer. Interestingly, in the inactive dimer structure, the N-helix:CTR region is oriented so that both N-termini face the mitochondrial outer membrane (MOM).

To establish that S202 phosphorylation occurs in trans, they showed that WT *PhPINK1* catalytic domain can phosphorylate D334A *PhPINK1* in vitro on S202. Importantly, they showed that a S202A mutant *PhPINK1* can neither autophosphorylate nor phosphorylate

Ub, establishing the importance of S202 phosphorylation in PhPINK1 activation. They also showed that the phosphorylation of both S202 residues within a dimer is required for dissociation into monomers.

Through their structural analysis, they also uncovered an additional mode of PINK1 regulation resulting from the formation of a disulfide bond under oxidative conditions between Cys169 in one monomer with Cys169 in the second monomer. The Ub kinase activity of a C169A mutant PhPINK1 was less sensitive than WT PhPINK1 to H₂O₂ treatment, consistent with Cys169 being an oxidation target. Ala mutation of Cys360 in the ATP-binding pocket, which lies 15 Å away from Cys169, by itself did not render PhPINK1 less sensitive to H₂O₂ but a C169/360A double mutant was significantly more resistant to H₂O₂ treatment, consistent with Cys169 and Cys360 oxidation being functional oxidation sensors, but for different reasons. Consistently, the D357A mutant PhPINK1 catalytic domain, which is predominantly dimeric, showed a much greater propensity to form disulfide-bonded dimers than the WT PhPINK1, and dimers of human PINK1 could also be observed when hPINK1 was treated with H₂O₂.

In a final step, they partially crosslinked dimers and oligomers by H₂O₂ treatment of unphosphorylated PhPINK1 catalytic domain, and then allowed them to undergo autophosphorylation, before submitting them to cryo-EM structural analysis. By analysis of different molecular classes, they obtained two distinct types of structure - a symmetric PhPINK1 dimer trapped during the process of trans-autophosphorylation, with both molecules phosphorylated at S202. However, the second PhPINK1 molecule in each dimer adopted two different conformations, with one structure in a partially unfolded state suggesting that this molecule of phospho-PhPINK1 had been trapped in the process of undergoing a conformational change to become an active Ub kinase.

Finally, they used the new AlphaFold and ColabFold structure prediction algorithms to predict structures of hPINK1 monomers and dimers. The monomer structure predicted was very similar to that previously reported for the hPINK1-Parkin-Ubl complex, and the predicted hPINK1 dimer had a dimer interface almost identical to that reported here for PhPINK1, suggesting that the lessons learned from the PhPINK1 structural analysis described here are highly relevant to the mechanisms of activation and regulation of hPINK1.

This is a very convincing and well written story, which provides new insights into the mechanism of activation of the PINK1 kinase in repose to mitochondrial damage. The cryo-EM structure of the PINK1 dimer trapped in the act of transphosphorylation of Ser222 in one monomer by the other, is particularly exciting, and the dimer structure with both termini pointing towards the membrane implies that the organization of PINK1 on the surface of mitochondria, and in particular in the vicinity of the TOM transport pore will be important in its activation in response to damage and its ability to phosphorylate Ub at S65. In addition, the new evidence that PINK1 can be inhibited by oxidation of two Cys residues, suggests that PINK1 may be negatively regulated in the cell by oxidative stress, which given the key role of the mitochondrion in generating ROS could be important.

We would like to thank the Reviewer for the kind summary of our findings. We would like to clarify that the N-helix is not in an inactivating position, rather, it restrains potential

movement of PINK1 as it closely tethers PINK1 to the MOM. This is an important observation as it potential provides a mechanism to restrict ubiquitinated MOM substrate availability and targeting by PINK1. Secondly, the hPINK1-Parkin-Ubl complex was a prediction we received from AlphaFold2, and as such was not reported previously. The previous structure was that of *Ph*PINK1 bound to a ubiquitin mutant.

We have improved clarity in the text around these two issues.

Points:

1. Figure 2A/page 4: It is hard to see colors of the lines in the 66K peak – the three peaks could be labeled more clearly. The D334A Ph PINK1 mutant cannot autophosphorylate either, but does not form dimers or oligomers. Is this because the S202-D334 interaction is missing?

We thank the reviewer for pointing this out and we have changed the colouring as suggested. The D334A is a mutation of the HRD motif, an essential motif for kinase activity, and the mutant is indeed kinase dead as expected. The reviewer is entirely correct that we believe that we do not see dimer/oligomer formation, since we are losing the essential S202-D334 contact. Please see Extended Data Fig 8, which elaborates on this finding.

Interestingly, we can see the same loss-of-dimerisation also for a S202A mutant, which is consistent with our results and the Reviewer's interpretation.

2. Figure 2B/page 4: The dodecameric PhPINK1 structure is intriguing, even if it is unique to PhPINK1. The authors state that TcPINK1 was not found to oligomerize, but it is not clear if the contact residues that stabilize the dodecamer are present in TcPINK1, or, in one sense more importantly, in hPINK1. The overall dimensions of this dodecameric structure are not given, and if this structure could form in the cell how would these dimensions relate to those of the dimeric TOM complexes? With regard to the hexameric symmetry of this PhPINK1 complex, it is interesting to note that the PGAM5 phosphatase, another MOM resident protein, was recently shown to be able to adopt a hexameric structure in vitro (PMID: 30705304).

We were equally intrigued by the dodecameric structure, or the hexameric half-structure of the PhPINK1 assembly, especially when we could show that we were able to induce the hexamer in unphosphorylated wild-type PhPINK1. The Reviewer's point about PGAM5 is indeed intriguing. Whilst we were conscious when interpreting these observations that we were working with truncated PhPINK1, the N-termini transmembrane domains extend from the same surface of the hexamer, suggesting that the hexamer could assemble on the MOM. Whilst the relevance of this hexameric form of PINK1 in cells will be interesting to explore in future studies, several lines of evidence do not support its physiological relevance. Firstly, the interacting residues are generally poorly conserved, and are quite different in human PINK1; this would remove several hydrogen-bonding interactions and likely be detrimental to oligomerisation. Lack of conservation argues against a conserved mechanism of oligomerisation. We have included this analysis in new **Extended Data Fig. 4e**.

Secondly, there is no evidence to suggest formation of PINK1 oligomeric species in several papers that studied PINK1 on mitochondria (Lazarou et al, Dev Cell 2012; Okatsu et al, JBC 2013). In the main studies, PINK1 was part of a large (~700 kDa) complex as assessed by Native PAGE analysis. PINK1 (63 kDa) was present as a dimer in these studies, and other components of the complex were the TOM complex (TOM70, TOM40, TOM22, TOM20). The TOM complex on its own as purified from mitochondria, is roughly 500 kDa in these studies, and it shifts to a ~700 kDa complex in presence with PINK1. A PINK1 hexamer (or dodecamer) would be incompatible with these data. Often, these studies were done with stabilised, ie active PINK1 which we show to be monomeric.

We are gearing up to perform more detailed cell biological imaging and tomography studies on damaged mitochondria in future, and we will certainly try to see whether we can ever observe oligomers of PINK1.

3. Page 4: Did the authors test if an equivalent mutant form of hPINK1 can form oligomers? Can higher order oligomers be revealed by crosslinking studies, which might stabilize unstable oligomers?

We would love to do these studies, but we (and others) have to date been unable to produce human PINK1 in quantities large enough to perform any kind of biophysical or SEC/MALS studies.

Our cellular studies reveal a crosslinked dimer in cells (as discussed now in **Fig. 3g**) but not other kinds of clearly defined oligomers as we might expect from a higher oligomer formation. Together with the notion of distinct amino acid sequence as described above, we feel this is somewhat unlikely. With the advances in imaging and cryo-electron tomography, it will be interesting to investigate this as part of future work as mentioned above.

4. Page 5 top: This section could be written more clearly to indicate that S202 in one monomer is interacting with substrate binding site of the second monomer to facilitate trans-phosphorylation by the other monomer, and also to indicate that S202 is in a loop in the N-lobe.

We agree and have adjusted the text to explain this more clearly.

5. Figure 3B: Perhaps, the S202 and D334 residues/labels could be highlighted in some way so that it is easier for the reader to appreciate their interaction.

We agree and have added clearer labels.

6. Figure E6/page 6: Can more than one of these three Ser in hPINK1 be phosphorylated simultaneously, or is their phosphorylation mutually exclusive? There is some discussion of this in the legend, but it would interesting to know whether either S229 or S230 could be accommodated in the active site of the other monomer once S228 is phosphorylated, and whether having multiple phosphorylated Ser might lead to “stronger” activation.

The Reviewer raises an interesting question. During PINK1 activation in cells following mitochondrial depolarisation, PINK1 is indeed phosphorylated at more than one site in this loop it seems from our studies.

To address whether phosphorylation of these serines is important for PINK1 activity, we have now included a complete analysis of all possibilities for phosphorylation of these residues. Previously we showed AAA, SSS and ASS, and single phosphorylation site variants SAA, ASA, AAS; we now include the remaining two mutants SAS, SSA (new **Extended Data Fig 6h**). Analysis on Phos-tag gels indicated that each site can be phosphorylated in isolation and in context of other phosphorylation events. For the specific question asked by the reviewer: In the cases of SAS and SSA, there are multiple Phos-tag bands including a higher species that is indicative of a doubly phosphorylated version. Importantly, as shown in analysis of ubiquitin phosphorylation, the data confirm the requisite role for Ser228 phosphorylation to turn PINK1 into a ubiquitin kinase. No other site can substitute for Ser228, in singly or doubly phosphorylated forms of hPINK1.

With regards to access, we now discuss how the dimer asymmetry, and the fact that PINK1 can undergo a conformational change whilst in the dimer (EM structure Fig 4) unravels the N-lobe and α C helix in situ, which would likely be necessary and sufficient to get a second and third phosphate onto the Ser202-loop.

8. Pages 7 and 9: As the authors indicate, the fact that the C169A and C360A PhPINK1 mutants are still at least partially sensitive to H₂O₂ (in Figure 4C, PhPINK1 seems to be as sensitive as WT PhPINK1 to H₂O₂) suggests oxidation of either Cys to sulfenic acid or a higher order oxidation state? Is inhibitory, but particularly for Cys169 one would like to know what dictates the formation of a disulfide bond versus sulfenic acid. Can PINK1 dimers be detected in cells treated with H₂O₂?

This was a great question that we focussed on during the revision process (see also response to Reviewer 1). Indeed, we had shown for recombinant PhPINK1 (Fig. 4c) that mutations of the two key Cys (a) reduce inherent kinase activity and (b) render PhPINK1 less susceptible to inhibition by H₂O₂. We also confirmed that the same disulphide-linked dimers can be detected in cells stimulated to undergo mitophagy with OA following treatment with H₂O₂. On these data, we proposed a potential regulatory mechanism for PINK1.

We now extend these findings as follows:

- 1) We show that the inhibitory effect of H₂O₂ on PhPINK1 is reversed by DTT. This was an important result as it revealed that PhPINK1 is not undergoing irreversible oxidation (SO₂H or SO₃H), but that the inhibitory species formed are either Cys sulphenic acid (SOH) or Cystines/Cys disulphides. The result is also interesting from the perspective of regulation, as it suggests reversibility of an inhibitory mechanism, as would be anticipated (new **Extended Data Fig 9c**, also see new **Fig. 5d-f** for in vitro experiments on hPINK1 variants).
- 2) We next went into cells to show this oxidative mechanism for human PINK1. Here, we exploited a new insight relating to PINK1 evolution, already discussed in response to Reviewer 1. Consistent with our observations with recombinant *PhPINK1*, human

PINK1 C166A or C166S are less active in phosphorylating ubiquitin in cells (new **Extended Data Fig 9h**). However, compound mutation of Ser167 to Asn, equivalent to fish PINK1, recovered activity in hPINK1 (hPINK1 SN) (new **Extended Data Fig 9h**). This active variant, enabled us to test the role of Cys166 in the oxidative control of hPINK1.

To do this, we extracted mitochondria-enriched membrane fractions from PINK1^{-/-} HeLa cells ectopically expressing hPINK1 or hPINK1 SN after 2h Oligomycin/ Antimycin A (OA) treatment. The mitochondria-enriched fractions were then assessed for their ability to phosphorylate recombinant ubiquitin in the presence or absence of H₂O₂.

The results (new **Figure 5d-f**) indicate that the activity of wild-type PINK1 is reduced with H₂O₂ pre-treatment, whilst the PINK1 SN variant was not. Importantly, the H₂O₂ inhibitory effect on wild-type PINK1 was reversed with DTT treatment.

- 3) When analysed on non-reducing gels, wild-type PINK1 forms disulphide-linkable dimers, but not the active PINK1 SN mutant (new Fig 5e), confirming that Cys166 is key to disulphide-linkage of the dimer.

These data are consistent with our previous findings for recombinant PhPINK1 (Fig 4 in previous version, now Fig 5b and Extended Data Fig 9). Moreover, it shows that hPINK1 activity is regulated reversibly by oxidation of Cys166. We are very excited by these new insights, that pave the way for future experiments in cell and animal models to understand the physiological relevance of this oxidative control of PINK1, mitophagy and Parkinsonism. As discussed in response to Reviewer 1, there are two competing mechanisms for how Cys166 oxidation can regulate PINK1 activity: (1) through disulphide formation of a dimer or potentially even in *cis*, (2) via affecting ubiquitin binding. The fact that the PINK1 SN mutant rescues activity of PINK1, supports the latter mechanism, and clearly indicates that this area of PINK1 is essential for ubiquitin binding (as we would expect from the structure). However, as we discuss in our model for PINK1 activation (new **Fig. 5g**), formation of a disulphide-linked dimer could contribute to loss of PINK1 activity. It is possible that both mechanisms are working in concert, and our experiments cannot currently distinguish between these scenarios. Other factors, e.g. the pH of the cytosol, could also contribute to this regulatory mechanism.

7. Page 7, line 213: From Figure E8 it appears that these partially crosslinked WT PhPINK1 oligomers were a result of H₂O₂ oxidation. If this is the case, it should be indicated in the text, and also why this oligomer is not simply a dimer, which what the data in Figure 4 suggests it should be.

We were also surprised by the large number of cross-linked species apparent on the non-reducing gel, which had formed in the phosphorylated *PhPINK1* sample treated with H₂O₂.

Structurally, there are no other obvious Cys residues at an interface within the *PhPINK1* oligomer, though we note that some loops remain disordered. PINK1 molecules in the oligomer would be closely packed, potentially enabling alternative peroxide-induced cross-links. We also noted that the micrographs had more unidentified (and unused) particles in

them, which we associated with the crosslinking. We were delighted that the used particles nonetheless revealed the intriguing insights discussed in Fig. 4.

8. Page 7: Since ROS generated by the ETC and exiting from the mitochondrion will be present at very high concentrations locally, one might expect these Cys to be constitutively oxidized, unless GSH levels are also very high in the vicinity of the MOM.

We agree with this comment, and it is this consideration that has us intrigued about a potential for regulation of PINK1 by oxidation. We now show that PINK1 oxidation is reversible, which we consider an important requirement for regulation. We have also screened a variety of mitophagy stimuli, including depletion of GSH (see response to Reviewer 1 and **Response Fig 1** above), which indicated formation of disulphide-linked dimers in the absence of exogenous oxidant, but were confounded by cellular toxicity. Together our new data further support reversible oxidation as a new regulatory mechanism for PINK1 and we now propose that PINK1 may monitor the oxidation state of the cell.

10. Figure 3A/Figure 5F/page 8: With regard to the authors' modeling of the PINK1 activation process, it is not clear whether the TOM dimer complex in the lower panel of Figure 3A is drawn to scale. There are Cryo-EM structures of the human TOM complex (125 Å across) that could be included and possibly docked onto the PhPINK1 dimer (or even dodecamer) to see whether this model is realistic.

This comment was similarly raised by Reviewer 1 and we have discussed it there. We note that the scale of the TOM complex is not entirely off in **Fig. 3a**. We now provide a manual docking model, for scale purposes, in **Extended Data Fig. 5d**. There is a lot of activity in this field (see recent Baker lab paper from yeast, which predict a direct interaction between Tom20 and Tom22). The human TOM complex EM structures lack TOM20, a main interactor for PINK1, and modelling Tom complexes with the new insights from the Baker lab suggests that TOM20 may sit across the pore (potentially clashing with PINK1?). We are reluctant to overemphasise the docking model and should instead work to achieve a complex structure.

Minor point: Page 7, line 204: The authors need to indicate that OA (okadaic acid) is a PP2A inhibitor, and spell out why it was used.

We apologise for not explaining this carefully. OA refers to oligomycin/antimycin A, which are inhibitors of the electron transport chain complexes V and III, respectively, and which serve to depolarise mitochondria, to trigger PINK1 accumulation and mitophagy.

We have now added this definition in the text. Different groups in the field use the abbreviations AO or OA, we have opted for the latter for consistency.

Reviewer #3 (comments to authors):

This manuscript uses X-ray crystallography and cryo-EM to study regulation of head lice PINK1 kinase. It identifies a) the location of the N-terminal helix, b) an unexpected dimer in the non-phosphorylated state that explains why Ser202 (Ser228 in human) is the main regulatory phosphorylation c) a redox switch mediated by C169 and d) the structural

consequences of Ser202 auto-phosphorylation, nicely validated by alphafold2 prediction. The combination of the dimer structure and the N-terminal helix location suggests how PINK1 oriented relative to TOM complex on the membrane. The relevance of the dimerization, the phosphorylation at Ser228 and the occurrence of the disulfide bound dimer are validated in human PINK1 in HeLa PINK1 $-/-$ cells by expression of PINK1 variants. Much of this work is enabled by an unexpected dodecamer found in the D357A mutant of PhPINK1, which enabled the high resolution cryo-EM analysis.

It is an elegant study and the experiments are well presented, with gorgeous figures. The main finding in my opinion is the identification of the PINK1 dimer, as this made it possible to bring mechanistic clarity to the activation by Ser202 phosphorylation and the PINK1 location relative to the mitochondrial membranes. It also suggested the presence of a potential redox switch. Such a redox switch at that location on mitochondria is indeed evocative.

We thank the reviewer for the summary of our findings and for the below highly constructive and helpful comments.

Comments

It remains to be seen what the functional consequences of the redox switch are, as validation was limited to showing that it can occur in an overexpression system. It is clear that the authors attempted to limit overexpression artefacts, but it would be good to add a control to show how their expression compares to WT expression.

The reviewer is correct that PINK1 is used in overexpression systems in cells. We are using a mutated CMV promotor (labelled in Response Fig. 2 as CMVd1-d3) to limit PINK1 expression to near endogenous levels. Despite this, PINK1 expression was still higher than endogenous PINK1 (Response Fig. 3). Importantly, however, ectopic expression of PINK1 did not trigger mitophagy unless cells were treated with OA, and so enabled us to assess PINK1 activity via ubiquitin phosphorylation for our mechanistic studies.

Response Fig. 2. Mutated CMV promotors reduce PINK1 expression. We use CMVd3 throughout cell-based experiments in this manuscript.

Our new experiments with PINK1 SN (see response to Reviewer 1 and 2) have now allowed us to further support the role of oxidation in regulating human PINK1 when expressed in cells.

Response Fig. 3. Final comparison of endogenous and expressed PINK1 from CMV-d3 in direct comparison. Expression of PINK1 from the CMVd3 promotor does not trigger phospho-ubiquitin generation without OA treatment.

The occurrence of the dodecamer is of course fortuitous as it allowed high resolution cryo-EM analysis. The success of that analysis shows this is a very well-behaved dodecamer, and then one really wonders if it could have a functional role after all. The manuscript rather easily dismisses that option, it would at least be necessary to discuss the reasons why and possibly the processing of the PINK1 protein in more detail

This was also raised by Reviewer 2, please see key arguments there. We would agree that a dodecamer (or perhaps more likely, a hexamer, in which all N-helices would be in the MOM) may be a structure observable for *Ph*PINK1 with depolarised mitochondria. In human or *Tc*PINK1, key interface residues are mutated (in two cases, to opposing charges between insect and human PINK1) – see new **Extended Data Fig. 4e**. If dodecamer (hexamer) formation was physiologically important, we would expect to see better conservation.

Also, hPINK1 has been analysed in quite some detail using native PAGE analysis – a 600 kDa complex that we can easily generate by dephosphorylation or crosslinking, has never been observed (although, these experiments would have been done in PINK1-stabilising conditions that activate PINK1).

The validation of the dimer in the human cells is primarily through its disulfide formation, although not strictly necessary, some other form of validating the existence of the dimer would be nice.

We felt that our identification of the crosslinked dimer, its removal by a point mutation, the consistent AlphaFold2 predicted hPINK1 dimer, as well as the bioRxiv manuscript by the Trempe group on *Tc*PINK1 dimer, were good evidence that dimer formation is similar or even identical in hPINK1/across species.

We did however address the reviewer comment in a different way, that we feel expands the manuscript. We have introduced, in hPINK1, 11 patient mutations found in regions we would predict to affect dimer formation especially in the structurally conserved activation

segment. We performed oxidative cross-linking / dimer formation, PINK1 autophosphorylation and ubiquitin phosphorylation experiments with these mutants. The results are shown in new **Extended Data Fig. 5h** and are indeed very interesting.

We found that most mutants are unable to dimerise (crosslink), and are deficient in auto-and ubiquitin phosphorylation (F485L, C388R, P416R and others).

Some mutants remain somewhat proficient in dimer formation but deficient in autophosphorylation or ubiquitin phosphorylation (G386A, which like the control mutant D384A, is part of the kinase DFG motif), or proficient in dimer formation and somewhat autophosphorylated, but deficient in ubiquitin phosphorylation (G409V). We would classify these as catalytic mutants.

The fact that dimer formation is readily observed by H₂O₂ treatment of cells, could be an interesting additional tool in future studies of hPINK1 mutated patient samples.

With respect to the H₂O₂ experiment, what is the band that is induced at 100 kD by H₂O₂?

This was a great question. Indeed, we see the 100 kDa cross-linking product in all experiments treated with H₂O₂, with or without the presence of Cys166. The formed complex could hence be a relevant physiological binding partner.

We performed several experiments to identify this band, but unfortunately did not return a positive ID as of yet. With 100 kDa size, the protein would need to be in the common range of 40-50 kDa (hPINK1 is 63 kDa). We excluded that the additional component is a known protein of the TOM complex, including TOM70, TOM40 or TOM22 (See **Response Fig. 4A**). We could identify some potential cross-linked bands for TOM20, which would be very consistent with the literature (Lazarou et al, Dev Cell 2012), but closer inspection suggested that the sizes did not match PINK1 bands (See **Response Fig. 4B**).

A first attempt at mass-spectrometry did not reveal any sensible or enriched binding partners either (but also did not show above-background peptides for PINK1). We will need to significantly scale up these experiments, and hope for a positive ID. This will be done as a follow-up study, and may give insights into new complexes PINK1 may form, or validate known ones.

A

B

Response Figure 4. Candidate approach for 100 kDa crosslinked band of PINK1 after H₂O₂ treatment. A) cross-linking, with BMOE, identifies the PINK1 dimer and the 100 kDa band also observed with H₂O₂. Flag-tagged PINK1 cross-linked complexes were precipitated and blotted for known components of the TOM complex, including TOM20, TOM22, TOM40 and TOM70. TOM20 showed a promising pattern of bands in the IP. B. Repeat experiments focussing on TOM20 showed cross-linked IPed material, however the size did not match the most prominent, 100 kDa band. We hence are not convinced that the 100 kDa band contains TOM20.

Why is TcPINK1 affected by H₂O₂ if it lacks ser 202

We assume the Reviewer means that *TcPINK1* lacks Cys169, yet is still affected in its activity by H₂O₂ (**Extended Data Fig 9b**). *TcPINK1* is still affected by H₂O₂ as there are 9 Cys residues in the *TcPINK1* construct used, including the equivalent Cys for Cys360 mutation of which has a mild effect for *PhPINK1*, similar to the effect seen for oxidated *TcPINK1*.

For the last structure it would be good if local resolution in the region that changes between upon phosphorylation could be made explicit

We have now included zoomed in local resolution maps for the extended and kinked αC helices in phosphorylated *PhPINK1* in **Extended Data Fig. 7f**.

There is a nice table for refinement of the X-ray structure but nothing for the cryo-EM structures, please add an equivalent table 1.

We are sorry this Table was hidden in the respective file on a different page, it was and remains part of the Supplementary Tables pdf file, as **Supplementary Table 2**.

The attempt to keep to length restrictions seems to have really hurt precision and clarity in many places; the main issues that were most distracting are

We thank the Reviewer for taking the time to improve presentation, much appreciated. We addressed their comments in full in the revised manuscript.

1) Details of the specific experiment: all cryo-EM is presumably performed with D357A, but this is only noted in a few places and omitted elsewhere;

The first cEM structure was of *PhPINK1* D357A, but the second one was of wild-type *PhPINK1*, phosphorylated and cross-linked. We appreciate this detail could be lost, and we have tried to make this more explicit in the Figures. We have now included the workflow for the second EM structure to the top of **Fig. 4 (a-d)**.

2) Some kinase specific terms could use better introduction in a general journal 'the HRD', C and R spines, 'a second kinase inactivating mutant' the phosphorylation sites, acceptor site

Yes, we agree. We added an explanatory Figure labelling these kinase motifs in **Extended Data Fig 1d**.

3) The Parkinson introduction has some errors probably due to brevity: it suggests that all EOPD have known mutations, also suggests that only EOPD has no disease-modifying treatment. In fact as far as I know it is only one of the EOPD mutations for which there is hope of a disease modifying treatment (GBA mutations), whereas idiopathic Parkinson is not different from EOPD (except sometimes faster in progression)

We agree this was imprecise. We have adjusted our introduction accordingly.

4) It would be helpful to indicate more precisely which structure is what, e.g. by giving pdb files in the schematic model of figure 5F, explaining what PDB is meant in line 232 where 'the PhPINK1:Ub complex' suddenly pops up, uniprot file numbers for the alphafold predictions,

We have added this more explicitly. We have used Colabfold to use Alphafold2 predictions to predict the hPINK1 dimer, so these structures are not immediately downloadable.

5) Precision in general,

- a. Line 62: divergent Ser/Thr protein kinase: divergent relative to what?
- b. Line 64: that contain only two insertions : relative to what? I guess this should be 2 of these four, but then you immediately wonder in which lobe;
- c. line 66: adopt typical folds typical of what? typical kinase? or specific to these?

- d. Line 102: A second kinase inactivating mutation second relative to what?
- e. Line 130 Modelling ATP places the γ -phosphate: its gamma-phosphate
- f. Line 131: resembling the PhPINK1:Ub: grammar wrong, not clear what resembling refers to
- g. Line 188: this inhibition could partially be restored with C169A mutation (Fig. 4c). I guess they mean prevented....
- h. Line 213: a partially crosslinked wild-type PhPINK1 oligomer: a few more words to explain would be useful
- i. Line 238 there are two intact contacts, which are subsequently resolved by phosphorylation at Ser202 and α C helix kinking : it is not clear what is meant here
- j. Line 248: phosphorylated ser 202 leads to : I'm not sure this is inference or proven
- k. Line 261: extra words, remove in the : "even in the without hPINK1 phosphorylation."

We adjusted these issues. Thanks for pointing them out.

Reviewer Reports on the First Revision:

Ref # 1

N/A

Ref #2

N/A

Ref #3

The authors have addressed all outstanding questions. The new data on the human oxidation in fig5/extended fig 5 provide nice validation. This manuscript provide important new insights on PINK1 regulation.

I do not have strong views on what model of the TOM1 complex (as shown in extended figure 5d (vs 3A) is shown as long as the legend is clear, but in either case it could be beneficial to write 'computation model' in the actual figure to prevent this getting too much weight.

Author Rebuttals to First Revision:

Referees' comments:

Referee #3 (Remarks to the Author):

The authors have addressed all outstanding questions. The new data on the human oxidation in fig5/extended fig 5 provide nice validation. This manuscript provide important new insights on PINK1 regulation.

I do not have strong views on what model of the TOM1 complex (as shown in extended figure 5d (vs 3A) is shown as long as the legend is clear, but in either case it could be beneficial to write

'computation model' in the actual figure to prevent this getting too much weight.

We thank the Reviewer for their assessment.

We have added a label to Extended Data Fig. 5d, indicating that the PhPINK1 dimer was manually docked onto a recent TOM complex structure.